*Method*

# Optochemical profiling of NMDAR molecular diversity at synaptic and extrasynaptic sites

Antoine Sicard [iD][1], Meilin Tian [iD][1,3], Zakaria Mostefai[1], Sophie Shi[1], Cécile Cardoso[1], Joseph Zamith[1], Isabelle McCort-Tranchepain [iD][2], Cécile Charrier [iD][1], Pierre Paoletti [iD][1,4][✉] & Laetitia Mony [iD][1,4][✉]

## Abstract

Neurotransmitter receptors are critical for neuronal communication. They often form large multimeric complexes that differ in their subunit composition, distribution, and signaling properties. *N*-methyl-D-aspartate receptors (NMDARs), a class of glutamate-gated ion channels with essential roles in brain development and plasticity, co-exist as multiple subtypes, with GluN2A diheteromers, GluN2B diheteromers, and GluN2A/GluN2B triheteromers prevailing in the adult forebrain. Studying individual subtypes in native tissues with subunit stoichiometry resolution remains challenging, and the relative abundance and subcellular distribution of these subtypes remain controversial. Here we develop and use the photochemical tool Opto2B for specific and reversible modulation of GluN2B diheteromers, while leaving other receptor subtypes (in particular GluN2A/GluN2B triheteromers) unaffected. Using Opto2B, we characterize the differential contribution of GluN2B diheteromers to synaptic and extrasynaptic NMDAR pools during mouse development. Our results suggest that GluN2A receptors predominate in both pools in adult hippocampal CA1 pyramidal cells, with no preferential contribution of GluN2B diheteromers to extrasynaptic currents, challenging the common view that GluN2A and GluN2B NMDARs segregate in synaptic and extrasynaptic compartments, respectively. Our study addresses long-standing questions on extrasynaptic NMDARs and paves the way for interrogating NMDAR signaling diversity with unprecedented molecular and spatio-temporal resolution.

**Keywords** Extrasynaptic Receptors; Glutamate; NMDA Receptors; Optopharmacology; Synapse
**Subject Categories** Membranes & Trafficking; Methods & Resources; Neuroscience

## Introduction

With their ability to convert a chemical message—the presence of the neurotransmitter—into an electrical signal—a change in membrane potential—neurotransmitter receptors are the linchpin of neuronal communication (Smart and Paoletti, 2012). They also constitute therapeutic targets of prime importance against neurological and psychiatric disorders (Geoffroy et al, 2022; Lemoine et al, 2012). Neurotransmitter receptors assemble as large macromolecular complexes formed by the association of multiple subunits, usually encoded by large multigenic families. The combinatorial association of constitutive subunits into functional receptors generates a vast diversity of receptor subtypes differing in their molecular composition, localization and functional properties. For instance, GABA$_A$ receptors, which mediate the bulk of inhibitory neurotransmission, associate into pentamers from no less than 19 different subunits giving rise to tens of different GABA$_A$ receptor subtypes in the CNS (Sente et al, 2022; Sigel and Steinmann, 2012; Sun et al, 2023). Similarly, ionotropic glutamate receptors (iGluRs), which mediate the bulk of excitatory neurotransmission, assemble as tetramers from a repertoire of several different subunits resulting into a wide variety of receptor subtypes (Hansen et al, 2021; Yu et al, 2021; Zhao et al, 2019). Individual neurons or synapses may assemble distinct receptors subtypes across spatial locations, developmental stages, and physiological or disease states. There is little doubt that this large variety allows specific receptor subtypes to engage into distinct neuronal functions. This diversity also holds strong promise for next generation therapeutics through the development of target-specific precision drugs more efficient and better tolerated (Geoffroy et al, 2022; Sieghart and Savić, 2018). Nevertheless, we currently lack a clear understanding of the physiological and pathological relevance of the large plurality of neurotransmitter receptor subtypes. Current methodologies based on pharmacology and genetic modifications, although powerful, usually have limitations in terms of molecular specificity and spatiotemporal resolution. Therefore, new strategies are required to better discriminate between receptor subtypes, and dissect their distribution and associated signaling pathways.

[1]Institut de Biologie de l'Ecole Normale Supérieure (IBENS), Ecole Normale Supérieure, Université PSL, CNRS, INSERM, 46 rue d'Ulm, 75005 Paris, France. [2]Université Paris Cité, Laboratoire de Chimie et Biochimie Pharmacologiques et Toxicologiques, CNRS UMR8601, 75006 Paris, France. [3]Present address: Shenzhen Institute of Advanced Technology, Chinese Academy of Sciences, 518055 Shenzhen, China. [4]These authors contributed equally to this work as senior authors: Pierre Paoletti, Laetitia Mony.
[✉]E-mail: pierre.paoletti@ens.psl.eu; laetitia.mony@ens.psl.eu

By combining the power of light, pharmacology and genetics, optogenetic pharmacology offers means to overcome some of these limitations (Berlin and Isacoff, 2017; Broichhagen and Levitz, 2022; Hüll et al, 2018; Kramer et al, 2013; Paoletti et al, 2019). This approach consists in tethering a photoswitchable ligand to a desired receptor subunit. Upon illumination with different wavelengths, the ligand can reach its binding site on the receptor target, or undock from it, resulting in pharmacological modulation of the receptor activity. Optogenetic pharmacology affords exquisite molecular and spatiotemporal resolution, as well as on-demand reversibility (Berlin and Isacoff, 2017; Hüll et al, 2018; Kramer et al, 2013; Paoletti et al, 2019).

NMDA receptors (NMDARs), a subfamily of iGluRs, play essential roles in brain development and function (Paoletti et al, 2013). Normal NMDAR signaling controls synaptic plasticity, a cellular substrate of learning and memory. Conversely, abnormal NMDAR function is deleterious causing neuronal injury, cognitive deficits and maladaptive behaviors (Paoletti et al, 2013; Hanson et al, 2024; Zhou and Sheng, 2013). At the molecular level, NMDARs are obligatory hetero-tetramers containing two GluN1 subunits and two GluN2 or GluN3 subunits, of which there are six versions (GluN2A-2D and GluN3A-B). NMDARs can assemble as diheteromers (with two identical copies of GluN2 or GluN3 subunits) or triheteromers (with two different copies of GluN2 or GluN3 subunits) that co-exist in native tissues, with over ten receptor subtypes identified to date (Hansen et al, 2021; Paoletti et al, 2013; Sheng et al, 1994; Stroebel et al, 2018). In the adult forebrain, GluN2A diheteromers (GluN1/GluN2A), GluN2B diheteromers (GluN1/GluN2B) and GluN2A/GluN2B triheteromers (GluN1/GluN2A/GluN2B) predominate, at mixtures that differ according to cell type, brain region and developmental stage (Dupuis et al, 2023; Hansen et al, 2021; Paoletti et al, 2013; Stroebel et al, 2018). The GluN2A to GluN2B subunit ratio has been shown to be an essential parameter controlling key neural processes, such as critical periods, bidirectional synaptic plasticity, as well as excitotoxic damage (Cui et al, 2013; Paoletti et al, 2013; Sanz-Clemente et al, 2013; Yashiro and Philpot, 2008). Yet the respective contribution of GluN2A and GluN2B diheteromers relative to GluN2A/GluN2B triheteromers in these processes remains controversial (Shipton and Paulsen, 2014; Parsons and Raymond, 2014). In addition, the subcellular distribution of these various receptor subtypes, i.e., their distribution between synaptic and extrasynaptic compartments, is also contentious. In the adult forebrain, many studies argue for an enrichment of GluN2A-containing NMDARs (GluN2A-NMDARs) at synapses, and an enrichment of GluN2B-NMDARs at extrasynaptic sites (Fellin et al, 2004; Papouin et al, 2012; Scimemi et al, 2004; Tovar and Westbrook, 1999; Köhr, 2006; Groc et al, 2009). However, other studies found no difference in NMDAR subunit composition between synaptic and extrasynaptic sites (Petralia et al, 2010; Harris and Pettit, 2007; Mohrmann et al, 2000; Le Meur et al, 2007). Such controversies likely stem, in part, from inherent limitations of genetic and pharmacological approaches, which hamper drawing unambiguous conclusions about the prevalence of the different receptor subpopulations. Indeed, genetic manipulation of GluN2A or GluN2B subunits indifferently affects diheteromers and triheteromers. Similarly, available pharmacological agents, such as the 'GluN2A selective' inhibitors zinc and TCN-201, or the 'GluN2B selective' inhibitor ifenprodil (and derivatives), poorly discriminate between their respective diheteromers and triheteromers (Hansen et al, 2014; Hatton and Paoletti, 2005; Stroebel et al, 2018, 2014).

In this work, using optogenetic pharmacology, we designed a new photoswitchable tool targeting NMDARs with subunit stoichiometry resolution. This tool (coined 'Opto2B') allows specific and reversible manipulation of GluN2B diheteromers independently of GluN2A/GluN2B triheteromers (and GluN2A diheteromers). Because Opto2B is based on an allosteric, rather than orthosteric (i.e., targeting the agonist binding sites), photoswitchable ligand, it also minimally interferes with the normal pattern of receptor activation, thus preserving endogenous signaling. Using opto2B, we defined the contribution to hippocampal CA1 NMDA currents of GluN2B diheteromers relative to GluN2A-containing receptors (GluN2A diheteromers and GluN2A/GluN2B triheteromers), both at extrasynaptic and synaptic sites and during brain development up to adulthood. Our study reveals that, early on in postnatal development, GluN2A subunits are excluded from extrasynaptic sites but incorporate readily in synaptic NMDARs, while in adult CA1 pyramidal cells, GluN2B diheteromers are expressed at low levels at synaptic but also extrasynaptic sites.

# Results

## Design of photocontrollable GluN2B-NMDARs using optogenetic pharmacology

GluN2B-NMDARs are selectively potentiated by polyamines (spermine and spermidine), a class of positively-charged compounds binding at the interface between the lower lobes of GluN1 and GluN2B NTDs (Geoffroy et al, 2022; Mony et al, 2011; Williams, 1997). Polyamines promote GluN1 and GluN2B NTD lower lobe apposition, a motion coupled to a downstream rolling motion between the two GluN1/GluN2B ABD dimers, which in turn leads to an increase in the receptor channel activity (Mony et al, 2011; Esmenjaud et al, 2019; Tajima et al, 2016; Tian et al, 2021) (Fig. 1A). We decided to target this allosteric site to modulate GluN2B-NMDARs with light. To this aim, we designed a photoswitchable spermine derivative called MASp (for Maleimide-Azobenzene-Spermine), a tri-partite ligand composed of (Fig. 1B; Appendix Text S1—Spectra S1,S2): (i) a cysteine-reactive maleimide moiety allowing covalent attachment to a cysteine; (ii) an azobenzene group (the photoswitch), which can reversibly alternate between an extended *trans* and a bent *cis* configuration depending on the illumination wavelength; and (iii) a spermine moiety (the ligand), acting as the pharmacological head-group. MASp was best switched from *trans* to *cis* by 365 nm light illumination, and from *cis* to *trans* by wavelengths of ~500 nm, as revealed by UV-visible spectroscopy (Fig. EV1A). Hence, thereafter, we used the wavelength of 365 nm to optimally photoswitch MASp from *trans* to *cis*, and wavelengths from 490 to 530 nm to induce *cis* to *trans* conversion. Further characterization of MASp physicochemistry revealed efficient and reversible photoswitching as well as bistability, properties highly suitable for biological use (Fig. EV1B,C; Appendix Text S1- Spectra S1G,H).

We first tested potential background effects of MASp labeling on wild-type (WT) GluN1/GluN2B receptors. Currents from WT GluN2B diheteromers labeled with MASp were not affected by 365 or 490 nm illumination (Fig. 1C–E). However, MASp labeling by itself induced a marked (~threefold) increase in receptor channel open probability (Po), as assessed by measuring the kinetics of

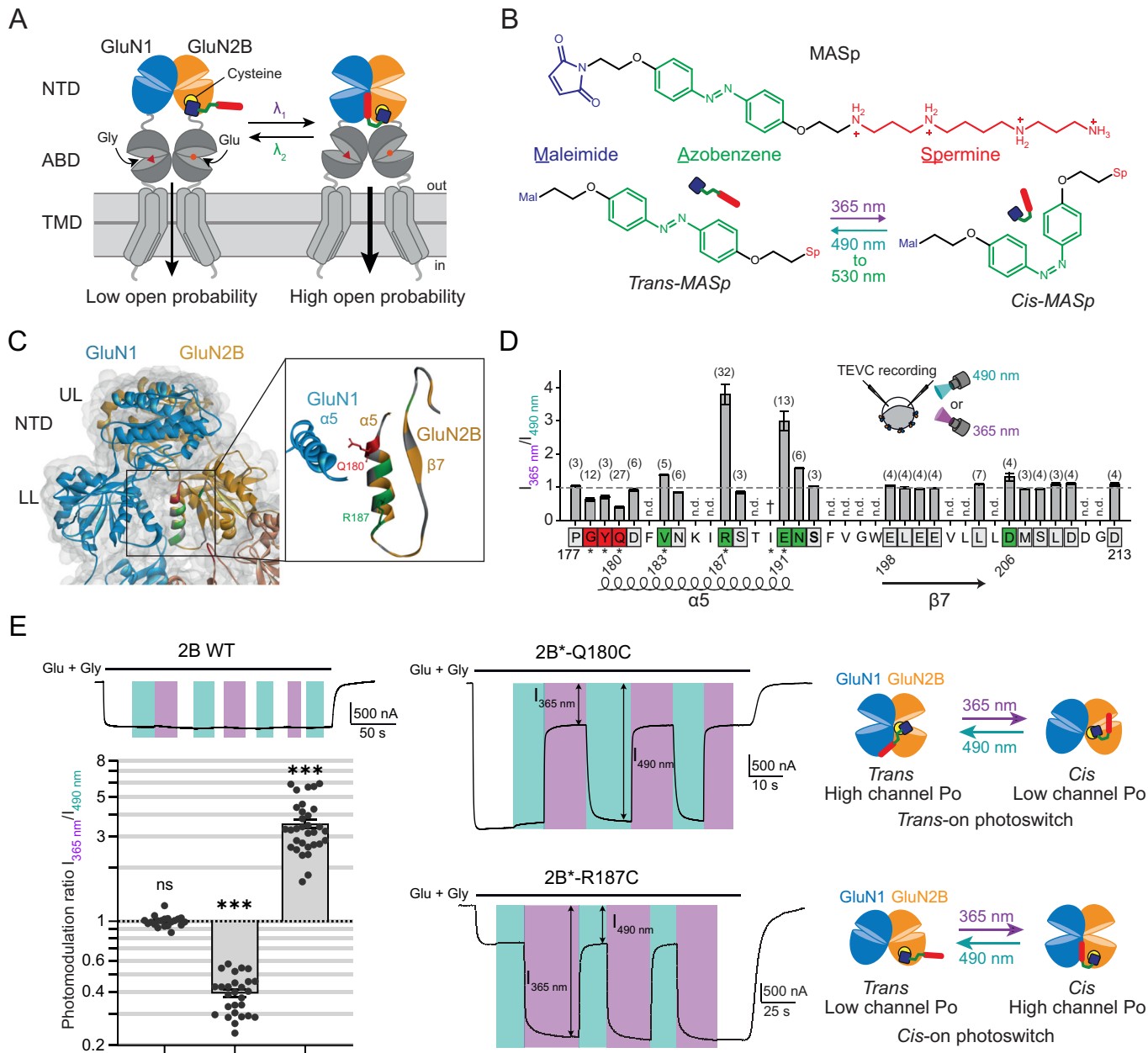

current inhibition by MK-801, an NMDAR open channel blocker (Chen et al, 1999; Blanke and VanDongen, 2008; Gielen et al, 2009; Talukder et al, 2010; Hansen et al, 2013) (Fig. EV1E,F). This effect was due to conjugation of MASp to the endogenous cysteine C395, located in the short linker segment between the NTD and the ABD of the GluN2B subunit (Fig. EV1E). Indeed, MASp-induced Po increase was fully abolished in mutant GluN1/GluN2B-C395S receptors, in which C395 was rendered non-reactive (Fig. EV1G,H). To avoid effect of MASp on WT GluN2B-receptors, we used in most of subsequent experiments receptors containing the background mutation GluN2B-C395S (noted hereafter GluN2B*). Importantly, the C395S mutation by itself had no significant impact on GluN2B-receptor channel Po (Fig. EV1H).

We next searched for a MASp attachment site on the GluN2B subunit for optimal photomodulation. Ideally, MASp tethered to the GluN2B subunit should be inert in one configuration (*cis* or *trans*), while inducing receptor potentiation in the other configuration (Fig. 1A). To identify potential MASp attachment sites, we performed individual cysteine scanning mutagenesis targeting solvent-accessible residues of the GluN2B α5-β7 region proposed to participate in the spermine binding site (Mony et al, 2011; Tajima et al, 2016; Tian et al, 2021) (Fig. 1C,D). Screening was performed in Xenopus oocytes expressing various cysteine mutants of the GluN2B subunit, and labeled with MASp (see "Methods" and Fig. EV1D). Because spermine potentiation displays strong pH-dependence (Mony et al, 2011; Traynelis et al, 1995), functional

◄

**Figure 1. Engineering light-sensitive GluN2B-NMDARs.**

(A) Schematic of a dimer of GluN1 and GluN2B subunits. GluN1 and GluN2B NTDs are colored in blue and orange, respectively. To photo-enhance GluN2B-NMDAR activity, the idea is to introduce a photoswitchable polyamine derivative in proximity of the polyamine positive allosteric modulation site, at the interface between the lower lobes of GluN1 and GluN2B NTDs (Mony et al, 2011). (B) Chemical structure of MASp, a photoswitchable spermine derivative, in its *trans* and *cis* configurations. (C, D) Screening for MASp labeling positions yielding photomodulation. (C) Left, X-ray structure of an NTD dimer in its "active" form (pdb 5FXG, (Tajima et al, 2016)) with the positions tested for photomodulation highlighted on the GluN2B subunit. Right, close-up view of the NTD lower lobe—lower lobe interface. Positions were screened along the α5 helix and β7 strand of the NTD of the GluN2B subunit. Screened labeling positions are color-coded as follows: grey, positions yielding no or little (<30%) photomodulation of MASp-labeled GluN1/GluN2B; red, positions for which UV light induces an inhibition >30%; green, positions for which UV light induces a potentiation >30% (D). UL upper lobe, LL lower lobe. (D) Summary of the amount of photomodulation (expressed as the current ratio between 365 and 490 nm light) obtained on NMDARs containing a GluN2B subunit mutated with a cysteine at the shown positions and labeled with MASp. For each position, the number of oocytes tested is indicated in parenthesis. n.d., not tested; †, no expression of the cysteine mutant; *, the cysteine mutation was made on the GluN2B-C395S background. Same color code as in (C). (E) Current traces from oocytes expressing GluN1/GluN2B WT, GluN1/GluN2B-Q180C-C395S (2B*-Q180C) or GluN1/GluN2B-R187C-C395S (2B*-R187C) receptors following application of glutamate (Glu) and glycine (Gly) (100 µM each) and under illumination with 365 (violet bars) or 490 nm light (blue-green bars), and summary of the photomodulation ratios ($I_{365\,nm}/I_{490\,nm}$) for these three constructs. Right, proposed mechanism of MASp light-dependent action when bound to Q180C (top) or R187C (bottom) (see Fig. EV2A,B). n.s., $P > 0.05$; ***$P < 0.001$; multiple one sample Wilcoxon tests against the value 1, $P$ values were adjusted for multiple comparisons using Bonferroni correction. Average basal current values ($I_{490\,nm}$ for WT and 2B*-R187C, $I_{365\,nm}$ for 2B*-Q180C) are indicated in Appendix Table S3. All recordings were performed at pH 6.5. Values (mean ± s.e.m.) and cell numbers are described in Appendix Table S1. Exact $P$ values are summarized in Dataset EV1. Source data are available online for this figure.

screening was performed at an acidic pH (pH 6.5) to maximize chances of observing photomodulation.

Among the 23 labeling positions tested, significant (>30%) photomodulation was observed for 8 of them (Fig. 1D). Most of these positions were clustered on the GluN2B α5 helix, which occupies a central position at the dimer interface between GluN1 and GluN2B NTDs in their active state (Tajima et al, 2016) (Fig. 1C,D). Interestingly, at 3 positions located at the 'top' (N-terminal end) of the α5 helix, currents under 365 nm light were smaller than under 490 nm light (Fig. 1C,D, red squares), while at downstream positions the opposite was observed (currents under 365 nm light stronger than under 490 nm light; Fig. 1C,D, green squares). The strongest photomodulation ratios were obtained for positions GluN2B-Q180C and -R187C, with 2.5-fold UV-induced current inhibition ($I_{365\,nm}/I_{490\,nm} = 0.39 \pm 0.02$, $n = 27$) and close to fourfold UV-induced current potentiation ($I_{365\,nm}/I_{490\,nm} = 3.5 \pm 0.2$, $n = 32$) (Fig. 1D,E; Appendix Table S1), respectively. As observed with other azobenzene-based photoswitchable ligands (Banghart et al, 2004; Berlin et al, 2016; Durand-de Cuttoli et al, 2018; Levitz et al, 2013; Lin et al, 2015), the photomodulation showed high reversibility and reproducibility, allowing multiple cycles of illumination without fatigability (Fig. 1E). These experiments also confirm that *cis*-MASp is kinetically stable, since no current relaxation was observed upon interruption of the UV illumination (see end of illumination cycle in Fig. 1E). The opposite light effects observed at the Q180C and R187C positions are due to opposing effects of MASp *cis* and *trans* configurations. Indeed, at the Q180C position, MASp acts as a *trans*-on photoswitch: it constitutively occupies the spermine binding-site in its *trans* form (under 490 nm light, as revealed by the ~threefold Po increase after MASp labeling), hence potentiating receptor activity (Figs. 1E and EV2A). Conversion by UV light to its *cis* isomer withdraws the spermine moiety from its potentiating binding site, allowing the receptor to revert to its basal state (similar Po as the non-labeled receptor). In contrast, MASp conjugated to GluN2B-R187C acts as a *cis*-on photoswitch: there is no effect on receptor channel Po under 490 nm light, while Po is increased by ~3.5-fold under UV-light (*cis* configuration, Figs. 1E and EV2B). We thus have on-hand two complementary optopharmacological tools for remote and

reversible photocontrol of GluN2B-NMDARs with either UV (R187C) or green light (Q180C) illumination.

We then assessed the extent of MASp photomodulation of GluN1/GluN2B*-Q180C and GluN1/GluN2B*-R187C receptors at physiological pH (pH 7.3). Similarly to what was observed for spermine potentiation of GluN2B-NMDARs (Mony et al, 2011; Traynelis et al, 1995), photomodulation at pH 7.3 was lower than at acidic pH (Fig. EV2C,D). The UV-induced current inhibition of MASp-labeled GluN1/GluN2B*-Q180C receptors dropped from ~2.5-fold at pH 6.5 to ~1.2-fold at pH 7.3 ($I_{365\,nm}/I_{490\,nm} = 0.84 \pm 0.08$, $n = 13$; Fig. EV2C,D; Appendix Table S1). Photomodulation at pH 7.3 was also lower for MASp-labeled GluN1/GluN2B*-R187C receptors, although it remained robust, with ~twofold photomodulation of receptor activity ($I_{365\,nm}/I_{490\,nm} = 1.97 \pm 0.17$, $n = 21$; Fig. EV2C,D). Finally, we evaluated the functional impact of the GluN2B-Q180C and -R187C mutations (together with the C395S background mutation) on receptor activity, as well as on receptor sensitivity to agonists (glutamate and glycine) and endogenous allosteric modulators spermine and protons (Zhu and Paoletti, 2015; Hansen et al, 2018). The Q180C mutation itself (- MASp condition in Fig. EV2E–I; Appendix Table S2) had no significant effect on any of the above parameters. MASp binding to Q180C in its inactive (*cis*) form (+ MASp, UV condition) did not either induce any significant change in receptor function, except a decrease in spermine sensitivity, likely because MASp partially occupies (or hinders) the polyamine site. Functional effects of the R187C mutation and its subsequent labeling with MASp were also mild, although more marked than for Q180C. The R187C mutation by itself induced modest effects on receptor activity (Fig. EV2E–I; Appendix Table S2), while conjugation of MASp in its inactive (*trans*) form (490 nm PSS, Fig. EV2E–I) did not further perturb receptor function. It even restored spermine and proton sensitivities to WT levels (Fig. EV2H,I; Appendix Table S2). Overall, it appears that R187C combines many advantages, including robust *cis*-on photomodulation at physiological pH, making this site particularly attractive for implementation in native tissues (Fig. EV2J; Appendix Table S2). Therefore, we focused on receptors containing the GluN2B*-R187C subunit conjugated with MASp (Opto2B tool).

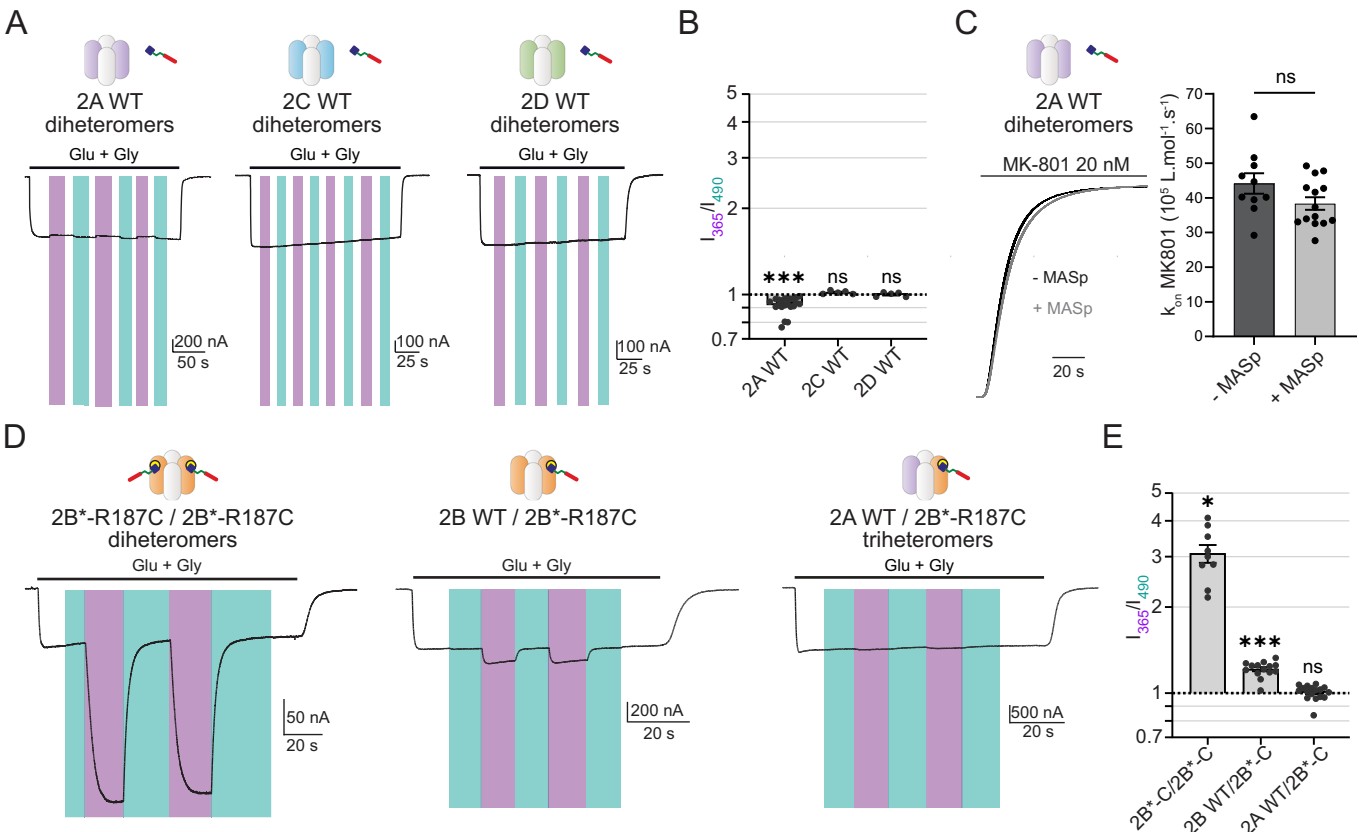

**Figure 2. Selective photomodulation of GluN2B diheteromers.**

(A–C) MASp labeling induces no or little photodependent effect on the function of GluN2A, GluN2C and GluN2D diheteromers. (A) Current trace from a MASp-labeled Xenopus oocyte expressing GluN1/GluN2A, GluN1/GluN2C or GluN1/GluN2D NMDARs upon perfusion of agonists glutamate (Glu) and glycine (Gly) (100 μM each) and submitted to illumination with 365 nm (violet bars) or 490 nm light (blue-green bars). (B) Summary of photomodulation ratios ($I_{365\,nm}/I_{490\,nm}$). n.s., $P > 0.05$; ***$P < 0.001$; one sample Wilcoxon tests against a value of 1, $P$ values were adjusted for multiple comparisons using Bonferroni correction. (C) Left, superposed MK-801 inhibition traces of unlabeled (− MASp, black) and labeled (+ MASp, grey) WT GluN1/GluN2A NMDARs kept in the dark (MASp in its *trans* state). Right, MK-801 inhibition rates of unlabeled and labeled WT GluN1/GluN2A NMDARs. ns, $P > 0.05$, Mann–Whitney test. (D) Current traces from Xenopus oocytes expressing NMDARs with defined subunit stoichiometry using the retention signal strategy described in ref (Stroebel et al, 2014), and labeled with MASp, following perfusion of agonists (glutamate and glycine, 100 μM each) under 365 or 490 nm illumination: GluN1/GluN2B*-R187C-r1/GluN2B*-R187C-r2 diheteromer (left), GluN1/GluN2Bwt-r1/GluN2B*-R187C-r2 (middle) and GluN1/GluN2Awt-r1/GluN2B*-R187C-r2 (right) triheteromers. (E) Summary of the photomodulation ratios ($I_{365}/I_{490}$) for the different constructs. n.s., $P > 0.05$; *$P < 0.05$; ***$P < 0.001$; one sample Wilcoxon tests against the value 1, $P$ values were adjusted for multiple comparisons using Bonferroni correction. All recordings were performed at pH 6.5. Photomodulation values (mean ± s.e.m.) and cell numbers are summarized in Appendix Table S1. Average basal current values ($I_{490\,nm}$) are indicated in Appendix Table S3. Exact $P$ values are summarized in Dataset EV1. Source data are available online for this figure.

## Selective photomodulation of GluN2B diheteromers

In vivo, NMDARs exist as multiple subtypes that differentially populate brain regions and cell types (Paoletti et al, 2013). To determine the MASp selectivity towards the different NMDAR subtypes, we first verified the effect of MASp labeling on diheteromeric GluN1/GluN2A receptors, an abundant pool of NMDARs in the adult brain. Currents from Xenopus oocytes expressing MASp-treated WT GluN1/GluN2A receptors displayed only modest photomodulation, with a small (8%) UV-induced inhibition ($I_{365\,nm}/I_{490\,nm} = 0.92 \pm 0.02$, $n = 22$; Fig. 2A,B; Appendix Table S1). Moreover, MASp labeling by itself had no significant effect on GluN1/GluN2A receptor channel Po (Fig. 2C). Similarly, MASp-treated GluN1/GluN2C and GluN1/GluN2D receptors displayed no photomodulation ($1.02 \pm 0.006$, $n = 5$; and $1.00 \pm 0.008$, $n = 5$, respectively; Fig. 2A,B; Appendix Table S1). This is consistent with the fact that the spermine potentiating site

at the NTD level is present on GluN2B-, but not GluN2A-, GluN2C- and GluN2D-NMDARs (Mony et al, 2011; Traynelis et al, 1995; Williams et al, 1994).

Since GluN2A/GluN2B triheteromers are thought to form a sizeable proportion of NMDAR subtypes in the adult forebrain (Stroebel et al, 2018), we characterized the photosensitivity of GluN2A/GluN2B*-R187C triheteromers labeled with MASp. For that purpose, we used a previously published approach based on the endoplasmic-reticulum (ER) retention signals (hereby named r1 and r2) of GABA$_B$ receptors fused to the GluN2 subunits (Stroebel et al, 2014; see also Hansen et al, 2014). NMDAR complexes containing either two GluN2-r1 or two GluN2-r2 subunits are retained in the ER and cannot reach the cell surface. On the contrary, association of r1 with r2 masks the ER retention signals allowing selective cell surface expression of tri-heteromeric NMDARs containing one GluN2-r1 and one GluN2-r2 subunit (Stroebel et al, 2014; Hansen et al, 2014). No

significant photomodulation of MASp-labeled GluN2A/GluN2B*-R187C triheteromers was observed (GluN2A-r1/GluN2B*-R187C-r2; $I_{365 \text{ nm}}/I_{490 \text{ nm}} = 1.01 \pm 0.01$, $n = 17$; Fig. 2D,E; Appendix Table S1). This lack of photosensitivity could not be attributed to the insertion of the retention signals since MASp-tethered GluN2B*-R187C diheteromers expressed using the same retention system displayed strong photomodulation similar to their control counterparts (GluN2B*-R187C-r1/GluN2B*-R187C-r2; $I_{365 \text{ nm}}/I_{490 \text{ nm}} = 3.07 \pm 0.22$, $n = 9$; Fig. 2D,E). Finally, GluN2B-NMDARs containing only one copy of the mutant GluN2B*-R187C subunit displayed an intermediate level of photomodulation (GluN2B-r1/GluN2B*-R187C-r2; $I_{365 \text{ nm}}/I_{490 \text{ nm}} = 1.21 \pm 0.02$, $n = 14$; Fig. 2D,E). MASp therefore discriminates according to subunit copy number. A single GluN2B subunit is insufficient to endow light sensitivity while strong photocontrol is gained when two photosensitive copies of GluN2B are assembled in the receptor, thus allowing selective modulation of GluN2B diheteromers. In that regard, Opto2B is unique, outperforming currently available subtype-specific pharmacological tools, such as the GluN2B-specific inhibitor ifenprodil, or the GluN2A-specific inhibitor zinc, which all poorly distinguish between diheteromers and triheteromers (Hatton and Paoletti, 2005; Stroebel et al, 2018, 2014; Hansen et al, 2014).

## Fast on-demand photocontrol of GluN2B diheteromers in mammalian cells

We investigated MASp-induced photomodulation in mammalian cells. Similarly to *Xenopus* oocytes, MASp-labeled WT GluN2B diheteromers expressed in HEK cells were not photosensitive and WT GluN2A diheteromers displayed only minor UV-induced inhibition (Appendix Fig. S1A,B). GluN2B*-Q180C diheteromers displayed a small photo-inhibition ($I_{365 \text{ nm}}/I_{490 \text{ nm}} = 0.77 \pm 0.07$, $n = 4$; Appendix Fig. S1A,B) similar to the one observed in *Xenopus* oocytes at physiological pH (see above). In contrast, currents from MASp-labeled GluN2B*-R187C diheteromers expressed in HEK cells displayed strong and fully reversible UV-induced potentiation at physiological pH ($I_{365 \text{ nm}}/I_{525 \text{ nm}} = 3.2 \pm 0.3$, $n = 18$, pH 7.3; Appendix Fig. S1A,B), thus qualitatively mirroring the results obtained in oocytes. Quantitatively, however, the extent of photomodulation (~threefold) was significantly higher than that observed in oocytes at physiological pH (~twofold). This difference presumably stems from the spherical nature and opacity of oocytes, while HEK293 cells are transparent and flatter, allowing more efficient and broader photoswitching of expressed GluN2B*-R187C diheteromers. Current potentiation by UV illumination was similar whether light was applied during agonist application (active state) or before agonist application (resting state) (Appendix Fig. S1C,D). UV illumination by itself had no (or little, see below) effect on the baseline current (recorded in the absence of agonist), as expected from MASp acting as a genuine allosteric ligand, modulating but not directly activating the receptors. Finally, the kinetics of GluN2B*-R187C photoswitching were fast, with time constants of ~60 ms for UV-induced potentiation and of 13 to 450 ms for return to the basal state depending on the illumination wavelength (Appendix Fig. S1F,G and Appendix Text S2).

We finally tested MASp-induced photomodulation in cultured mouse cortical neurons transfected with the GluN2B-R187C* subunit. NMDAR currents from non-fluorescent neurons, which do not express the mutant GluN2B*-R187C subunit, displayed minimal photomodulation ($I_{365 \text{ nm}}/I_{525 \text{ nm}} = 0.968 \pm 0.009$, $n = 9$; Appendix Fig. S2A,B and Appendix Table S1), consistent with minimal effects of MASp-labeling on native, WT NMDARs. In contrast, UV illumination robustly potentiated currents from GFP-positive neurons expressing the GluN2B*-R187C subunit and labeled with MASp ($I_{365 \text{ nm}}/I_{525 \text{ nm}} = 1.81 \pm 0.09$, $n = 19$; Appendix Fig. S2A,B and Appendix Table S1). The lower photomodulation ratio observed in neurons compared to HEK cells is likely due to the presence of endogenous GluN2A and GluN2B-NMDARs, which contribute to the NMDAR current but are photo-insensitive. Overall, these experiments demonstrate that the Opto2B tool is transposable to cultured neurons with intact potentiality.

## Selective photomodulation of synaptic and NMDA tonic currents in brain slices

Motivated by the successful implementation of Opto2B in cultured neurons, we next moved to more native preparations using acute brain slices. As a proof-of-concept, we used cortex-directed *in utero* electroporation of E15.5 mouse embryos to allow the sparse and specific modification of layer II/III cortical pyramidal neurons from the somatosensory cortex in their intact environment (Fig. 3A). This technique furthermore allows monitoring fluorescent (i.e. electroporated) neurons and non-fluorescent (i.e. control) neurons, in the same preparation (Fossati et al, 2019; Meyer-Dilhet and Courchet, 2020). Similarly to cultured cortical neurons, we restricted the electroporation to the GluN2B*-R187C subunit, not including the GluN1 subunit, to limit NMDAR overexpression. Acute slices were incubated during 30 min in 6.6 μM MASp, and then washed before recording (Fig. 3A and see "Methods"). Synaptic currents were induced by stimulation near the apical dendrite of the patched cortical pyramidal neuron (Fig. 3A).

In fluorescent neurons expressing the GluN2B*-R187C subunit and labeled with MASp, UV light induced an increase in NMDA-EPSC amplitude of up to 1.45-fold (in P11-14 mice) compared to EPSCs recorded under green light (Fig. 3B,C; Appendix Table S3, and see "Methods" and Fig. EV3A,B for the protocol of light stimulation). As in recombinant systems, this photomodulation was reversible and reproducible, with no sign of photo-fatigue for at least two cycles of UV-green light stimulations (Figs. 3B and EV3B). No photomodulation of NMDA-EPSCs was observed in control, non-fluorescent neurons (Figs. 3B and EV3B; Appendix Table S3). We also detected a significant photomodulation in fluorescent cells from P21-P25 mice (Fig. EV3C; Appendix Table S3). We found that MASp labeling by itself had minimal impact on synaptic transmission, as indexed by the absence of modification of the NMDA/AMPA ratio (Fig. EV3C) and the lack of photomodulation of AMPA-EPSCs (Fig. EV3E–G). It furthermore had minimal impact on the intrinsic neuronal electrical properties and excitability (Fig. EV3H–J). Taken together, these results show that MASp is operational in intact neuronal networks allowing selective manipulation and detection of GluN2B diheteromers with minimal off-target effects.

During the course of our recordings, we systematically observed in MASp-treated slices a large (up to several hundreds of pA) and transient increase of the tonic (i.e. holding) current at the onset of the UV illumination pulse (Fig. 3D). UV-induced increase in tonic current occurred in the ~15 ms time-range (Appendix Fig. S3A,B)

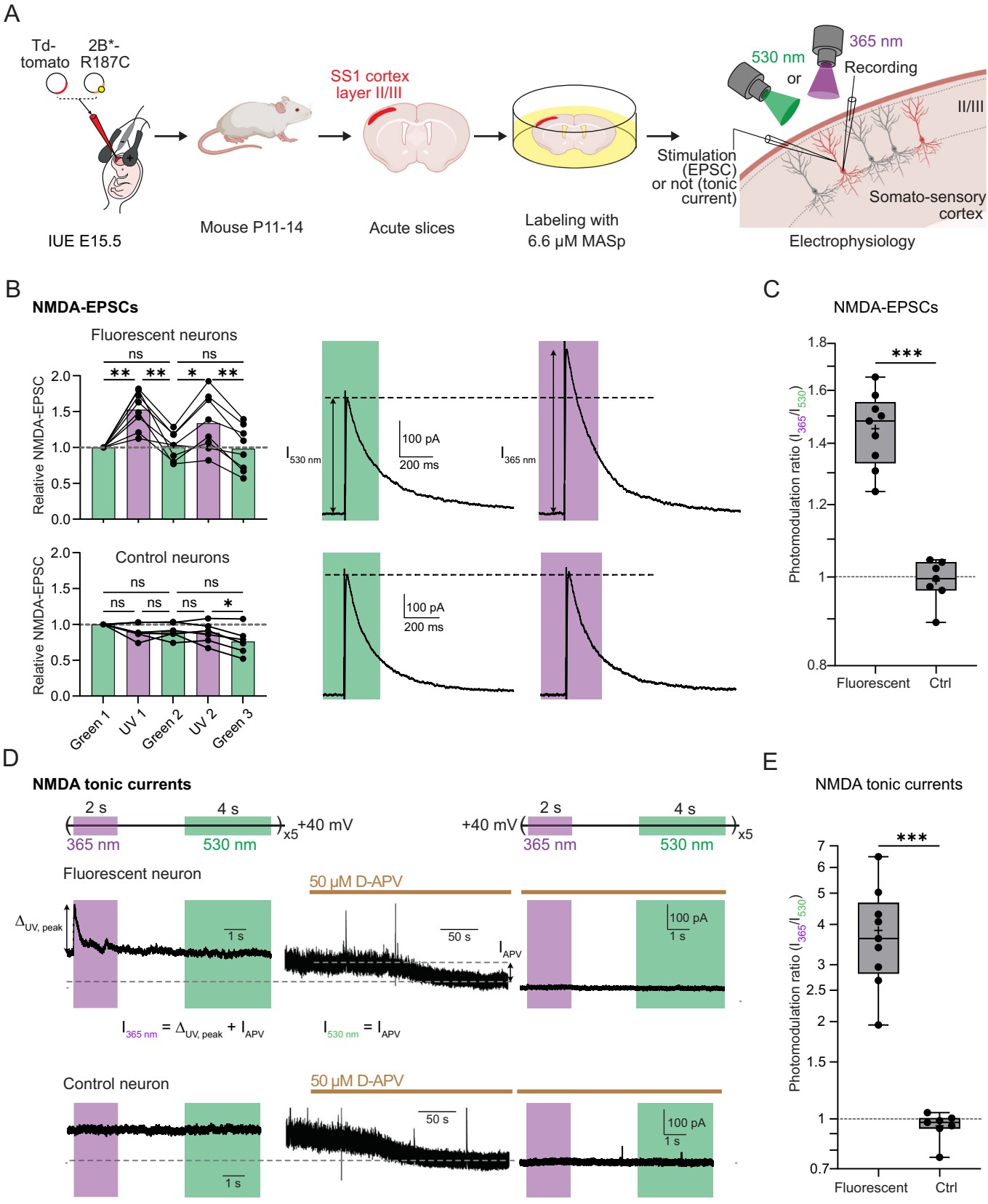

**Figure 3. Photocontrol of NMDA synaptic and tonic currents in brain slices.**

(A) Experimental workflow for in vivo expression of GluN2B*-R187C and ex vivo photomodulation of currents from GluN2B diheteromers. See Main Text and "Methods" for more details. (B, C) Photocontrol of NMDA-EPSCs. (B) Left, NMDA-EPSC amplitudes of MASp-labeled, fluorescent (i.e., electroporated, top) and non-fluorescent (control, bottom) neurons from P11-14 mice during alternating cycles of 530 nm (green bars) and 365 nm light (violet bars), normalized to the amplitude of the first set of EPSCs under green light. Each dot represents the average amplitude of 3 NMDA-EPSCs for each illumination cycle for each cell (see Fig. EV3B). n.s., $P > 0.05$; $*P < 0.05$; $**P < 0.01$, Repeated-measures ANOVA followed by Tukey's multiple comparison test. Right, representative NMDA-EPSCs from a fluorescent (top) and a control (non-fluorescent, bottom) neuron. These traces are the average EPSC traces over three bouts of illumination (9 EPSCs in total) for the 530 nm light condition, and over two bouts of illumination (6 EPSCs in total) for the 365 nm light condition (see Fig. EV3B and "Methods" for the photostimulation protocol). (C) NMDA-EPSC photomodulation ratios (calculated as the average NMDA-EPSC amplitudes in 365 nm light over the average NMDA-EPSC amplitudes in 530 nm light) of MASp-labeled, fluorescent and control cortical neurons from P11-14 mice. $***P < 0.001$, Mann–Whitney test. (D, E) Photocontrol of NMDA tonic currents. (D) Top, protocol of tonic current photomodulation (no electrical stimulation). Bottom, tonic current traces of MASp-labeled, fluorescent and control (non-fluorescent) neurons from a P13 animal before (left) and during (middle and right) application of 50 µM D-APV. Left and right traces are the average of 5 traces. Arrows indicate the currents measured to calculate the photomodulation ratio. (E) Photomodulation ratios (calculated as $I_{365\,nm}/I_{530\,nm}$) of NMDA tonic currents of MASp-labeled, fluorescent and control neurons from P11–P14 animals. $***P < 0.001$, Mann–Whitney test. All recordings in brain slices were performed at physiological pH. Box plots: centerlines show the median; crosses show the mean; box limits indicate the 25th and 75th percentiles; whiskers extend to the minimum and maximum. Photomodulation values (mean ± s.e.m.) and cell numbers are summarized in Appendix Table S6. Exact $P$ values are displayed in Dataset EV1. Source data are available online for this figure.

---

but the current relaxed close to basal level with a time constant of ~200 ms despite the continuous presence of UV light (no green light application) (Fig. 3D; Appendix Fig. S3A,B). However, illumination with green light was necessary to allow subsequent UV-induced potentiation of the tonic current and, similarly to synaptic currents, this process was reproducible over several cycles of illumination without visible photo-fatigue (Appendix Fig. S3C). This current peak was absent in non-fluorescent neurons (Fig. 3D,E; Appendix Fig. S3C,D) and was fully abolished by application of 50 µM of the NMDAR-specific antagonist APV (Fig. 3D; Appendix Fig. S3E). Thus, we can safely conclude that this transient UV-induced current is mediated by light-sensitive GluN2B diheteromers. In principal neurons of the forebrain, extrasynaptic NMDARs contribute to tonic currents by binding ambient glutamate present at low concentrations in the extracellular space (Le Meur et al, 2007; Papouin et al, 2012; Papouin and Oliet, 2014; Sah et al, 1989; Wu et al, 2012). Monitoring the large and transient potentiation of NMDA tonic currents should therefore allow us to probe with high sensitivity the presence of GluN2B*-R187C diheteromers at extrasynaptic sites.

We hypothesized that the transient nature of the UV-induced potentiation of tonic NMDA currents reflected glutamate dissociation from potentiated receptors. Spermine potentiation of GluN2B diheteromers (but also UV potentiation by MASp, Fig. EV2E; Appendix Table S2) is known to come with a slight decrease in the receptor's glutamate sensitivity (Williams, 1994). Recordings from cultured cortical neurons transfected with GluN2B*-R187C confirmed that MASp photo-potentiation decayed with time when sub-saturating glutamate concentrations were used, while under saturating agonists, UV light yielded sustained, non-desensitizing potentiation (Appendix Fig. S4A,B). Thus, in brain slices, the low (i.e. non-saturating) glutamate concentrations likely account for the transient nature of the UV potentiation of tonic NMDA currents. To confirm this hypothesis, we artificially raised ambient glutamate extracellular concentration in brain slices by perfusing the glutamate transporter inhibitor DL-threo-β-benzyloxyaspartic acid (TBOA). As expected, application of TBOA resulted in a strong increase of tonic current levels (average ± s.e.m. tonic current at +40 mV of 438 ± 101 pA before TBOA vs 1469 ± 108 pA after TBOA treatment, $n = 8$, $P$ value = 0.008, Wilcoxon matched-pairs signed rank test). Upon subsequent UV-light application, the UV-induced peak did not fully decay, reaching a steady-state current

significantly larger than basal levels, and that was reversed by green light ($\Delta_{UV,\ SS}/\Delta_{UV,\ peak} = 0.15 \pm 0.02$ before TBOA and $0.46 \pm 0.08$ after TBOA; $n = 8$; Appendix Fig. S3F,G). In some cases, the UV potentiation was sustained with minimal decay of the tonic NMDA currents, indicating that glutamate concentrations following TBOA treatment were high enough to saturate the receptors' glutamate binding sites (Appendix Fig. S3F). By comparing the ratio of steady-state over peak of the UV-induced current increase in absence of TBOA ($\Delta_{UV,\ SS}/\Delta_{UV,\ peak} = 0.15 \pm 0.02$, Appendix Fig. S3G) to the values measured in dissociated neurons (Appendix Fig. S4B), we estimated the basal glutamate concentration to be in the 30 to 100 nM range around cortical pyramidal neurons, consistent with the literature (Kalivas, 2011; Herman and Jahr, 2007; Moldavski et al, 2020). Given the low occupancy of extrasynaptic NMDARs by tonic glutamate at these concentrations and the large peak current elicited by UV illumination, our findings indicate that extrasynaptic sites express a high amount of GluN2B diheteromers. Hence, the very fast control of GluN2B diheteromers by light allows separation of the two opposite effects of polyamine modulation: increase of channel Po and decrease of glutamate potency (Mony et al, 2011; Williams, 1994). As MASp is converted from trans to cis by UV light, it reaches its NTD modulatory site leading to an increase of Po in a ~15 ms time range, which corresponds to the peak of UV-induced increase in tonic current. Subsequent UV peak relaxation reflects glutamate dissociation, with much slower kinetics (~200 ms, see above) that are similar to the NMDA EPSC decay kinetics measured from fluorescent neurons (weighted $\tau_{off} = 198 \pm 13$ ms, $n = 8$). In addition, the amount of relaxation makes it possible to estimate the local tonic glutamate concentration around the recorded neuron.

Unlike steady-state UV potentiation, peak UV potentiation as assessed on recombinant receptors was largely independent on glutamate concentration, with <10% variation of photomodulation between 0.03 and 100 µM of glutamate (Appendix Fig. S4C–E and Appendix Text S3). Measurement of UV-induced potentiation at the peak therefore allows comparison of the proportions of GluN2B diheteromers between neurons and neuronal compartments regardless of their local glutamate environment. Varying concentrations of the co-agonist glycine also had only small effects on the extent of photomodulation measured at the UV peak (Appendix Fig. S3F–H and Appendix Text S3). For each individual cell in brain slices, we thus estimated the level of tonic NMDAR current

photo-enhancement by measuring the UV-induced peak current and comparing it to the amplitude of basal tonic current mediated by NMDARs deduced from APV inhibition under green light ($I_{365\ nm}/I_{530\ nm}$ on Fig. 3D; and see "Methods"). Using this approach, we found that UV light induced massive potentiation of tonic NMDARs, >3.5-fold in slices from juvenile (P11-14) mice ($I_{365\ nm}/I_{530\ nm} = 3.8 \pm 1.3$, $n = 9$; Fig. 3E; Appendix Table S3, and see also Appendix Fig. S3H for older ages). These results indicate a strong presence of GluN2B diheteromers at extrasynaptic compartments at young postnatal ages, confirming previous work (Papouin and Oliet, 2014; Parsons and Raymond, 2014). They also show the potential of our tool to precisely estimate the amount of GluN2B diheteromers at synaptic and extrasynaptic sites relatively to existing pharmacological agents such as ifenprodil or polyamines, whose effects on tonic currents are difficult to interpret due to the glutamate-dependence of their mode of action (Kew et al, 1996; Williams, 1994).

## Developmental regulation of endogenous synaptic and extrasynaptic NMDAR subtypes in the hippocampus

With its ability to probe the proportion of GluN2B diheteromers at synaptic and extrasynaptic sites, Opto2B appears well suited to dissect the molecular composition of NMDARs across development and subcellular compartments. To avoid overexpression and maintain endogenous levels and patterns of GluN2B-NMDAR expression, we generated a knock-in (KI) mouse containing a mutant GluN2B subunit allowing for MASp attachment and photomodulation (Opto2B mouse). While in Xenopus oocytes (see above and Fig. EV2) and HEK cells (Appendix Fig. S5A), neutralizing the endogenous cysteine GluN2B-C395 was required to avoid background effects of MASp, no such undesired effects were observed in cultured cortical neurons (similar large photomodulation between GluN2B*-R187C and GluN2B-R187C receptors, Appendix Fig. S5B and Appendix Table S1). This suggests that, in neurons, MASp does not react with GluN2B-C395. Accordingly, to limit the number of introduced mutations, we generated a KI mouse harboring the single GluN2B-R187C mutation (Opto2B mouse, Fig. 4A). KI mice were viable and reproduced. Western blot analysis revealed no significant difference of expression of GluN1, GluN2A, GluN2B, and AMPAR GluA1 subunits between Opto2B and WT animals (Fig. EV4A–D). In addition, using kinetics measurements and classical pharmacology, we verified that the basal GluN2A/GluN2B subunit ratio at hippocampal CA3-CA1 synapses (synapses of interest, see below) was unaffected in MASp-labeled Opto2B mice compared to unlabeled WT animals (Fig. EV4E–J), indicating no significant alteration of NMDA synaptic currents and subunit content by the GluN2B-R187C mutation and MASp labeling step.

We focused on hippocampal CA1 neurons and CA3-CA1 synapses, which have been extensively studied for their NMDAR receptor content. Adult CA1 neurons express high levels of GluN2A and GluN2B subunits, but little or no GluN2C and GluN2D subunits (Akazawa et al, 1994; Monyer et al, 1994; Watanabe et al, 1993). CA1 neurons also display the typical GluN2B to GuN2A 'switch' whereby, following birth, expression of the GluN2A subunit gradually increases, leading to increasing incorporation of GluN2A-containing NMDARs and a proportional decrease in GluN2B-containing receptors (Hansen et al,

2021; Paoletti et al, 2013; Sanz-Clemente et al, 2013). However, in which proportions GluN2A and GluN2B subunits associate to form GluN2A and GluN2B diheteromers, or GluN2A/2B triheteromers, and how these subtypes segregate between synaptic and extrasynaptic sites remains elusive and subject to controversy (Papouin and Oliet, 2014; Parsons and Raymond, 2014; Hardingham and Bading, 2010; Köhr, 2006; Gladding and Raymond, 2011).

To study synaptic NMDARs through development, we prepared acute brain slices from Opto2B mice at different age ranges (P5, P8-P12, P20-P23 or P37-P47), which were labeled with 6.6 µM MASp. NMDA synaptic currents (NMDA-EPSCs) were recorded on patched CA1 pyramidal neurons by stimulating the Schaeffer collaterals, either under UV light, or under green light (Fig. 4B, similar protocol as in Fig. EV3B). UV light induced ~1.4-fold potentiation of NMDA-EPSC peak currents compared to green light at young ages (P5-P12), while no effect of light was observed on MASp-labeled slices from WT animals (at P8-P12, $I_{365\ nm}/I_{530\ nm} = 1.41 \pm 0.06$, $n = 20$ for Opto2B mice vs $1.02 \pm 0.04$, $n = 10$ for WT; Fig. 4C,D; Appendix Table S4). The photomodulation in Opto2B animals decreased progressively with development to reach 1.14-fold around P40, indicative of a progressive decrease in the GluN2B diheteromer contribution to synaptic currents during development (Fig. 4C, left; Fig. 4D, dark-blue bars; and Appendix Table S4). To gain insights into the origin of this drop in photomodulation, we crossed the Opto2B mouse line with a GluN2A-KO mouse line in order to prevent expression of any GluN2A-NMDARs (including GluN2A diheteromers and GluN2A/GluN2B triheteromers) (Sakimura et al, 1995). Compared to Opto2B slices, photomodulation of NMDA-EPSCs was much stronger in Opto2B/GluN2A-KO slices at all tested age ranges (P8 to P47; Fig. 4C,D; Appendix Table S4). This shows that GluN2B diheteromers already co-exist with GluN2A-NMDARs at synaptic sites at young ages. Contrary to Opto2B animals, the difference of NMDA-EPSC photomodulation between P8-12 and P37-47 in Opto2B/GluN2A-KO mice was not statistically significant ($P = 0.105$ after correction for multiple comparisons; two-sided Mann–Whitney test), indicating that the principal factor for photomodulation decrease in Opto2B mice is the increase of GluN2A-NMDAR contribution to synaptic currents. Our photomodulation results are consistent with the developmental acceleration of NMDA-EPSC decay kinetics in slices from Opto2B mice, an effect absent in Opto2B/GluN2A-KO animals (Fig. EV4E,F). However, despite being not statistically significant, there is a trend towards a decrease of NMDA-EPSC photomodulation with age in Opto2B/GluN2A-KO mice. This suggests that other factors than GluN2A incorporation might also contribute to the developmental decrease of NMDA-EPSC photomodulation in Opto2B mice (see "Discussion").

We next studied the photomodulation of tonic NMDA currents across development on slices from Opto2B and Opto2B/GluN2A-KO animals to estimate the evolution of the proportion of GluN2B diheteromers in extrasynaptic compartments (Figs. 5A,B and EV5A). We first verified that no significant photomodulation of tonic NMDA currents was present on MASp-treated slices from WT animals ($0.94 \pm 0.04$, $n = 4$; Figs. 5B and EV5B). At early postnatal ages (before P12), a large, and quantitatively similar, UV potentiation of NMDA tonic currents (>2.5-fold) was observed for Opto2B mice and Opto2B/GluN2A-KO animals (Fig. 5A,B; Appendix Table S4), indicating a lack of GluN2A-NMDARs. Therefore, at early postnatal ages, extrasynaptic NMDARs appear

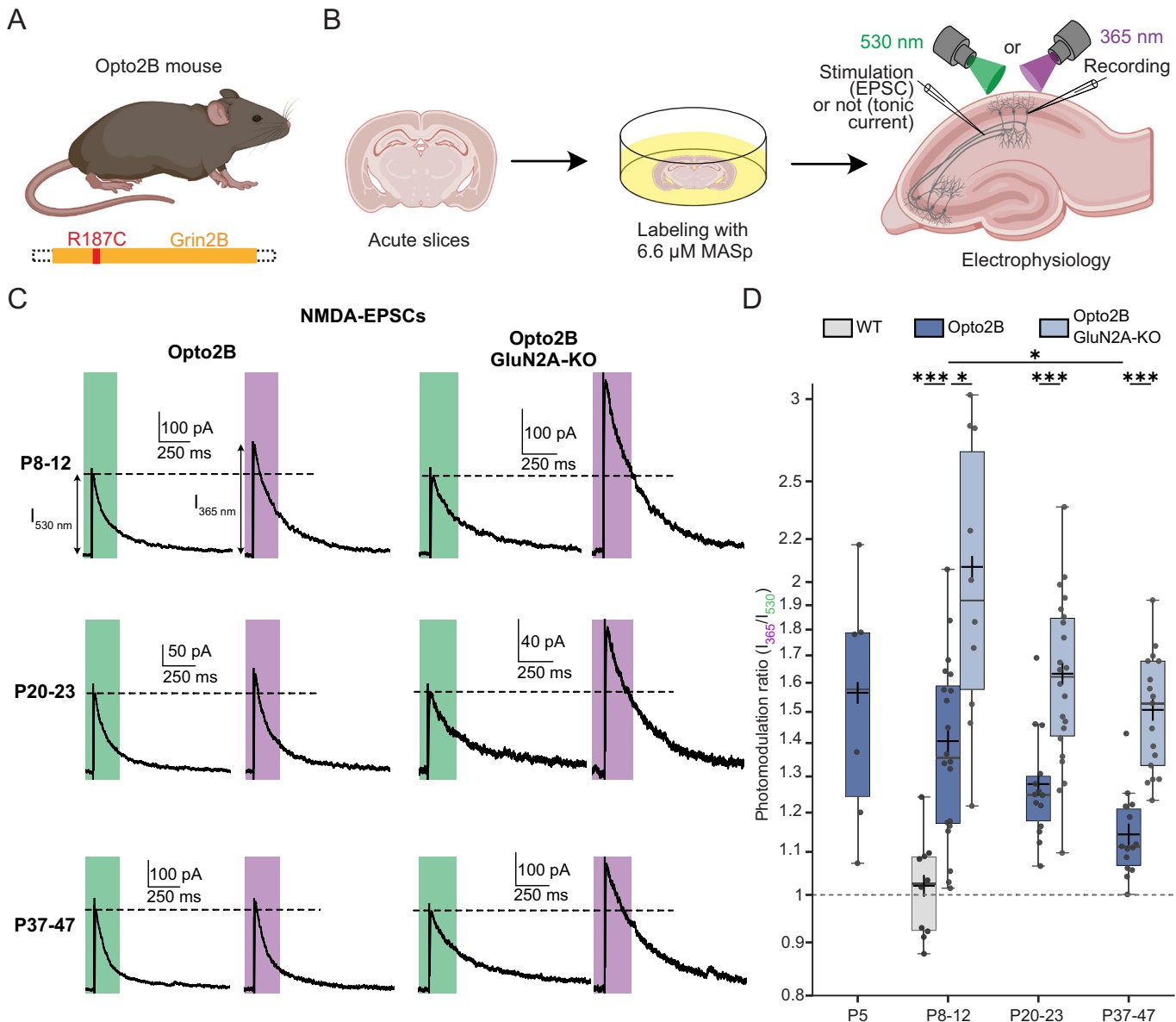

**Figure 4. Optical profiling of NMDAR subtypes at CA3–CA1 synaptic sites.**

(A) Generation of a KI mouse endogenously expressing GluN2B-R187C (Opto2B mouse). (B) Experimental workflow for ex vivo photomodulation of GluN2B diheteromers at the CA3-CA1 synapse of the hippocampus. See Main Text and "Methods" for more details. (C) Representative NMDA-EPSCs of MASp-labeled, CA1 pyramidal neurons from Opto2B (left) or Opto2B/GluN2A-KO (right) mice at different age ranges (P8-P12, top; P20-P23, middle; P37-P47, bottom) under 530 nm (green bars) and 365 nm light (violet bars). These traces are the average EPSC traces over two bouts of illumination (6 EPSCs in total) for the 530 nm light condition and over one bout of illumination (3 EPSCs in total) for the 365 nm light condition, which is sandwiched between the two previous ones (green—UV—green cycle, see Methods). (D) Summary of NMDA-EPSC photomodulation ratios for MASp-labeled, CA1 pyramidal neurons from WT (grey), Opto2B (dark blue) and Opto2B/GluN2A-KO (light blue) mice at different age ranges. All recordings in brain slices were performed at physiological pH. Box plots: centerlines show the median; crosses show the mean; box limits indicate the 25th and 75th percentiles; whiskers extend to the minimum and maximum, excluding outliers. Photomodulation values (mean ± s.e.m.) and cell numbers are summarized in Appendix Table S7. Basal NMDA EPSCs ($I_{530\ nm}$) are indicated in Appendix Table S8. *$P < 0.05$; **$P < 0.01$; ***$P < 0.001$; Opto2B/GluN2A-KO P8-12 vs Opto2B/GluN2A-KO P37-47, $P = 0.105$ (not shown in (D)); multiple Mann–Whitney tests; $P$ values were adjusted for multiple comparisons using Bonferroni correction. Only the indicated comparisons were performed. Exact $P$ values are displayed in Dataset EV1. Source data are available online for this figure.

to be almost entirely GluN2B diheteromers. Interestingly, the maximal photopotentiation observed on tonic NMDAR currents was similar to that observed in vitro in GluN2B-R187C transduced cultured neurons ($I_{365\ nm}/I_{530\ nm} = 3.30 \pm 0.39$, $n = 15$ for P8-P12 Opto2B slices, Fig. 5B and Appendix Table S4, vs $I_{365\ nm}/I_{525\ nm} = 3.04 \pm 0.25$, $n = 20$ for GluN2B-R187C-expressing cultured cortical

neurons, Appendix Fig. S5 and Appendix Table S1), showing high transposability of our tool across experimental conditions. Strikingly, for older Opto2B animals (after P12), the photomodulation drastically dropped (to levels close to 1.5-fold), while it was maintained at high levels for Opto2B/GluN2A-KO animals (Fig. 5A,B; Appendix Table S4). This drop in photomodulation

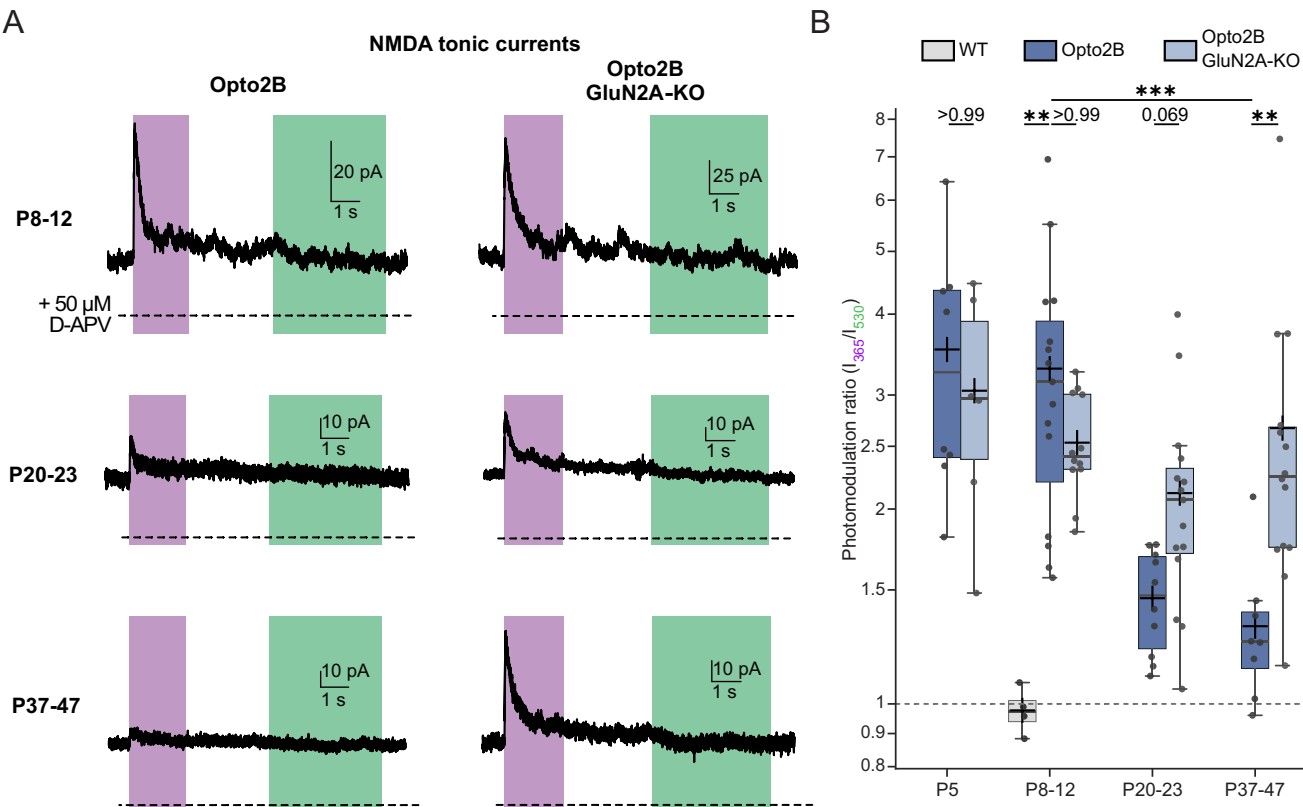

**Figure 5. Optical profiling of NMDAR subtypes at CA1 extrasynaptic sites.**

(A) Representative tonic NMDA current traces of MASp-labeled, CA1 pyramidal neurons from Opto2B (left) or Opto2B/GluN2A-KO (right) mice at different age ranges (P8-P12, top; P20-P23, middle; P37-P47, bottom) under illumination by 365 nm (violet bars) and 530 nm light (green bars). The dotted line shows the level of tonic current under application of 50 μM D-APV (as in Fig. EV5B). Tonic current traces are the averages of 5 to 10 traces. (B) Summary of the photomodulation ratios of NMDA tonic currents of MASp-labeled, CA1 pyramidal neurons from WT (grey), Opto2B (dark blue) and Opto2B GluN2A-KO (light blue) mice at different age ranges. All recordings in brain slices were performed at physiological pH. Box plots: centerlines show the median; crosses show the mean; box limits indicate the 25th and 75th percentiles; whiskers extend to the minimum and maximum excluding outliers. Photomodulation values (mean ± s.e.m.) and cell numbers are summarized in Appendix Table S7. For non-significant P values, P values are indicated directly in the graph. **P < 0.01; ***P < 0.001; multiple Mann–Whitney tests, P values were adjusted for multiple comparisons using Bonferroni correction. Only the indicated comparisons were performed. Exact P values are displayed in Dataset EV1. Source data are available online for this figure.

could not be accounted for a progressive decrease in tonic NMDA currents with age (there was instead an increase of APV-dependent tonic current with age; Fig. EV5C,D). Rather, the developmental drop of photomodulation ratio for Opto2B neurons was mediated by a decrease of the absolute amplitude of UV-induced peak currents (Fig. EV5E). Altogether, these results show that the proportion of GluN2B diheteromers at extrasynaptic sites strongly decreases with age, and that GluN2B diheteromers are progressively replaced by GluN2A-NMDARs.

## Discussion

As increasing evidence reveal that most neurotransmitter receptors assemble as heteromers with multiple subunit stoichiometries, developing tools to selectively control and monitor individual receptor subtypes becomes a growing challenge (Sente et al, 2022; Sun et al, 2023; Yu et al, 2021; Zhao et al, 2019; Belkacemi et al, 2025; DeDominicis et al, 2017; Meng et al, 2022). We have developed an optopharmacological tool (Opto2B) to enhance

selectively the activity of NMDARs containing two copies of the GluN2B subunit (GluN2B diheteromers), while receptors containing a single GluN2B copy (GluN2A/GluN2B triheteromers) are unaffected. Opto2B relies on the covalent reaction of a photoswitchable spermine, MASp, to a cysteine-modified GluN2B subunit, GluN2B-R187C. We show that, when bound to GluN2B-R187C, MASp allows fast, reversible and reproducible photoenhancement of GluN2B diheteromer activity in recombinant systems, cultured neurons and brains slices with minimal off-target effects on basal synaptic transmission. Using this tool, we were able to investigate the evolution of NMDAR subtype composition during postnatal development. We reveal that at hippocampal CA1 pyramidal cells both synaptic and extrasynaptic populations undergo synaptic maturation through a GluN2B-to-GluN2A switch mechanism, but that this maturation occurs at a later time for extrasynaptic populations. At adult stage, our work indicates that GluN2A-containing NMDARs, i.e. GluN2A diheteromers and GluN2A/GluN2B triheteromers, likely form the majority of NMDAR subtypes in hippocampal CA1 neurons at both synaptic and extrasynaptic sites.

## Opto2B, a GluN2B-selective PAM with subunit stoichiometry resolution

To our knowledge, Opto2B is the first tool allowing the discrimination between NMDARs containing distinct stoichiometries of GluN2A and GluN2B subunits, the two main GluN2 subunits in the adult forebrain (Paoletti et al, 2013). Currently available GluN2A-selective NAMs acting at the level of the NTDs (zinc ions) or GluN2A-selective PAMs have intermediate effects on GluN2A/GluN2B triheteromers. In addition, effects of GluN2A-selective NAMs acting at the LBD level (e.g. TCN-201 (Bettini et al, 2010) or MPX-004 (Volkmann et al, 2016)) are almost as large on GluN2A/GluN2B triheteromers as on GluN2A diheteromers (Hackos et al, 2016; Hansen et al, 2014; Stroebel et al, 2014; Yi et al, 2016). On the other hand, ifenprodil derivatives, commonly used GluN2B-selective inhibitors, have an intermediate effect on GluN2A/GluN2B triheteromers (Hansen et al, 2014; Stroebel et al, 2014). Polyamines, the only known GluN2B-selective PAMs, have a large potentiating effect on GluN2B diheteromers (Mony et al, 2011; Williams et al, 1994), with minimal potentiation of GluN2A/GluN2B triheteromers (Cheriyan et al, 2016; Stroebel et al, 2014). However, these compounds also produce a non-selective inhibition of all NMDAR subtypes through a pore block mechanism. They furthermore have multiple non-specific effects on various membrane receptors and channels, which might interfere with the GluN2B-specific potentiating effect and complicate their use in native systems (Mony et al, 2009; Williams, 1997; Yi et al, 2019). In contrast, the Opto2B tool relies on the covalent binding of a polyamine ligand, MASp, near the GluN2B-specific polyamine potentiating site. After labeling and extensive washout of the free compound, only the bound-MASp remains, hence avoiding off-target pore block of NMDARs, as well as off-target binding to other receptors in native tissues. Accordingly, unlike free spermine, treatment of non-GluN2B NMDARs (GluN2A, 2C and 2D diheteromers) with MASp yielded no or very little light-dependent effect on their activity (Fig. 2). Interestingly, covalent binding of MASp to GluN2B-R187C in the context of a GluN2A/2B triheteromer did not either photosensitize the activity of this receptor subtype. This is reminiscent of the poor effect of free spermine on GluN2A/2B triheteromers (see above) and is consistent with the GluN2A subunit dominating the biophysical and pharmacological properties of GluN2A/GluN2B triheteromers (Hansen et al, 2014; Stroebel et al, 2014; Yi et al, 2016).

Despite the exquisite selectivity of Opto2B towards GluN2B diheteromers, MASp can conjugate to the endogenous cysteine GluN2B-C395 in recombinant systems (Xenopus oocytes and HEK cells), leading to an increase in channel Po but no light sensitivity. This increase in basal Po explains the smaller UV potentiation observed when MASp was able to react with both cysteines at positions 187 and 395 (GluN1/GluN2B-R187C receptors; 1.75-fold UV-potentiation in HEK cells) than when C395 reactivity was neutralized, allowing conjugation of MASp only at position R187 (GluN1/GluN2B-R187C-C395S receptors) (3 to 3.7-fold UV-potentiation in HEK cells) (Appendix Figs. S1 and S5A). However, in cultured cortical neurons, whether C395 was neutralized or not had no significant impact on the photomodulation ratio (Appendix Fig. S5B). Moreover, we obtained large photomodulation ratios in brain slices from Opto2B mice (Figs. 4 and 5), close to the ones obtained from cortical brain slices electroporated with the GluN2B-R187C-C395S subunit (Fig. 3). This suggests that background conjugation of MASp to the endogenous cysteine GluN2B-C395 does not occur in native systems. The factors underlying the lack of reactivity of GluN2B-C395 in neurons compared to other (non-neuronal) cells remain to be established.

## Monitoring GluN2B diheteromers with subcellular precision

In addition to its exquisite molecular specificity, the fast and reversible properties of the Opto2B tool allowed us to probe independently, within the same cell, the molecular composition of synaptic and extrasynaptic NMDARs. It is considered that NMDA-EPSCs evoked by single electrical stimulations of presynaptic afferents reflect the activity of the synaptic pool of NMDARs (Papouin and Oliet, 2014). On the other hand, NMDA tonic currents, recorded in absence of any electrical stimulation, are considered to be mediated primarily by extrasynaptic receptors (Le Meur et al, 2007; Papouin and Oliet, 2014; Wu et al, 2012). Hence, by measuring in CA1 pyramidal neurons the amount of UV-induced potentiation of NMDA-EPSCs on one side, and NMDA tonic current on the other, we were able to monitor the proportion of GluN2B diheteromers from the CA3-CA1 synaptic pool and the extrasynaptic pool, respectively.

In many respects, Opto2B outperforms previous methods to investigate the molecular nature of NMDARs. In particular, with its ability to reveal with high sensitivity the presence of extrasynaptic GluN2B diheteromers activated by very low concentrations of ambient glutamate, Opto2B circumvents many limitations of classical pharmacology. Investigation of the molecular nature of extrasynaptic NMDARs has largely relied on classical pharmacology, assessing the extent of tonic NMDAR current inhibition by subunit-specific NMDAR antagonists like zinc or ifenprodil (see, for example, (Papouin et al, 2012)). NMDAR-dependent tonic currents are usually small (30–60 pA) and inhibitors typically take minutes to elicit their effect in brain slices, making the quantification of tonic current inhibitions challenging. In addition, modulation by subunit-specific NMDAR antagonists is usually agonist-dependent, resulting in large variations in the amplitude and direction of modulation depending on agonist concentration. For instance, at low (~100 nM) glutamate concentrations, ifenprodil has a potentiating rather than an inhibitory effect on GluN2B-NMDARs (Kew et al, 1996). Similarly, the extent of zinc inhibition strongly depends on the occupancy of the glutamate sites (Zheng et al, 2001; Paoletti et al, 1997). Thus, the effects of conventional pharmacological agents on tonic NMDA currents, which are activated by ambient glutamate concentrations that likely differ from preparation to preparation (Herman and Jahr, 2007), appear to be unreliable readouts for determining the receptor subunit composition.

The Opto2B tool also harnesses the power of light and allows temporal segregation of GluN2B diheteromer potentiation (occurring in a 15 ms time range) linked to the increase in channel Po, from a 10-time slower receptor "depotentiation", reflecting glutamate dissociation due a decrease in glutamate affinity. The time separation between these two opposite effects yields a clear sharp peak following illumination of the NMDA tonic current. The large amplitude of the current peak (tens of pA) and its independence on glutamate concentration allows reliable estimation of the contribution of GluN2B diheteromers at extrasynaptic

sites and its direct comparison with the contribution of this receptor subtype at synaptic sites. On the other hand, the amount of UV-induced potentiation of NMDA tonic currents at steady-state, which strongly depends on glutamate concentration, allows estimation of glutamate concentrations at the vicinity of the receptors (Appendix Fig. S4). With its unique properties, the Opto2B tool appears well suited to compare NMDAR subunit composition from different cell compartments and to probe the local glutamate microenvironment.

## Opto2B reveals distinct developmental profiles of synaptic and extrasynaptic NMDARs

Decades of research have established that the GluN2B subunit is enriched at embryonic stages while expression of the GluN2A subunit gradually increases following birth, resulting in a GluN2A/GluN2B ratio that increases with age (Hansen et al, 2021; Paoletti et al, 2013; Sanz-Clemente et al, 2013; Papouin et al, 2012; Akazawa et al, 1994; Bellone and Nicoll, 2007; Gray et al, 2011; Ferreira et al, 2017). However, what is still unclear and highly debated is how the GluN2B subunit partitions between GluN2B diheteromers or in GluN2A/GluN2B triheteromers during brain maturation. Early studies based on subunit co-immunoprecipitation concluded that, at adult stage, the GluN2B subunit is either incorporated preferentially in GluN2B diheteromers (Al-Hallaq et al, 2007; Chazot and Stephenson, 1997), or, in contrary, in GluN2A/GluN2B triheteromers (Luo et al, 1997) (see also ref (Stroebel et al, 2018)). At adult CA3-CA1 hippocampal synapses, several studies have indicated that the majority of receptors are GluN2A/GluN2B triheteromers (and thus that GluN2B diheteromers are a minority) (Stroebel et al, 2018; Gray et al, 2011; Rauner and Köhr, 2011; Tovar et al, 2013). Yet, more recently, super-resolution microcopy data on cultured hippocampal neurons revealed a maximum of ~30% of GluN2A and GluN2B subunit colocalization at synaptic sites, suggesting preferential incorporation of GluN2A and GluN2B subunits into diheteromers (Kellermayer et al, 2018). In absence of tools with subunit stoichiometry resolution, functional investigations of NMDAR subunit composition is challenging (see above). The Opto2B tool solves this issue in a large part by allowing direct probing of GluN2B diheteromer content.

We generated a knock-in mouse expressing the mutated GluN2B-R187C subunit to allow the investigation of the developmental regulation of GluN2B diheteromers at synaptic and extrasynaptic sites, while keeping endogenous levels of GluN2B subunit expression. Using Opto2B mice, we observed an increase of NMDA-EPSC amplitude under UV light at all ages, demonstrating that GluN2B diheteromers are present at CA3-CA1 synapses throughout development. However, we also observed a clear decrease of the extent of photomodulation with age reflecting a decreased contribution of synaptic GluN2B diheteromers during development, so that this population becomes a minority at adult stage. Even at early stages, the UV photomodulation remained modest in amplitude (compared to the photomodulation observed on 'pure' GluN2B diheteromers), indicating a scarcity of GluN2B diheteromers. Combining the Opto2B mouse line with the GluN2A-KO mouse line revealed that GluN2A-NMDARs (GluN2A diheteromers and/or GluN2A/GluN2B triheteromers) are already present with GluN2B diheteromers at early postnatal ages (<P12) at the CA3-CA1 synapse. Therefore, following birth, GluN2A-NMDARs readily access synaptic sites (Fig. 6A). For the youngest

age tested (P5), the low photomodulation could additionally be linked to the contribution of residual (spermine-insensitive) GluN2D-containing receptors (von Engelhardt et al, 2015), with the GluN2D subunit being highly expressed at embryonic stages before rapidly declining during the first postnatal week in CA1 pyramidal cells (Akazawa et al, 1994; von Engelhardt et al, 2015). Altogether our results demonstrate a gradual disappearance of GluN2B diheteromers at CA3-CA1 synapses during development in favor of GluN2A-containing NMDARs (either GluN2A diheteromers or GluN2A/GluN2B triheteromers, Fig. 6B), in line with the previously described GluN2B-to-GluN2A developmental switch. They also add the new information that CA3-CA1 synaptic sites do not favor the clustering of GluN2B diheteromers, but rather of GluN2A-containing receptors, even at early postnatal stages when GluN2B expression still dominates. A slight decrease in NMDA-EPSC photomodulation with age was also observed in Opto2B/GluN2A-KO mice, suggesting contributions of other mechanisms than incorporation of GluN2A-NMDARs at synaptic sites. These include a potential progressive incorporation of GluN1 subunits containing the exon 5 splice variant (GluN1-1b), known to decrease polyamine potentiation (Traynelis et al, 1995); or developmental modifications of the nanoscale organization of GluN2B diheteromers at the synapse (Kellermayer et al, 2018), which might affect the extent of EPSC photomodulation by variations in glutamate concentrations sensed by the receptors. Altogether, our findings raise key questions that remain to be addressed about the rules governing NMDAR subunit assembly and their specific targeting to distinct neuronal compartments.

In contrast to synaptic receptors, we observed that GluN2B diheteromers form the vast majority of extrasynaptic NMDAR subtypes in CA1 pyramidal neurons at young postnatal ages (<P12), which is consistent with the literature (Papouin and Oliet, 2014; Parsons and Raymond, 2014). At these juvenile ages, our results on Opto2B/GluN2A-KO mice show that GluN2A-NMDARs do not contribute or contribute minimally to NMDA tonic currents. However, we also observed that the contribution to tonic current of GluN2B diheteromers decreases drastically through development, Opto2B/GluN2A-KO mice revealing a growing contribution of GluN2A-NMDARs at extrasynaptic sites (Fig. 6B). At adult stage, our results show that GluN2B diheteromers only form a minor population of extrasynaptic NMDARs, while GluN2A-NMDARs predominate (Fig. 6A,B). At first glance, this result challenges the commonly admitted view that GluN2A-NMDARs and GluN2B-NMDARs segregate in synaptic and extrasynaptic compartments, respectively (Fellin et al, 2004; Papouin et al, 2012; Scimemi et al, 2004; Tovar and Westbrook, 1999). Several studies, however, do not support such a strict differential repartition, with equivalent GluN2A and GluN2B contents in both compartments (Harris and Pettit, 2007; Le Meur et al, 2007; Mohrmann et al, 2000; Petralia et al, 2010). This discrepancy likely originates from different methodological approaches, various preparations (neuronal culture or brain slices), brain regions and developmental stages of the studied systems (Papouin and Oliet, 2014). The limitations of classical pharmacological agents to study extrasynaptic receptors and their complex effects at low agonist concentrations (see above) add to the lack of consensus. Leveraging the unique attributes of Opto2B, we here establish that, similarly to synaptic NMDARs, extrasynaptic NMDARs also undergo a marked GluN2B-to-GluN2A switch in

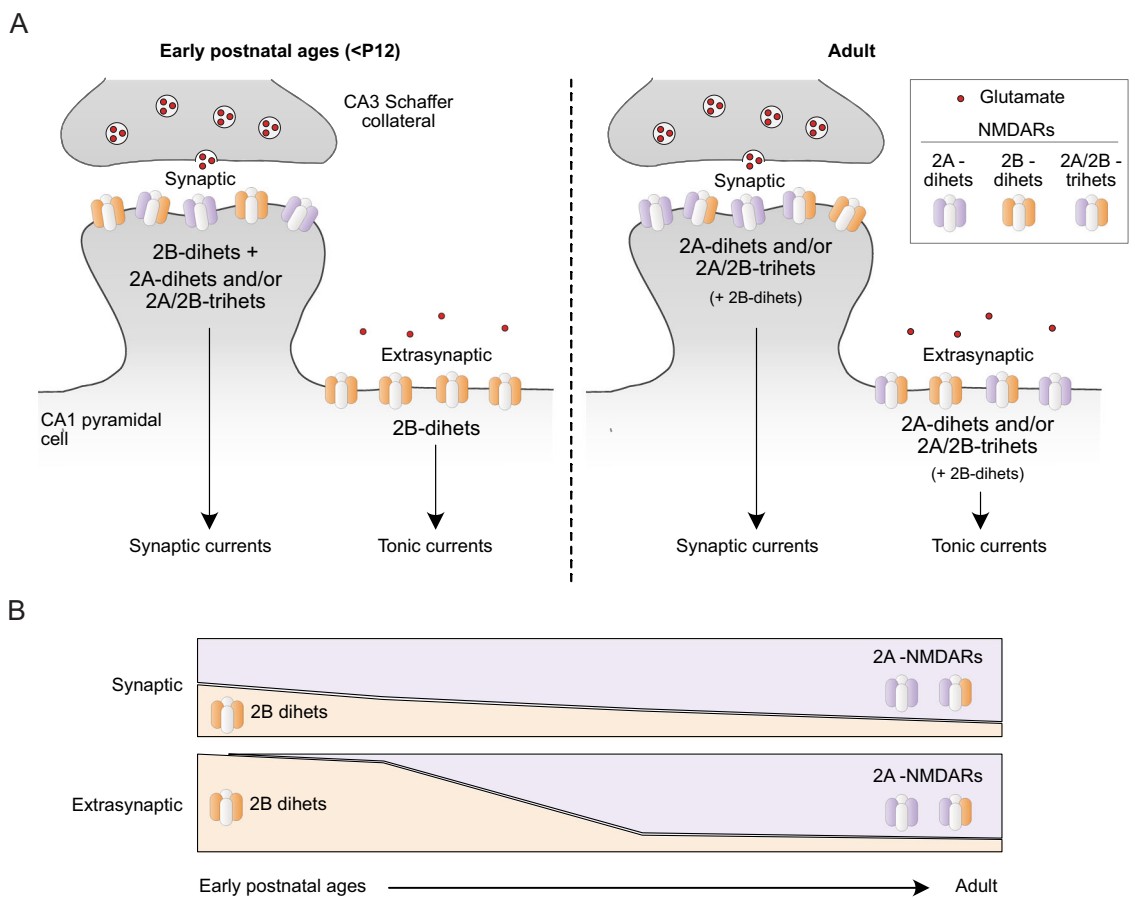

**Figure 6. Model of developmental regulation of synaptic and extrasynaptic NMDARs in CA1 pyramidal cells.**

(A) Left, at early postnatal ages, GluN2B diheteromers are present at the synapse together with GluN2A-NMDARs (GluN2A diheteromers and/or GluN2A/GluN2B triheteromers). On the contrary, extrasynaptic NMDARs are mainly composed of GluN2B diheteromers. Right, in adolescent and adult mice, GluN2A-NMDARs form the major NMDAR populations at both synaptic and extrasynaptic sites. (B) Qualitative model of the evolution of the relative abundance of synaptic (top) and extrasynaptic (bottom) GluN2B diheteromers and GluN2A-NMDARs (GluN2A diheteromers and GluN2A/GluN2B triheteromers) across development.

hippocampal CA1 pyramidal cells. The comparison between synaptic and tonic photomodulations also reveals a previously unknown shift in the developmental profile of NMDAR molecular composition between synaptic and extrasynaptic sites (Figs. 4 and 5): GluN2A is incorporated earlier at the synapse but surprisingly constitutes a major part of all NMDARs in the adult mouse at both sites (see model, Fig. 6). These results challenge the prevailing idea of an enrichment of extrasynaptic receptors in GluN2B diheteromers throughout development. Given the robust expression of GluN2B subunit in the adult forebrain (Akazawa et al, 1994; Monyer et al, 1994; Watanabe et al, 1993), we propose that a substantial pool of synaptic but also of extrasynaptic NMDARs is constituted by GluN2A/GluN2B triheteromers. Establishing the molecular identity of the various pools of NMDARs, each of which presumably triggers distinct signaling pathways (Hansen et al, 2021; Paoletti et al, 2013; Sanz-Clemente et al, 2013), remain of prime importance to untangle the complex effects of subtype-specific NMDAR signaling on brain circuits and behavior. It also provides valuable information for the design and biological understanding of precision pharmaceutics targeting NMDARs (Geoffroy et al, 2022; Hanson et al, 2024).

## Limitations of the study

While we established that GluN2B diheteromers form a minor portion of NMDARs relative to GluN2A-NMDARs in mature CA1 neurons, our study does not allow to determine in which proportions these GluN2A-NMDARs distribute between GluN2A diheteromers and GluN2A/GluN2B triheteromers. To answer this question, new tools selective for either GluN2A diheteromers or GluN2A/GluN2B triheteromers are required. In addition, determination of the molecular content of synaptic and extrasynaptic NMDAR pools relies on electrophysiological measurements of NMDA EPSCs and tonic currents, respectively. This paradigm is classically used to discriminate between the two receptor pools (Papouin and Oliet, 2014), yet it lacks accuracy about the precise topological distribution of the various NMDAR populations within the neuron. In addition, although the sharp UV-induced potentiation of tonic currents allows investigation of NMDAR subunit composition with much better sensitivity than traditional pharmacology, part of the variability of photomodulation of NMDAR tonic currents that we observe might arise from difficulties in quantifying small tonic current amplitudes. Finally, our study has likely missed

NMDAR pools (e.g. perisynaptic NMDARs) that may not be activated by single synaptic stimulations and may not contribute to tonic currents either. Further investigations using glutamate uncaging combined with optopharmacology and imaging approaches will be necessary to tackle the molecular and functional heterogeneity of NMDARs with submicrometer resolution.

In conclusion, we have engineered, validated and implemented a new tool, Opto2B, that enables fast and selective interrogation of native GluN2B diheteromers in isolation from other co-expressed NMDARs. Using Opto2B, we established the developmental sequence that shapes the distribution of GluN2A and GluN2B

subunits at synaptic and extrasynaptic sites in CA1 pyramidal neurons, while clarifying a long-standing contentious issue, that of the relative abundance of GluN2A- and GluN2B-containing receptors between both locations. We foresee broad applications of Opto2B and other optopharmacological tools targeting NMDARs to study their molecular and functional adaptation to contextual experience and disease states.

# Methods

**Reagents and tools table**

| Reagent/resource | Reference or source | Identifier or catalog number |
|---|---|---|
| **Experimental models: animals** | | |
| Mouse: C57BL/6J | Charles River Laboratories | |
| Mouse: Opto2B | This study | |
| Mouse: GluN2A-KO | Sakimura et al, 1995 | |
| Mouse: Floxed-Grin2B | Von Engelhardt et al, 2008 | B6.Cg-Grin2btm1Mony/Crl |
| *Xenopus laevis* | TEFOR Paris-Saclay (Saclay, France) https://tefor.net/ European Xenopus Resource Center (Portsmouth, UK) https://xenopusresource.org/ | |
| **Experimental models: cell lines** | | |
| HEK cells | ECACC (https://www.culturecollections.org.uk/ECACC) | 96121229 |
| **Antibodies** | | |
| Anti-GluN2B | Mouse monoclonal clone 13; BD transduction | AB_397797; Clone 13/NMDAR2B |
| Anti-GluN2A | Rabbit monoclonal clone A12W; Merck Millipore | 04-901 |
| Anti-GluN1 | Mouse monoclonal clone N308/48; NeuroMab | AB_2877408; N308/48 |
| Anti-GluA1 | Rabbit monoclonal clone EPR5479; Abcam | Ab109450 |
| Anti-tubulin | Mouse monoclonal clone DM1A, Merck Millipore | 05-589 |
| Peroxidase Affinipure Goat Anti-mouse IgG (H + L) | Jackson ImmunoResearch | AB_10015289; 115-035-003 |
| Peroxidase Affinipure Goat Anti-rabbit IgG (H + L) | Jackson ImmunoResearch | AB_2313567; 111-035-003 |
| **Recombinant DNA** | | |
| pRCCMV_GluN1-1a (rat) | Paoletti et al, 1995 | |
| p3αpA_GluN2B (mouse) | Paoletti et al, 1995 | |
| pCDNA3_GluN2A (rat) | Paoletti et al, 1995 | |
| pRK5_GluN2C (rat) | Rachline et al, 2005 | |
| pRK5_GluN2D (rat) | Rachline et al, 2005 | |
| p3αpA_GluN2B-X, X = S188C, N192C | Esmenjaud et al, 2019 | |
| p3αpA_GluN2B-X, X = E198C, E200C, E201C, D206C | Mony et al, 2011 | |
| p3αpA_GluN2B-D210C | Zhu et al, 2013 | |
| p3αpA_GluN2B-P177C | Tian et al, 2021 | |
| p3αpA_GluN2B-X, X = D181C, N184C, S193C, L199C, L204C, M207C, S208C, L209C, D213C | This study | |
| p3αpA_GluN2B-X-C395S, X = G178C, Y179C, Q180C, V183C, R187C, I190C, E191C | This study | |

| Reagent/resource | Reference or source | Identifier or catalog number |
|---|---|---|
| pRCCMV-GluN1-6A | Stroebel et al, 2014 | |
| pCDNA3_GluN2A-r1 | Stroebel et al, 2014 | |
| p3αpA_GluN2B-r1 | Stroebel et al, 2014 | |
| p3αpA_GluN2B-R187C-C395S-r1 | This study | |
| p3αpA_GluN2B-r2 | Stroebel et al, 2014 | |
| p3αpA_GluN2B-R187C-C395S-r2 | This study | |
| pCAG_GluN2B-IRES-GFP | Sanz-Clemente et al, 2013 | |
| pCAG_GluN2B-R187C-IRES-GFP | This study | |
| pCAG_GluN2B-R187C-C395S-IRES-GFP | This study | |
| pCAG_GluN2Bwt | This study | |
| pCAG_GluN2B-R187C-C395S | This study | |
| pCAG_Cre | Fossati et al, 2019 | |
| pCAG_GFP | Fossati et al, 2019 | |
| pCAG_TdTomato | Fossati et al, 2019 | |
| **Oligonucleotides and other sequence-based reagents** | | |
| Genotyping primers for Opto2B mice: 5'-TCTGTCATGCTCAACATCATGGAAG-3' and 5'-GATGGCAATCCCATCTCTCACTCTG-3' | This study | |
| **Chemicals, enzymes and other reagents** | | |
| **Chemicals and drugs** | | |
| MASp (1-[2-(4-{(E)-[4-(2-{[3-({4-[(3-aminopropyl)amino]butyl}amino)propyl]amino}ethoxy) phenyl] diazenyl}phenoxy)ethyl]-1H-pyrrole-2,5-dione tetrakis(trifluoroacetate)) | This study, now sold by Spectrum Info Ltd. (https://www.spec-info.com/) | F9995-4157 |
| Salts, buffers, glucose | Sigma-Aldrich | |
| D-serine | Sigma-Aldrich | S4250 |
| L-glutamic acid, monosodium salt | Sigma-Aldrich | G1626 |
| Glycine | Sigma-Aldrich | G7403 |
| Diethylenetriamine-pentaacetic acid (DTPA) | Sigma-Aldrich | D6518 |
| Spermine | Sigma-Aldrich | S3256 |
| Adenosine 5'-triphosphate magnesium salt | Sigma-Aldrich | A9187 |
| Guanosine 5'-triphosphate sodium salt hydrate | Sigma-Aldrich | G8877 |
| D-AP5 (APV) | HelloBio | HB0225 |
| NMDA | HelloBio | HB0454 |
| NBQX disodium salt | HelloBio | HB0443 |
| DL-threo-β-Benzyloxyaspartic acid (DL-TBOA) | HelloBio | HB0258 |
| Strychnine | Sigma-Aldrich | S8753 |
| Picrotoxin | HelloBio | HB0506 |
| MK801 | Ascent Scientific (now AbCam) | discontinued |
| Gentamycin 50 mg/mL | Gibco, ThermoFisher | 15750037 |
| MPX-004 | Alomone Labs | M-280 |
| Ro 25-6981 | Gift from Hoffmann-LaRoche | |
| Collagenase type II | Gibco, ThermoFisher | 17101015 |
| **Cell culture reagents** | | |
| DMEM + glutamax | Gibco, ThermoFisher | 31966021 |
| Fetal bovine calf serum | Gibco, ThermoFisher | 10270106 |
| Penicillin/streptomycin (10,000 U/mL) | Gibco, ThermoFisher | 15140122 |

| Reagent/resource | Reference or source | Identifier or catalog number |
|---|---|---|
| Polyethylenimine (PEI), linear, MW 2500, transfection grade | Polysciences Inc. | 23966 |
| HBSS, no Ca, no Mg (10x) | Gibco, ThermoFisher | 14180046 |
| HEPES for cell culture (1 M) | Gibco, ThermoFisher | 15630080 |
| 2.5% Trypsin (100x) | Gibco, ThermoFisher | 15090046 |
| MEM, no glutamine | Gibco, ThermoFisher | 21090022 |
| L-Glutamine (200 mM) | Gibco, ThermoFisher | 25030024 |
| Sodium pyruvate for cell culture (100 mM) | Gibco, ThermoFisher | 11360070 |
| Horse serum | Gibco, ThermoFisher | 16050130 |
| Poly-D-ornithine hydrobromide | Sigma-Aldrich | P8638 |
| Neurobasal medium | Gibco, ThermoFisher | 21103049 |
| B27 supplement (50X), serum free | Gibco, ThermoFisher | 17504044 |
| Molecular biology and biochemistry reagents | | |
| KpnI restriction enzyme | Thermofisher Scientific | ER0523 |
| NotI restriction enzyme | New England Biolabs | R089 |
| BglII restriction enzyme | New England Biolabs | R0144 |
| Pfu ultra DNA polymerase | Agilent | 600382 |
| Dream Taq DNA polymerase | ThermoFisher | EP0705 |
| MgCl$_2$ for molecular biology (25 mM) | ThermoFisher | R0971 |
| dNTPs | ThermoFisher | R0193 |
| NuPage 3-8% Tris-Acetate Midi gel 1.0 mm | ThermoFisher | WG1602BX10 |
| Spectra Multicolor High Range Protein Ladder | ThermoFisher | 26625 |
| HiMark Pre-stained protein standard | ThermoFisher | LC5699 |
| Software | | |
| Clampex 10.3 | Molecular Devices | |
| Graphpad Prism 9 and 10 | GraphPad | |
| Igor Pro 6.0.4.0 | Wavemetrics | |
| Sigmaplot 11 | Systat | |
| Python | | |
| ImageJ | National Institutes of Health (Rueden et al, 2017) | |
| ChemDraw Prime 18.0 | Revvity Signals | |
| Biorender.com | | |

## Chemicals

Salts, buffers, glucose, D-serine, DTPA (diethylenetriamine-pentaacetic acid), glucose, L-glutamate, glycine, spermine, strychnine, Mg-ATP and Na-GTP were purchased from Sigma-Aldrich (St. Louis, MO, USA). D-APV (D-(-)-2-Amino-5-phosphonopentanoic acid), NMDA (N-methyl-D-aspartate), NBQX (3-Dioxo-6-nitro-1,2,3,4-tetrahydrobenzo[*f*] quinoxaline-7-sulfonamide), TBOA (DL-*threo*-β-Benzyloxyaspartic acid), and picrotoxin were purchased from HelloBio (County Meath, ROI). MK801 was purchased from Ascent Scientific (now Abcam, Cambridge, UK). Gentamycin was purchased from GIBCO (Invitrogen, Rockville, MD, USA). Ro 25-6981 is a gift from F. Hoffmann-LaRoche. MPX-004 was purchased from Alomone Labs.

Stock solutions of L-glutamate (100 mM), glycine (100 mM), DTPA (10 mM), D-serine (500 mM), NMDA (100 mM), MK801 (50 μM), D-APV (50 or 100 mM), NBQX (10 mM) and strychnine (10 mM) were prepared in bi-distilled water. Picrotoxin (100 mM), MPX-004 (30 mM) and Ro 25-6981 (10 mM) stock solutions were prepared in DMSO. All stock solutions were stored at −20 °C.

## MASp chemical synthesis and characterization

MASp (1-[2-(4-{(E)-[4-(2-{[3-({4-[(3-aminopropyl)amino]butyl} amino)propyl]amino}ethoxy) phenyl] diazenyl}phenoxy)ethyl]-1H-pyrrole-2,5-dione tetrakis(trifluoroacetate)) was obtained as a trifluoroacetate salt from custom synthesis by Spectrum Info Ltd (https://www.spec-info.com/, Kiev, Ukraine; currently available at https://shop.lifechemicals.com/compound/1/F9995-4157). NMR and mass-spectroscopy characterizations are described in Text S1. MASp stock solutions were prepared in anhydrous DMSO at

concentrations of 16.5 mM (for labeling of Xenopus oocytes and brain slices) or 0.66 mM (for labeling of HEK cells and cultured neurons) and stored at −20 °C.

## MASp photochemical characterization

To avoid hydrolysis of the maleimide moiety during compound characterization, MASp was reacted in the dark with 2 equivalents of L-cysteine (Sigma-Aldrich) during at least 30 min in oocyte recording Ringer solution at pH 7.3 (see below). This reaction is predicted to produce four different cysteine-conjugated MASp (MASp$^{Cys}$) diastereoisomers (Appendix Fig. S6). Spectroscopic analyses and HPLC analyses were made on the crude product mixture. All experiments were performed at room temperature.

UV-visible absorption spectra were acquired in 1 cm long quartz cuvettes on a NanoPhotometer® NP80 spectrometer (Implen, Germany). MASp was diluted at 33 µM in Ringer (pH 7.3) from the 16.5 mM DMSO stock (0.2% DMSO final concentration) solution together with 100 µM L-cysteine to yield MASp$^{Cys}$. Blank solution was Ringer at pH 7.3 containing 0.2% DMSO and 100 µM L-cysteine. Photostationary states (PSS) of MASp$^{Cys}$ cis and trans isomers were obtained by continuous illumination of the quartz cuvette containing the MASp$^{Cys}$ solution with a multi-wavelength LED (pE-2 and pe-4000, CoolLED, UK) until no further change in the absorption spectra was observed. For all irradiation wavelengths tested, 5 min illumination was sufficient to reach steady state. Photostability of the cis state (365 nm PSS) was measured by irradiating the MASp$^{Cys}$ solution with 365 nm light during 10 min, then letting it relax in the dark, inside the spectrophotometer. Spectra were acquired at regular intervals, up to 8 h after irradiation.

Analytical HPLC was performed on an Agilent 1200 series equipped with a quaternary pump using a Proto 200 C18 column from Higgins Analytical Inc (particles size 3 µm, 100×4.6 mm column). The compounds (66 µM of MASp$^{Cys}$ products in Ringer solution) were eluted with a flow of 1 ml/min using a gradient of acetonitrile (0 to 100% over 10 min) in water, both solvents containing 0.1% TFA. The detection was performed at 220 nm, 280 nm and 440 nm (MASp$^{Cys}$ isosbestic point, see Fig. EV1A). MASp$^{Cys}$ cis/trans PSS in the dark and after illumination with 365 and 525 nm light were determined by HPLC using the relative integrated areas of the cis and trans peaks at 440 nm. HPLC of a baseline solution containing 0.4% DMSO and 200 µM L-Cysteine was measured and its chromatogram subtracted to the one of MASp$^{Cys}$. Due to the presence of two diastereoisomers per azobenzene configuration after reaction with cysteine, trans and cis MASp$^{Cys}$ were each represented by two peaks (see Fig. EV1C).

## Molecular biology

For expression in HEK cells and Xenopus oocytes, rat GluN2A and mouse GluN2B (ε2) subunits and eGFP were expressed using pcDNA3-based expression plasmids, and rat GluN1-1a (named GluN1 herein) using pRCCMV plasmid. For selective expression of tri-heteromeric NMDARs, DNAs coding for a modified GluN1 subunit (GluN1-6A) and GluN2A and GluN2B subunits containing the GABA$_B$ retention signals (GluN2A-r1, GluN2B-r1 and GluN2B-r2) were from Stroebel et al (2014). Point mutations were performed by Quikchange mutagenesis and DNA sequences verified by Sanger sequencing.

For ex utero and in utero electroporation, mouse GluN2B-R187C-C395S, GluN2B-R187C-C395S-IRES-GFP, GluN2B-R187C-IRES-GFP, Cre, GFP and Td-tomato constructs were expressed using pCAG-based plasmids. pCAG_GFP and pCAG_Td-tomato were from Fossati et al, (2019). pCAG_GluN2B-R187C-C395S was obtained by subcloning from the pCAG_GFP and p3αpA_GluN2B-R187C-C395S plasmids using KpnI (ThermoFisher Scientific) and NotI (New England Biolabs) restriction enzymes. pCAG_GluN2B-R187C-C395S-IRES-GFP and pCAG_GluN2B-R187C-IRES-GFP were obtained by Quikchange mutagenesis of the pCAG_GluN2B-IRES-GFP plasmid (a gift from Katherine Roche, Sanz-Clemente et al (2013)).

## Oocyte preparation and injection

Oocytes from *Xenopus lævis* were used for heterologous expression of recombinant NMDA receptors for two-electrode voltage-clamp (TEVC) experiments. Female *Xenopus laevis* were housed and ovary bags harvested according to European Union guidelines (husbandry authorizations #C75-05-31 and #D75-05-31; project authorizations #05137.02 and Apafis #28867-2020121814485893). Fragments of ovary bags were also purchased from the "Centre de Ressources Biologiques Xenopes" (now TEFOR, Paris Saclay, France) and from the European Xenopus Resource Center (EXRC, Portsmouth, UK). *Xenopus laevis* oocytes were harvested and prepared as previously described in Paoletti et al, (1995). Briefly, membranes of ovary bags were teared with forceps to expose the oocytes to the medium. Ovary bag fragments were then subjected to digestion with Collagenase type II (Gibco, 1–1.5 mg/mL) diluted in a calcium-free, OR2 medium (in mM: 85 NaCl, 5 HEPES, 1 MgCl$_2$, pH adjusted to 7.6 with KOH) under mild shaking (~110 rpm) until the oocytes were fully defolliculated (usually after ~1 h). Oocytes were then washed five times in OR2 then three times with a Barth solution (in mM: 88 NaCl, 1 KCl, 0.33 Ca(NO$_3$)$_2$, 0.41 CaCl$_2$, 0.82 MgSO$_4$, 2.4 NaHCO$_3$ and 7.5 HEPES, pH adjusted to 7.3 with NaOH). Defolliculated oocytes were stored at 12 °C in a Barth solution supplemented with gentamycin (50 µg/µL).

Expression of recombinant di-heteromeric NMDARs was obtained by oocyte nuclear co-injection of 37 nL of a mixture of cDNAs (at 10–30 ng/µL) coding for GluN1-1a and various GluN2 subunits (ratio 1:1). Expression of tri-heteromeric NMDARs using modified subunits containing the GABA$_B$ ER retention signals was obtained by co-injecting a mixture of cDNAs coding for GluN1-6A, GluN2-r1 and GluN2-r2 subunits at 45 ng/µL (ratio 2:1:1) (Stroebel et al, 2014). Co-injection of a mixture of GluN1-6A/GluN2-r1 or GluN1-6A/GluN2-r2 (45 ng/µL, 1:1 ratio), which is not supposed to yield membrane expression of NMDARs, was systematically performed to monitor leakage of the retention motifs (Stroebel et al, 2014). On the day of the experiment, currents from escaped diheteromeric receptors (i.e. from oocytes injected with GluN1-6A/GluN2-r1 or GluN1-6A/GluN2-r2) were systematically monitored and subsequent experiments were only performed if these currents were <10% of tri-heteromeric currents. After injection, the oocytes were transferred to 96-well plates filled with Barth supplemented with gentamicin (50 µg/µL) and 50 µM APV, a selective NMDAR antagonist. Plates were then stored at 18 °C for 24 h for expression of GluN1/GluN2A constructs, and 48–96 h for expression of GluN1/GluN2B, GluN1/GluN2C, GluN1/GluN2D and tri-heteromeric constructs.

## Oocyte labeling, TEVC recordings and NMDAR photomodulation

### Oocyte labeling

Oocytes (~5 oocytes per 1 mL labeling solution) were labeled in a Barth solution containing 0.66 μM MASp (diluted from a 16.5 mM stock, 0.004% DMSO in final dilution) during 15 min on ice (0 °C), in the dark, under mild shaking (90 rpm). Oocytes were then thoroughly washed in 4 mL of Barth solution during 10 min in the dark, on ice, under mild shaking (90 rpm), during three consecutive times (30 min total wash time). Oocytes were transferred to a new well containing 4 mL Barth between each washing step. Increasing the labeling step to 30 min instead of 15 min did not increase the amplitude of photomodulation. On the other hand, we observed strong non-covalent association of MASp with NMDARs and/or the oocyte membrane, so that increase of MASp concentration in the labeling solution (to 6.6 μM) or decrease of the wash time resulted in incomplete washout of the non-covalently bound compound. This was evidenced by a slow inhibition of WT GluN1/GluN2B or GluN1/GluN2A receptor currents by UV light, which was slowly reversed by green light. This effect slowly washed away as the cell was submitted to several illumination cycles. We attributed this slow and non-specific effect on NMDAR currents to a high affinity, photo-dependent pore block. It has indeed been shown that aromatic polyamines induce high affinity inhibition of NMDARs through a pore block mechanism (Williams, 1997; Igarashi et al, 1997; Kashiwagi et al, 1997). Reducing the labeling time, MASp concentration and increasing wash time allowed us to eliminate this non-covalent MASp effect, with no effect of light on WT GluN1/GluN2B NMDARs with our final labeling conditions (Fig. 1E).

### TEVC recordings

In all, 1–4 days following DNA injection, TEVC recordings were performed using an Oocyte Clamp amplifier OC-725 (Warner Instruments) computer-controlled via a 1440 A Digidata (Molecular Devices). Currents were sampled at 100 Hz and low-pass filtered at 20 Hz using an 8-pole Bessel filter (900 Series, Frequency Devices Inc). Data were collected with Clampfit 10.3. During the recording, the cells were continuously perfused with external recording Ringer solution at either pH 7.3 (in mM: 100 NaCl, 0.3 BaCl$_2$, 5 HEPES and 2.5 KOH, pH adjusted to 7.3 by addition of HCl) or 6.5 (in mM: 60 NaCl, 0.3 BaCl$_2$, 40 HEPES, 2.5 KOH, pH adjusted to 10.3 with NaOH, then back to 6.5 with HCl, see Mony et al (2011)). Unless otherwise noted, NMDA currents were induced by simultaneous application of L-glutamate and glycine (agonist solution) at saturating concentration (100 μM each), and DTPA (10 μM) to prevent receptor inhibition by ambient zinc (~20 nM, Paoletti et al (1997)). Control, agonist free solution contained 10 μM DTPA. All recordings were performed at a holding potential of −60 mV and at room temperature.

### Photomodulation

Photomodulation of NMDA currents on Xenopus oocytes was performed by irradiating the oocyte from the top (irradiation of the animal pole) during TEVC recording with a PE-2 light source (CoolLED) coupled to a liquid light guide, using wavelengths of either 365 nm (irradiance ~8 mW/mm²) or 490 nm (irradiance ~18 mW/mm²).

### Pharmacological characterization of NMDAR mutants

Pharmacological characterization was performed on unlabeled oocytes, or on MASp-labeled oocytes in the dark or under constant illumination with 365 or 490 nm light. Glutamate and glycine dose-response curves were recorded at pH 7.3. Glutamate dose-response curves were performed in presence of 100 μM glycine and varying concentrations of glutamate. Glycine dose-response curves were performed in presence of 100 μM glycine and varying concentrations of glutamate. For each cell, agonist dose-response curves were fitted with the Hill equation: $I = I_{max}/(1 + (EC_{50}/[A])^{nH})$, where $I_{max}$ is the maximum current calculated from the fit, $EC_{50}$ the agonist concentration necessary to induce 50% of $I_{max}$, $n_H$ is the Hill coefficient, and $[A]$ the agonist concentration. $I_{max}$, $n_H$ and $EC_{50}$ were fitted as free parameters. Proton dose-response curves were performed and analyzed as previously described (Gielen et al, 2008). Spermine sensitivity was assessed by measuring currents in absence and in presence of 200 μM spermine at pH 6.5, as previously described (Mony et al, 2011). Open probability (Po) of the different NMDAR mutants was assessed by measuring the rate of inhibition by 100 nM MK801, an open channel blocker whose inhibition kinetics correlate with channel Po (Chen et al, 1999; Blanke and VanDongen, 2008; Talukder et al, 2010; Hansen et al, 2013; Gielen et al, 2009). Recordings were performed at acidic pH (6.5) to maximize the differences of Po between the MASp-potentiated and non-potentiated receptors. MK-801 time constants of inhibition ($\tau_{on}$) were obtained by fitting inhibition currents with a single-exponential function. On-rate ($k_{on}$) constants were then calculated assuming a pseudo first-order reaction scheme: $k_{on} = 1/([MK\text{-}801]^*\tau_{on})$. In Fig. EV2G, all $k_{on}$ were normalized to the average $k_{on}$ of untreated, wt GluN1/GluN2B receptors measured in the same conditions on the same day.

## Mice

Mice were housed in the IBENS rodent facility duly accredited by the French Ministry of Agriculture. All experiments were performed in compliance with French and European regulations on care and protection of laboratory animals (EU Directive 2010/63, French Law 2013-118, February 6th, 2013), and were approved by local ethics committees and by the French Ministry of Research and Innovation (authorization numbers #05137.02, APAFIS #28867-2020121814485893 and APAFIS #29476-2021020311595454). Animals were maintained on a 12-h light/dark cycles with food and water provided ad libitum.

For ex vivo experiments performed in the cortex, in utero electroporation was performed on time-pregnant Swiss mice, and both male and female pups were used for recordings. For ex vivo recordings in the hippocampus, both male and female mice were used as well with their corresponding littermates from the following lines: WT C57Bl/6N, GluN2B-R187C KI (Opto2B mouse, generated in this study, see below), GluN2A KO (Sakimura et al, 1995), and Opto2B/GluN2A-KO generated by crossing the Opto2B and GluN2A KO lines. Opto2B mice are on a C57Bl/6N background, GluN2A KO mice on a C57Bl/6J background, and Opto2B/GluN2A KO mice on a mixed C57Bl/6N and C57Bl/6J background. For neuronal culture experiments, WT C57Bl/6J mice or floxed-Grin2B transgenic mice (B6.Cg-Grin2btm1Mony/Crl; von Engelhardt et al, 2008) were used. All the mice used in our experiments were

homozygous for their corresponding gene(s). Experiments were not performed blind.

### Generation of the Opto2B mouse and determination of genotype

The Opto2B mouse, containing the GluN2B-R187C mutation, was generated by the *Institut clinique de la souris*, Phenomin (Illkirch, France) using the CRISPR/Cas9 strategy. At the position equivalent to R187 on GluN2B, codon CGC was replaced by TGC, leading to substitution of an arginine to a cysteine. The C > T substitution also created a BglII enzyme restriction site, which was used to discriminate WT and mutant animals during genotyping (AGATC*T*, the T in bold italic being the substituted base).

Genotyping consisted of several steps. DNA was first dissociated in 25 mM NaOH during 1 h at 93 °C and the reaction was stopped with 40 mM TrisHCl. DNA was then amplified by PCR (DreamTaq DNA Polymerase—MgCl$_2$ 1 mM–0.2 mM dNTP; 34 cycles: 30 s at 94 °C, 30 s at 62 °C, 60 s at 72 °C, then 1 min at 72 °C) using primers 5′-TCTGTCATGCTCAACATCATGGAAG-3′ and 5′-GATGGCAATCCCATCTCTCACTCTG-3′. DNA products were then digested overnight using 0.1 U/µL of restriction enzyme BglII (New England Biolabs). Discrimination between genotypes was performed on a gel electrophoresis according to the presence or absence of WT (undigested bands, expected molecular weight: 435 bp) and digested mutant GluN2B characteristic bands (expected molecular weights: 341 and 94 bp).

## Dissociated cell culture and transfection/electroporation

### HEK cells

HEK-293 cells (obtained from ECACC, Cat #96121229) were cultured in DMEM + glutamax medium supplemented with 10% fetal bovine calf serum and 1% Penicillin/streptomycin (10,000 U/ml and 10,000 µg/mL, respectively), under standard cell culture conditions (5% CO$_2$, 37 °C). Transfections were performed using polyethylenimine (PEI, linear 25 kD; Polysciences, Inc., Eppelheim, Germany; stock at 1 mg/mL) with a cDNA/PEI ratio of 1:3 (v/v). Cells were co-transfected with a DNA-mixture containing plasmids encoding wild-type GluN1, GluN2A or GluN2B constructs, and eGFP. The total amount of DNA was 1 µg per 500 µL of transfected medium for a 12 mm$^2$ diameter coverslip. The DNA mass ratio for GluN1:GluN2B:eGFP was 1:2:1 and 1:1:1 for GluN1:GluN2A:eGFP. 150 µM of D-APV was added to the culture medium after transfection. Currents from HEK cells expressing GluN1/GluN2A and GluN1/GluN2B constructs were recorded 24 h and 48–72 h post transfection, respectively.

### Cortical neurons

Dissociated cultures of cortical neurons were prepared from mouse embryos at E15. After brain extraction, expression of GluN2B-R187C-C395S mutated subunit in cortical was performed by ex utero electroporation. Endotoxin-free cDNAs coding for GluN2B-R187C-C395S (pCAG_GluN2B-R187C-C395S plasmid, 0.5 µg/µL) and for a GFP fluorescent marker (pCAG_GFP plasmid, 0.5 µg/µL) were injected unilaterally into the lateral ventricle of the mouse embryos using a glass pipette. For experiments using the Cre-Lox strategy, GluN2B-R187C-C395S (pCAG_GluN2B-R187C_IRES_GFP, 1 µg/µL) or GluN2B-R187C (pCAG_GluN2B-R187C_IRES_GFP, 1 µg/µL) subunits were expressed as well as a Cre-recombinase (pCAG_Cre, 1 µg/µL) on floxed-Grin2B mice (von Engelhardt et al, 2008) following the same protocol. As the GFP

fluorescence from IRES constructs was barely visible, a Td-Tomato fluorescent marker (pCAG_Td-Tomato, 1 µg/µL) was systematically co-electroporated. The volume of injected DNA was adjusted depending on the experiments. Electroporation was performed using a square wave electroporator (ECM 830, BTX) and tweezer-type platinum disc electrodes (5 mm diameter, Sonidel). The electroporation settings were: 5 100 ms-long pulses at 18 V separated by 100 ms.

After removing meninges, electroporated cortices were placed in ice-cold HBSS solution supplemented with 20 mM HEPES. Cell dissociation was performed individually for cortices of each embryo. Cortices were incubated in 2.5% trypsin at 37 °C for 10 min, rinsed three times with 37 °C HBSS solution, and further dissociated by trituration with syringes of decreasing diameter (21 then 24 gauge). Cells were resuspended in attachment medium (MEM supplemented with 2 mM L-glutamine; penicillin/streptomycin at 5 U/ml and 5 µg/mL, respectively; 1 mM sodium pyruvate; and 10% horse serum) then plated on poly-DL-ornithine coated coverslips in 24-well culture dishes at a density of 1–2 × 10$^5$ cells per well. Cells were cultured under standard cell culture conditions (37 °C, 5% CO$_2$). Attachment medium was changed to Neurobasal medium (supplemented with 2 mM L-glutamine; 1× B27 supplement; and Penicillin/Streptomycin, 5 U/mL and 5 µg/mL, respectively) 2 h after plating. Cells were fed by changing ½ medium to fresh Neurobasal medium every 4 days. Cultures were used for experiments after 6 to 9 days in vitro (DIV6 to DIV9).

## Labeling and patch-clamp electrophysiology on dissociated cells

Before patch-clamp recording, HEK cells and cultured cortical neurons were labeled with MASp. Labeling was performed by incubating the coverslips in 500 µM of extracellular recording solution (in mM: 140 NaCl, 2.8 KCl, 1 CaCl$_2$, 10 HEPES, 20 sucrose and 0.01 DTPA; 290–300 mOsm; pH adjusted to 7.3 using NaOH) containing 0.66 µM MASp (0.1% final DMSO concentration) at 37 °C in the cell culture incubator. Coverslips were then transferred into a petri dish containing ~4 mL extracellular recording solution, then to the patch-clamp recording chamber.

Whole-cell patch-clamp recordings were performed on an Olympus IX73 inverted microscope. Positively transfected cells were visualized by GFP fluorescence. Patch pipettes had a resistance of 3–6 MΩ and were filled with a solution containing (in mM): 115 CsF, 10 CsCl, 10 HEPES and 10 BAPTA (280–290 mOsm), pH adjusted to 7.2 using CsOH. Currents were sampled at 10 kHz and low-pass filtered at 2 kHz using an Axopatch 200B amplifier, a 1550B digidata and Clampex 10.6 (Molecular Devices). Recordings were performed at a holding potential of −60 mV and at room temperature. Agonists (100 µM glutamate and glycine for HEK cells; 300 µM NMDA and 50 µM D-serine for cortical neurons) were applied using a multi-barrel solution exchanger (RSC 200, BioLogic). In Appendix Fig. S4A, NMDAR currents from cultured cortical neurons were elicited by varying concentrations of glutamate and isolated by adding 10 µM NBQX, 100 µM picrotoxin, 10 µM strychnine, and 20 µM glycine to the extracellular recording solution, in order to mimic the recording conditions of neurons in brain slices (see below).

Computer-controlled light pulses during electrophysiological recordings were provided from high power LEDs (Prizmatix). The

three following LEDs were used: Mic-LED-365 (365 ± 4 nm, 200 mW), UHP-Mic-LED-460 (460 ± 5 nm, 2 W) and UHP-Mic-LED-520 (520 ± 5 nm, 900 mW). The LED port was directly coupled to the fluorescence port of the microscope. The output beam of the LED entry was directed towards the sample thanks to a mirror (Chroma) and applied to the center of the recording dish through a 10X objective (Olympus, 0.30 N.A.). In Appendix Fig. S1G, light power was measured at the center of the recording chamber plane with an optical power meter (1916-C, Newport) equipped with a calibrated UV/D detector.

## In utero electroporation

In utero electroporation was performed as previously described (Fossati et al, 2019). Briefly, pregnant Swiss females at E14.5–15.5 (Janvier labs) were subcutaneously injected with 0.1 mg/kg of buprenorphine for analgesia and anesthetized with isoflurane (3.5% for induction and 2% during the surgery). The uterine horns were exposed after laparotomy. Endotoxin-free DNA diluted in 1× PBS with 0.1% Fast Green dye for visualization was injected unilaterally into the lateral ventricle of the mouse embryos using a glass pipette. The volume of injected DNA was adjusted depending on the experiments. Electroporation was performed using a square wave electroporator (ECM 830, BTX) and tweezer-type platinum disc electrodes (5 mm diameter, Sonidel). The electroporation settings were: 5 pulses of 50 V for 50 ms with 500 ms interval. DNAs were used at the following concentration: pCAG_Td-tomato, 1 µg/µL; pCAG_GluN2B-R187C-C395S, 2 µg/µL.

## Ex vivo patch clamp electrophysiology

### Slice preparation and labeling

Acute coronal slices (320 µm) were prepared at the indicated age by decapitation of the isoflurane-anesthesized animals. After brain extraction, slices were prepared using a vibratome (Leica VT1200S). Slicing was performed in a cold (~4 °C) and oxygenated (95% $O_2$, 5% $CO_2$) slicing solution containing (in mM): 92 choline chloride, 2.5 KCl, 1.2 $NaH_2PO_4$, 30 $NaHCO_3$, 20 HEPES, 25 glucose, 5 ascorbic acid, 3 sodium pyruvate, 10 Magnesium sulfate, 0.5 calcium chloride, pH adjusted to 7.3 by Tris base. Slices were transferred to an ACSF solution (in mM: 125 NaCl, 2.5 KCl, 1.25 $NaH_2PO_4$, 26 $NaHCO_3$, 20 D-glucose, 2 $CaCl_2$, 1 $MgCl_2$) at 32 °C under constant oxygenation and left to recover for 1–1.5 h. Labeling was performed by incubating the slices in oxygenated ACSF containing 6.6 µM MASp (from 16.5 mM stock, 0.04% DMSO in final dilution) during 30 min at 32 °C. Slices were then kept in ACSF before recording.

### Electrophysiology

Slice electrophysiology was performed on a Scientifica SliceScope upright microscope coupled to an OrcaFlash 4.0 camera for cell and fluorescence visualization. Electroporated cells were identified by their Td-tomato fluorescence. Whole-cell patch-clamp recordings were performed at ~28–30 °C in either layer 2/3 cortical pyramidal neurons of the somatosensory cortex or CA1 pyramidal neurons of the hippocampus. Currents and potentials were recorded using a Multiclamp 700B amplifier coupled to a 1550B Digidata and Clampex 10.6 (Molecular Devices). Currents were sampled at 10 kHz. Patch pipettes (Hilgenberg) had a resistance of 3 to 6 MΩ.

For voltage-clamp recordings, cells were patched using the following intracellular solution (in mM): 125 CsMeSO4, 10 BAPTA, 5 TEA, 10 HEPES, 4 Mg-ATP, 0.2 $Na_3$-GTP (280–290 mOsm, pH adjusted to 7.3 using CsOH). Series resistance (typically <20 MOhm before compensation) was compensated (around 40–60%) and monitored during the whole experiment. Extracellular synaptic stimulation was achieved by applying voltage pulses (3.5 ms, 5–50 V; Digitimer Ltd, UK) via a second patch pipette filled with HBS (in mM: 150 NaCl, 2.5 KCl, 1.25 $NaH_2PO_4$, 10 HEPES, 2 $CaCl_2$, 1 $MgCl_2$; pH adjusted to 7.4 with NaOH) and placed near the apical dendrite of the patched neuron (for cortical neurons; Fossati et al, 2019), or by stimulating the afferent Schaeffer collaterals in the *stratum radiatum* for CA1 hippocampal neurons. AMPA currents were recorded at −70 mV in ACSF. NMDA currents (NMDA EPSCs and tonic currents) were recorded at +40 mV and isolated by adding 10 µM NBQX, 100 µM picrotoxin, 10 µM strychnine and 20 µM glycine to ACSF (as in Yan et al, 2020). NMDA/AMPA ratios were recorded in regular ACSF and calculated as the ratio of EPSC amplitude at +40 mV holding potential measured 50 ms after peak (NMDA component) over EPSC at peak at −70 mV holding potential (AMPA component).

For current clamp experiments patch pipettes were filled with the following intracellular solution (in mM): 130 K-gluconate, 5 KCl, 10 HEPES, 0.6 EGTA, 2 $MgCl_2$, 0.2 $CaCl_2$, 2 Mg-ATP, 0.3 $Na_3$-GTP (290–300 mOsm, pH adjusted to 7.3 using KOH). Cell hyperpolarization/depolarization and action potentials were induced by 500 ms current injections ranging from −300 to +800 pA.

### Photomodulation and analysis of light-dependent effects

Computer-controlled light pulses during electrophysiological recordings were provided from ThorLabs LEDs. The two following LEDs were used: M365L2 (365 nm) and M530L3 (530 nm). The LED port was directly coupled to the fluorescence port of the microscope. The output beam of the LED entry was directed towards the sample thanks to a mirror (Chroma) and applied to the center of the recording chamber through a ×40 objective (Olympus, 0.1 N.A.). We observed that UV light at high power abolished synaptic transmission. UV light intensity was thus chosen so that in induces minimal decrease of synaptic transmission while still allowing photomodulation. Final powers measured in the recording chamber for UV and green light illuminations were ~2 mW for 365 nm light and ~0.3 mW for 530 nm.

Effect of light on NMDA EPSCs was monitored by illuminating the region around the patched neuron 1 s before stimulation until 200 ms after stimulation (1.2 s total illumination). Photomodulation ratios were measured by performing cycles of five stimulations at 0.1 Hz with green (530 nm) light, followed by five stimulations at 0.1 Hz with UV (365 nm) light, followed by three other green-UV-green cycles (Fig. EV3A). Due to the slow reversal of UV-induced potentiation by green light, steady-state current under green light was obtained only after the third green light illumination bout (Fig. EV3A). As a consequence, the average of only the last three EPSCs of the five-stimulation cycles (either with UV or green light) were considered to calculate the photomodulation ratios (Fig. EV3A, grey bars). For cortical neurons, photomodulation ratios were calculated as the ratios of the mean EPSC amplitude under UV light (3 EPSCs x 2 UV cycles = 6 EPSCs) over the mean

EPSC amplitude under green light (3 EPSCs x 3 green cycles = 9 EPSCs). The same protocol and data analysis were applied to investigate the photodependence of AMPA-EPSCs after MASp treatment of the slice (Fig. EV3D). For hippocampal neurons, the absolute NMDA-EPSC responses decreased after the first green-UV-green cycle for some cells. Hence, to decrease cell-to-cell variability, only the first green-UV-green cycle was taken into account to calculate the photomodulation ratio (meaning 3 EPSCs x 1 UV cycle = 3 EPSCs in UV light and 3 EPSCs x 2 green cycles = 6 EPSCs in green light).

NMDA-EPSC decay time constants ($\tau_w$ NMDA-EPSC) were calculated by fitting the decay phase of the averaged EPSCs under green light with a double exponential using the following formula: $I = I_0 + A_{fast} \cdot \exp\left(\frac{-(t-t_0)}{\tau_{fast}}\right) + A_{slow} \cdot \exp\left(\frac{-(t-t_0)}{\tau_{slow}}\right)$, where $\tau_{fast}$ and $\tau_{slow}$ represent the fast and slow time constants of EPSC decay, respectively, and $A_{slow}$ and $A_{fast}$ their relative weights. $\tau_w$ NMDA-EPSC represents the weighted time constant calculated as follows: $\frac{A_{fast}}{A_{fast}+A_{slow}} \cdot \tau_{fast} + \frac{A_{slow}}{A_{fast}+A_{slow}} \cdot \tau_{slow}$.

The protocol of photomodulation of tonic currents consisted of 5–10 cycles of 2 s of 365 nm illumination and 4 s of 530 nm illumination, repeated every 30 s (see Fig. 3D; Appendix Fig. S3A,C). Cells were maintained in the dark between illumination steps. Except for Appendix Fig. S3C, calculations were performed on the average current trace from the 5 or 10 illumination cycles. APV was systematically applied in the dark after the 5 or 10 cycles of illumination to allow calculation of photomodulation ratios, and the illumination protocol was repeated in presence of APV to check for proper inhibition of all NMDARs, including the photodependent receptors. Since APV was applied in the dark and after green light illumination, the APV-sensitive current ($I_{APV}$) represents the basal NMDA tonic current in the dark or under green light ($I_{APV} = I_{530\ nm}$). Under UV light, NMDA tonic current is calculated as $\Delta_{UV,\ peak} + I_{APV}$ at the time of peak, with $\Delta_{UV,\ peak}$ representing the UV-induced increase in tonic current at its peak (~35 ms after the onset of UV light, mean around the peak over a windows of 0.30 ms to avoid bias from non-specific noise) (see Appendix Fig. S3A). The photomodulation ratio was calculated as $(\Delta_{UV,\ peak} + I_{APV})/I_{APV}$ (Fig. 3D). Rising and decaying phases of the UV-induced current peak were fitted with single exponentials.

## Subtype-specific pharmacology on NMDA-EPSCs

For Ro 25-6981 or MPX-004 experiments, slices from Opto2B animals were labeled with MASp but not the ones from WT animals. NMDA-EPSCs were isolated by adding to ACSF 10 μM NBQX, 100 μM picrotoxin, 10 μM strychnine but, contrary to photomodulation experiments, no glycine was added to maximize the effect of MPX-004 (Yi et al, 2016). A baseline of NMDA-EPSCs was recorded for at least 5 min before adding the pharmacological agent. Ro 25-6981 was applied at 1 μM and MPX-004 at 30 μM. NMDA-EPSCs were recorded for at least 5 min after the effect of the inhibitors has reached steady state (total duration of ~30 min). Cells were maintained in the dark. Only cells with stable EPSCs before application of the inhibitor were analyzed. For each drug, percentage of inhibition effect was calculated as the ratio between the average of EPSC amplitudes over 3 min before drug application and the average of EPSC amplitudes over 3 min once stability is reached after drug action.

## Quantification of total brain expression of NMDA and AMPA subunits in Opto2B and WT mice by western blot analysis

−/− Opto2B mice and their WT littermates (+/+ Opto2B) aged P46-54 (both males and females) were anesthetized with isoflurane and decapitated. Brains (without cerebellum and olfactory bulbs) were extracted in ice-cold, oxygenated slicing solution (see above), and hemispheres were separated. Each hemisphere was flash-frozen in liquid nitrogen and stored at −80 °C until use.

One brain hemisphere from each animal was homogenized with a Teflon glass homogenizer in 1.2 mL lysis buffer containing: 290 mM saccharose; 12.5% v/v Tris 0.5 M pH 6.8; 1% SDS; and a protease inhibitor cocktail tablet (Complete, Mini; Roche) (0.1 tablet/mL). Lysed brains were centrifuged at $16,000 \times g$ for 15 min at 4 °C and the supernatant collected. Total protein concentration in each sample was measured by Pierce™ BCA Assay (ThermoFisher Scientific). Samples were then aliquoted in 50 μL aliquots and stored at −20 °C.

Samples (15 μg total protein each) were separated in reducing conditions by 3–8% SDS-PAGE, dry-transferred to nitrocellulose membrane and immunoblotted using the following antibodies: anti-GluN2B (1:615, mouse monoclonal clone 13, BD Transduction); anti-GluN2A (1:300, rabbit monoclonal clone A12W, Merck Millipore); anti-GluN1 (1:500, mouse monoclonal clone N308/48, NeuroMab); anti-GluA1 (1:1000, rabbit monoclonal clone EPR5479; Abcam); and anti-tubulin (1:1000 or 1:5000, mouse monoclonal clone DM1A, Merck Millipore) to normalize protein signals. Protein bands were visualized using secondary peroxidase-linked goat anti-rabbit or anti-mouse (1:10,000, Jackson ImmunoResearch), with the SuperSignal West Pico Chemiluminescent Substrate (ThermoFisher Scientific).

Western blot quantification was performed using the ImageJ software (Rueden et al, 2017). For each protein, band intensities were divided by the intensity of the corresponding α-tubulin band. Within the same blot, protein/tubulin ratios of WT and −/− Opto2B samples were normalized to the mean protein/tubulin ratio of WT samples.

## Data analysis and statistical analysis

Electrophysiological data were analyzed using Clampfit 10.3 (Molecular Devices), Prism (GraphPad software), Igor (Wavemetrics) and a built-in Python script. Graphs were generated and statistical analyses were performed using Sigmaplot 11, GraphPad Prism 9 and 10, or Python (using Matplotlib (Hunter, 2007)—version 3.7.4 -, Seaborn (Waskom, 2021)—version 0.13.1 -). Values are represented as mean ± s.e.m. $n$ represents the number of cells. Except for Fig. 1D, for each condition experiments were performed on at least three different batches of cells or three different animals. Statistical tests are displayed in the Figure Legends and exact $P$ values summarized in Dataset EV1.

## Scientific illustration

Chemical structures were drawn with ChemDraw Prime. Some parts of Figs. 3A, 4A,B, 6A, EV3A and EV5A were created using Biorender.com.

## Data availability

This study includes no data deposited in a repository. New materials are available upon request to the corresponding authors.

The source data of this paper are collected in the following database record: biostudies:S-SCDT-10_1038-S44318-025-00498-x.

## Peer review information

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

## Acknowledgements

We thank Nora Assendorp and Doris Wennagel for their help with in utero electroporation; Julie Lefrançois, Mélissa David and Mathilde Murat for their help with cell culture, molecular biology and genotyping; IBENS rodent facility for help with mouse line management; Maria Rodrigo for her help with electrophysiology; Nicolas Delsuc (Chemistry Department, ENS, Paris, France) for training and help on HPLC. The pCAG-GluN2B-IRES-GFP plasmid was a gift from Katherine Roche (NIH, Bethesda, MD, USA). This project was supported by the French Ministry of Research and the Fondation pour la Recherche Médicale (doctoral fellowships attributed to AS), the National Natural Science Foundation of China (#32300798 to MT), the European Commission (Marie-Sklodowska-Curie fellowship H2020-MSCA-IF-2015 Grant #701467 to LM), the European Research Council (ERC Advanced Grant #693021 to PP and ERC starting grant #803704 to CCh), the Fondation pour la Recherche Médicale (FRM) (fellowship no. FDT202304016679 to AS), and the French Agence Nationale de la Recherche (ANR JCJC Grant #22-CE16-0016 Opto2B to LM).

## Author contributions

**Antoine Sicard**: Data curation; Formal analysis; Validation; Investigation; Visualization; Methodology; Writing—original draft. **Meilin Tian**: Data curation; Formal analysis; Validation; Investigation; Visualization; Methodology. **Zakaria Mostefai**: Data curation; Formal analysis; Validation; Investigation; Visualization; Methodology. **Sophie Shi**: Data curation; Formal analysis; Validation; Investigation; Visualization; Methodology. **Cécile Cardoso**: Investigation; Visualization; Methodology. **Joseph Zamith**: Investigation; Methodology. **Isabelle McCort-Tranchepain**: Data curation; Formal analysis; Validation; Investigation; Visualization; Writing—original draft. **Cécile Charrier**: Supervision; Funding acquisition; Validation; Visualization; Methodology. **Pierre Paoletti**: Conceptualization; Resources; Data curation; Supervision; Funding acquisition; Validation; Investigation; Visualization; Methodology; Writing—original draft; Project administration; Writing—review and editing. **Laetitia Mony**: Conceptualization; Resources; Data curation; Formal analysis; Supervision; Funding acquisition; Validation; Investigation; Visualization; Methodology; Writing—original draft; Project administration; Writing—review and editing.

Source data underlying figure panels in this paper may have individual authorship assigned. Where available, figure panel/source data authorship is

listed in the following database record: biostudies:S-SCDT-10_1038-S44318-025-00498-x.

## Disclosure and competing interests statement

The authors declare no competing interests.

# Expanded View Figures

**Figure EV1.   (related to Fig. 1): Photochemical properties of MASp and design of the labeling conditions in Xenopus oocytes.**

(A) UV-visible spectra of MASp conjugated to L-cysteine (MASp$_{Cys}$, see "Methods" and Appendix Fig. S6) in oocyte recording medium (pH 7.3, 33 µM MASp) in the dark (black trace) and under illumination with increasing wavelengths, from 365 nm (dark violet) to 635 nm (red). Inset, absorbance at 358 nm (*trans*-MASp$_{Cys}$ absorbance peak) as a function of the illumination wavelength. (B) Thermal stability of *cis*-MASp$_{Cys}$. After recording a UV-visible spectrum of MASp$_{Cys}$ in the dark (mainly *trans* state, black trace), MASp$_{Cys}$ was irradiated for 10 min by 365 nm light to yield mainly *cis*-MASp$_{Cys}$ (violet spectrum). Spectra in violet/grey gradation represent the gradual *cis*-to-*trans* transition of MASp$_{Cys}$ in the dark at different time points (up to ~8 h) post UV irradiation. Inset, evolution over time of the absorbance at 358 nm. Single-exponential fit (grey line) yielded a time constant $\tau = 259$ min (~4 h) for *cis*-to-*trans* MASp$_{Cys}$ thermal relaxation in the dark. (C) HPLC chromatograms monitored at the isosbestic point (440 nm) of the photostationary states (PSS) of MASp$_{Cys}$ in the dark (black trace), after illumination with 365 nm light (violet trace) and subsequent illumination with 525 nm light (green trace). (D) Experimental workflow for heterologous expression of GluN1/GluN2B receptors, MASp labeling and photomodulation of NMDAR activity in Xenopus oocytes. (E–H) MASp binds to the endogenous cysteine GluN2B-C395. (E) Left, schematic of a GluN1/GluN2B dimer with the positions of the free cysteines (i.e. not involved in disulfide bridges) highlighted in yellow. Right, inhibition traces by MK-801 (100 nM), an open channel pore blocker, of unlabeled ($-$ MASp, black) and labeled ($+$ MASp, grey) GluN1/GluN2B WT NMDARs kept in the dark (MASp in its *trans* state). (F) Superposition of the normalized MK801 inhibition traces from (E). Note the increase of MK-801 inhibition rate after labeling with MASp, indicating an increase in the open probability when GluN1/GluN2B NMDARs are conjugated with MASp. (G) Left, neutralization of the reactivity of C395, a cysteine located in GluN2B NTD-ABD linker, by mutation into a serine. Right, superposed and normalized MK-801 inhibition traces for labeled ($+$ MASp, grey) and unlabeled (- MASp, black) GluN1/GluN2B-C395S mutants. Note that on this mutant, labeling does not affect the rate of inhibition by MK-801. (H) Summary of the rates of inhibition by MK-801. GluN1/GluN2B WT: -MASP, $n = 5$ cells; +MASP, $n = 6$ cells. GluN1/GluN2B-C395S: -MASP, $n = 6$ cells; +MASP, $n = 5$ cells. Data are displayed as mean ± s.e.m. n.s., $P > 0.05$; *$P < 0.05$; multiple Mann–Whitney tests, $P$ values were adjusted for multiple comparisons using Bonferroni correction. Only the pre-selected indicated comparisons were performed. Exact $P$ values are displayed in Dataset EV1. Source data are available online for this figure.

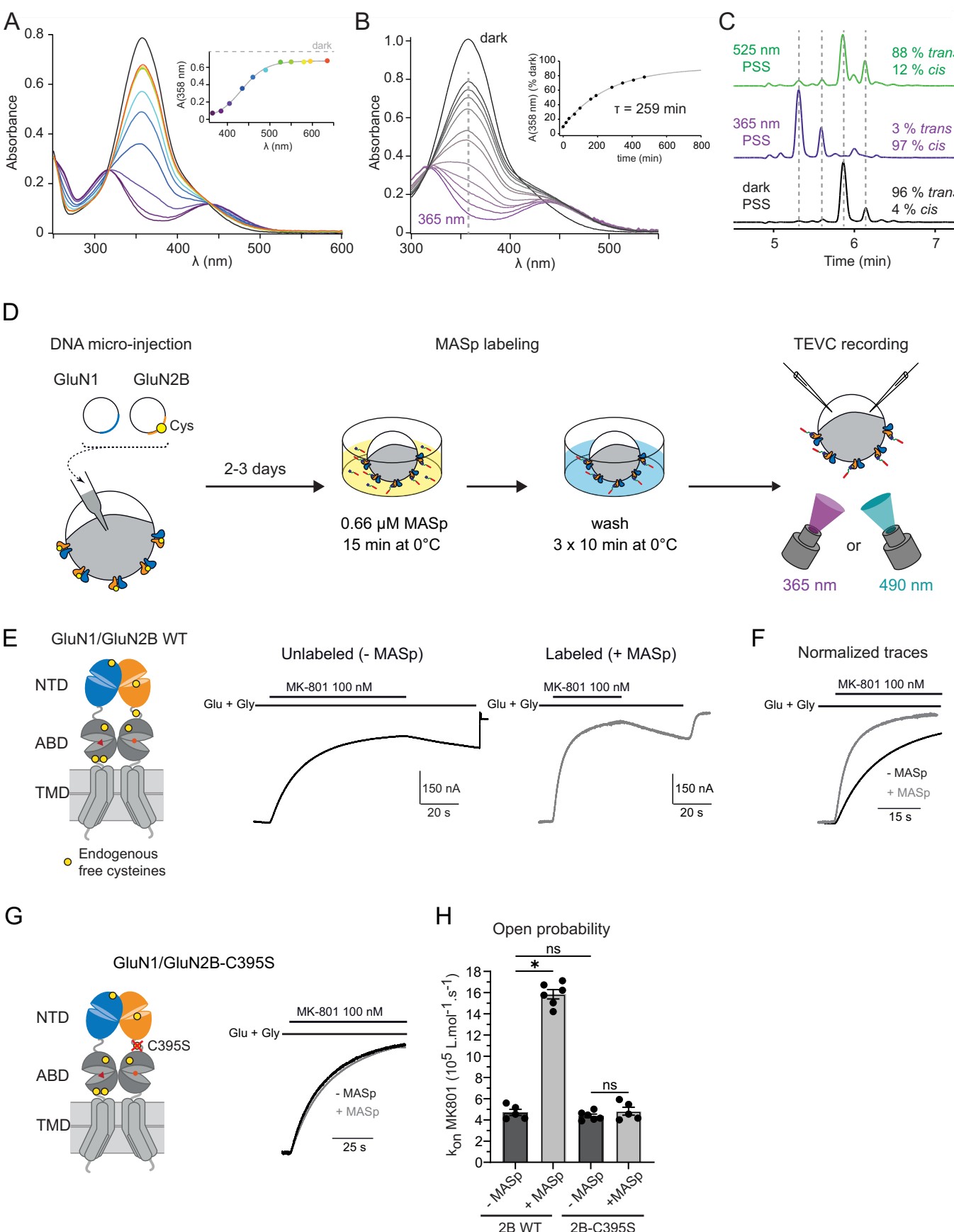

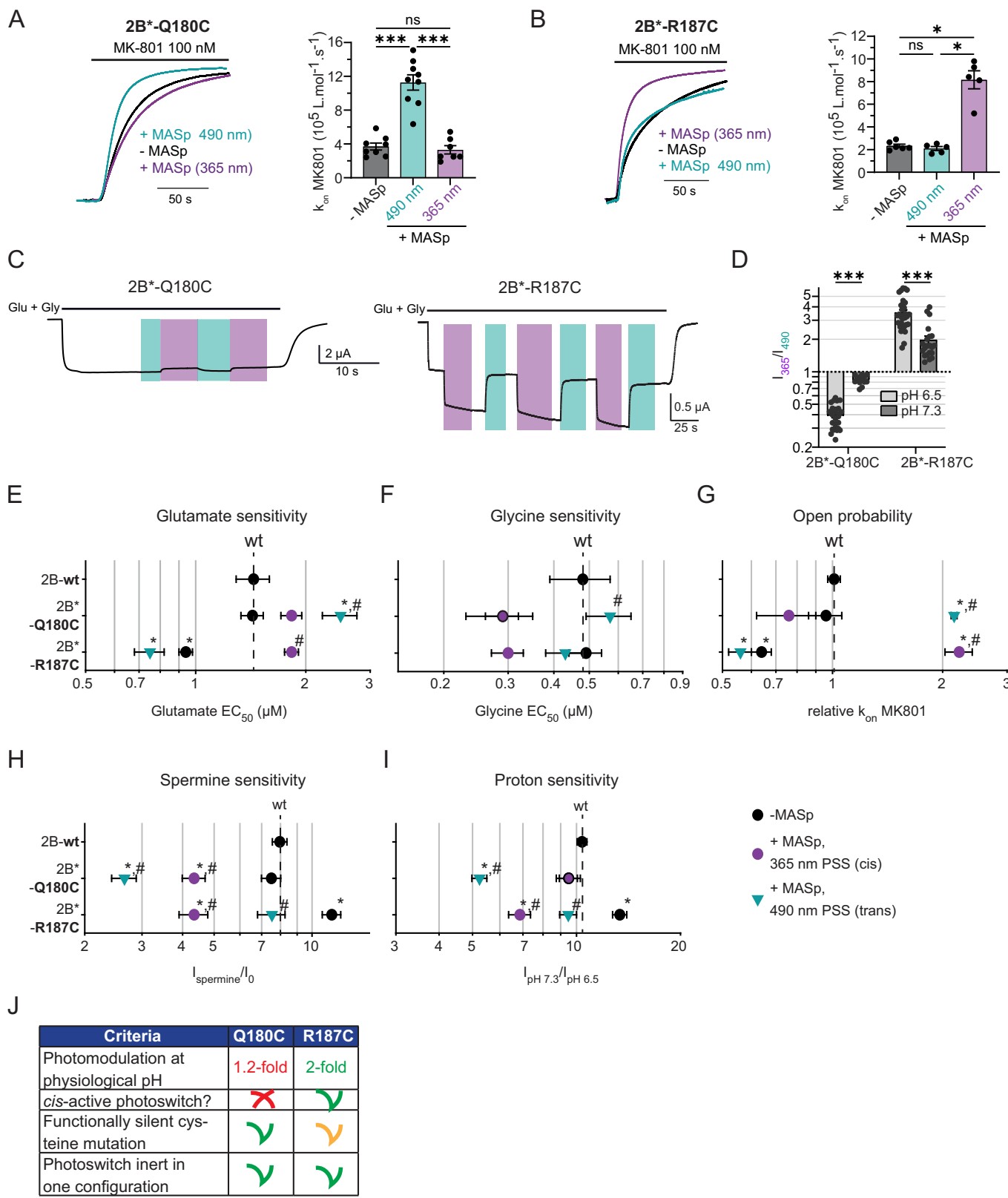

**Figure EV2.** (related to Fig. 1): **Photomodulation and pharmacological properties of GluN2B*-Q180C and -R187C mutants.**

(A, B) Evaluation of the relative Po of MASp-labeled GluN1/GluN2B*-Q180C (A) and GluN1/GluN2B*-R187C (B) under different light conditions. Left, superposed current traces following application of agonists and 100 nM MK-801 of unlabeled receptors (- MASp, black) and MASp-labeled receptors (+ MASp) under 365 (violet, mostly *cis*-MASP) or 490 nm light (blue-green, mostly *trans*-MASp). Right, summary of MK-801 inhibition rates. Kinetics of MK801 inhibition (mean ± s.e.m., number of cells): 2B*-Q180C, $k_{on} = 3.7 \pm 0.4 \ 10^5 \ L.mol^{-1}.s^{-1}$, $n = 8$ (- MASp); $11.3 \pm 0.9 \ 10^5 \ L.mol^{-1}.s^{-1}$, $n = 9$ (+ MASp, 490 nm), and $3.3 \pm 0.5 \ 10^5 \ L.mol^{-1}.s^{-1}$, $n = 7$ (+ MASp, 365 nm); 2B*-R187C, $k_{on} = 2.3 \pm 0.1 \ 10^5 \ L.mol^{-1}.s^{-1}$, $n = 6$ (- MASp); $2.1 \pm 0.2 \ 10^5 \ L.mol^{-1}.s^{-1}$, $n = 5$ (+ MASp, 490 nm), and $8.2 \pm 1.6 \ 10^5 \ L.mol^{-1}.s^{-1}$, $n = 5$ (+ MASp, 365 nm). n.s., $P > 0.05$; *$P < 0.05$; ***$P < 0.001$; multiple Mann–Whitney tests, $P$ values were adjusted for multiple comparisons using Bonferroni correction. When MASp was conjugated to GluN2B-Q180C in its *trans* configuration (under 490 nm light), channel Po increased by ~2-fold compared to unlabeled receptors, while labeling with *cis*-MASp (under UV light) produced no significant change in receptor activity. MASp thus acts as a *trans*-on photoswitch at position Q180C. In contrast, MASp conjugated to GluN2B-R187C had no effect on receptor channel Po under 490 nm light, while it increased Po by ~3.5-fold under UV-light. MASp thus acts as a *cis*-on photoswitch at position R187C (see Fig. 1E, right). (C, D) Photomodulation of MASp-labeled, GluN1/GluN2B*-Q180C and -R187C receptors at physiological pH. (C) Current traces from MASp-labeled GluN1/GluN2B*-Q180C (left) and GluN1/GluN2B*-R187C (right) at pH 7.3. (D) Summary of photomodulation ratios for the Q180C and R187C mutants at pH 6.5 and 7.3. Photomodulation values at pH 6.5 are from Fig. 1. ***$P < 0.001$ multiple Mann–Whitney tests, $P$ values were adjusted for multiple comparisons using Bonferroni correction. Only the pre-selected indicated comparisons were performed. Photomodulation values (mean ± s.e.m.) and number of cells are summarized in Appendix Table S1. (E–I) Determination of the relative open probability and sensitivity to different pharmacological agents of unlabeled (- MASp) and labeled (+ MASp) GluN1/GluN2B*-Q180C and GluN1/GluN2B*-R187C receptors compared to unlabeled WT GluN1/GluN2B receptors. *$P < 0.05$ between the condition and unlabeled WT GluN1/GluN2B receptors; #$P < 0.05$, between MASp-labeled mutant receptors under 365 or 490 nm light and unlabeled mutant receptors. One-way ANOVA followed by Tukey's test. Values (mean ± s.e.m.) are summarized in Appendix Table S2. (J) Pros and cons of photo-enhancing GluN2B-NMDARs via the Q180C or the R187C labeling position. The R187C is the most favored labeling position due to its strong photomodulation at physiological pH and the fact that MASp at this position is active in *cis*. Data are displayed as mean ± s.e.m. Exact $P$ values are summarized in Dataset EV1. Source data are available online for this figure.

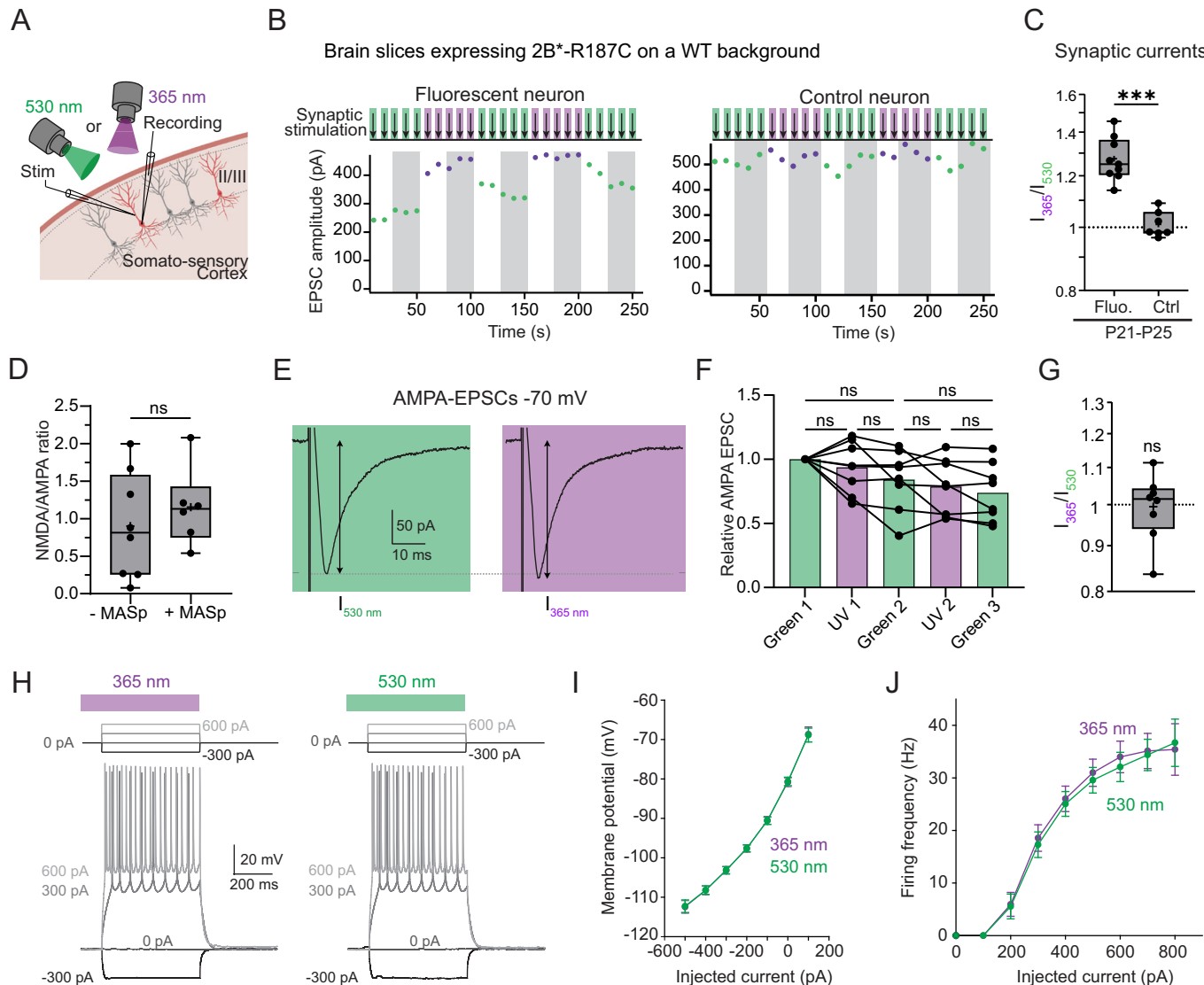

**Figure EV3.   (related to Fig. 3): Protocol of photomodulation of NMDA-EPSCs and determination of MASp background effects on brain slices.**

(**A**) Data were obtained from electroporated (fluorescent) and non-electroporated (non-fluorescent, control) layer II/II pyramidal neurons from the somatosensory cortex upon stimulation at the level of their apical dendrite. (**B**) Top, protocol of photomodulation of NMDAR synaptic currents (NMDA-EPSCs). Synaptic stimulation was performed continuously at 0.1 Hz. 525 nm (green bars) or 365 nm (violet bars) illumination started 1 s before synaptic stimulation and ended 200 ms after synaptic stimulation (1.2 s total). Illumination cycles consisted of alternations between 5 stimulations under 525 nm light and 5 stimulations under 365 nm light. Bottom, amplitudes of NMDA-EPSCs from a MASp-labeled, fluorescent (expressing GluN2B*-R187C) neuron from a P14 mouse under alternating 525 nm (green points) and 365 nm (violet points) illumination, showing reversible and reproducible photomodulation of NMDA-EPSCs. Given the slow reversion of EPSC amplitude by 525 nm light, only the three last EPSCs of each 5-stimulation cycle were considered for amplitude quantification and trace averaging (grey bars). (**C**) Photomodulation ratios ($I_{365}/I_{530}$) of NMDA-EPSCs of MASp-labeled, fluorescent and control neurons from P21-P25 animals. Photomodulation values (mean ± s.e.m.) and number of cells are summarized in Appendix Table S6. (**D–J**) MASp labeling has no or minimal impact on synaptic transmission and neuronal electrical properties. (**D**) NMDA/AMPA ratios for WT neurons either unlabeled (- MASp, $n = 8$ cells) or labeled ( + MASp, $n = 6$ cells) from P21-P25 animals. n.s., $P > 0.05$, Mann–Whitney test. (**E–G**) No photodependent effect of MASp labeling on AMPA-EPSCs. (**E**) Representative AMPA-EPSCs (recorded at −70 mV) from a MASp-labeled control (non-fluorescent) layer II/III cortical neuron under 530 nm (green bar, left) or 365 nm light (violet bar, right) in a slice from a P21 mouse. Same illumination conditions as in (**B**). (**F**) Summary of AMPA-EPSC amplitudes of MASp-labeled control neurons from P21-P22 mice under alternating 530 nm (green bars) and 365 nm (violet bars) illumination. Each dot represents the average amplitude of AMPA-EPSCs for each cycle for each cell. $n = 8$ cells. n.s., $P > 0.05$; Repeated measures Anova followed by Tukey's multiple comparison test. (**G**) Photomodulation ratios of AMPA-EPSCs from MASp-labeled control neurons, showing no significant photomodulation. Quantification of the photomodulation ratio was performed similarly to Fig. 3. $n = 8$ cells. n.s., $P > 0.05$; one sample Wilcoxon test against the value of 1. (**H–J**) No photodependence of the excitability properties of MASp-labeled control neurons. (**H**) Current-clamp voltage traces of a MASp-labeled, control neuron from a P23 mouse during 500 ms current injection steps under UV (365 nm, violet bar) and green light (530 nm, green bar). UV light was applied 50 ms before and during the current step, while green light was applied 2 s before and during the current step. The amount of current injected is written next to the corresponding voltage trace. (**I, J**) No light-dependence of the membrane potential ((**I**), $n = 7$–8 cells) and firing frequency ((**J**), $n = 5$–7 cells) as a function of injected current after MASp labeling of control neurons from P21-P24 animals. All recordings in brain slices were performed at physiological pH. Data are displayed as mean ± s.e.m. Box plots: centerlines show the median; crosses show the mean; box limits indicate the 25th and 75th percentiles; whiskers extend to the minimum and maximum. Exact $P$ values are summarized in Dataset EV1. Source data are available online for this figure.

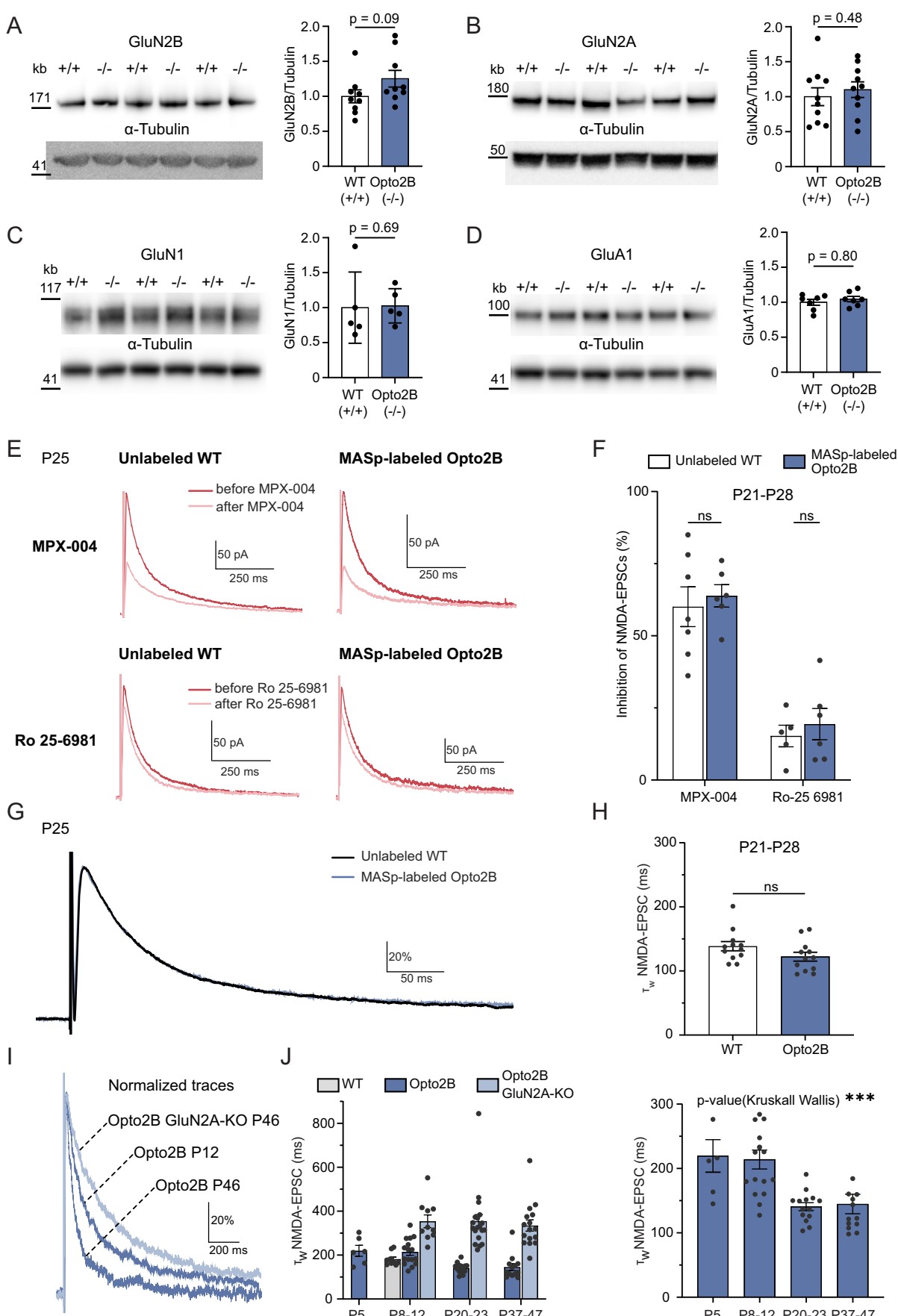

◀ **Figure EV4.** (related to Fig. 4): Similar NMDAR subunit profiles between Opto2B and WT mice.

(A–D) No significant difference of expression of NMDAR subunits GluN2B (A), GluN2A (B), and GluN1 (C), and of AMPAR subunit GluA1 (D) in P46-54 Opto2B mice (−/−) and their wt littermates (+/+). Quantification was performed by dividing the intensity of the subunit band (upper lane) by the intensity of the corresponding α-tubulin band (lower lane). Subunit/tubulin ratios were normalized to the average ratio of WT animals from the same blot. Uncropped blots are available in the Source Data files. $P > 0.05$ for all conditions, Mann–Whitney test. $n = 9$ WT and 9 Opto2B animals for GluN2B, 10 WT and 10 Opto2B animals for GluN2A, 5 WT and 5 Opto2B animals for GluN1, and 7 WT and 7 Opto2B animals for GluA1. (E) Top, superposition of NMDA-EPSCs of CA1 pyramidal neurons before and after application of MPX-004 (top) or Ro 25-6981 (bottom) from unlabeled WT (left) and MASp-labeled Opto2B mice (right) at age P25, recorded in the dark (MASp in its inactive, *trans* configuration). (F) Summary of NMDA-EPSC inhibition (%) by MPX-004 or Ro 25-6981 of CA1 pyramidal neurons from unlabeled WT (white) and MASp-labeled Opto2B mice (blue) at P21-P28. MPX-004: $n = 7$ cells for unlabeled WT and 6 cells for MASP-labeled Opto2B. Ro 25-6981: $n = 5$ cells for unlabeled WT and 6 for MASp-labeled Opto2B. n.s., $P > 0.05$; multiple Mann–Whitney tests, $P$ values were adjusted using Bonferroni correction. Only the pre-selected indicated comparisons were performed. (G) Superposition of normalized NMDA-EPSCs of CA1 pyramidal neurons from unlabeled WT (black) and MASp-labeled Opto2B mice (blue) at age P25. (H) Summary of NMDA-EPSC decay kinetics from unlabeled WT (white) and MASp-labeled Opto2B mice (blue) at P21-P28. $N = 12$ cells for unlabeled WT and 6 cells for MASp-labeled Opto2B. n.s., $P > 0.05$; Mann–Whitney test. Note the similar NMDA-EPSC decay time between WT and Opto2B animals, suggesting that the R187C mutation does not perturb developmental maturation of NMDA subtypes. (I) Superposition of normalized NMDA EPSCs of MASp-labeled CA1 pyramidal neurons from Opto2B mice at age P12 and P46, as well as from P46 Opto2B/GluN2A KO animals, recorded under green light (MASp in its inactive configuration). Each trace is the average of 12 individual EPSCs. (J) Left, decay time constants ($\tau_w$ NMDA-EPSC) of NMDA-EPSCs of MASp-labeled CA1 pyramidal neurons from WT (grey), Opto2B (dark blue) and Opto2B GluN2A-KO (light blue) mice at different age ranges. Note the absence of significant difference between the NMDA-EPSC decay kinetics of from WT and Opto2B animals at P8-P12, suggesting similar NMDAR subunit populations. NMDA-EPSC decay kinetics from Opto2B/GluN2A KO mice remained slow at older ages, consistently with what was previously shown for animals lacking GluN2A expression (Gray et al, 2011). Right, zoom on the Opto2B mouse condition for better visualization. ***$P < 0.001$, Kruskall-Wallis test on Opto2B condition. Note the acceleration of NMDA-EPSC decay time with age on Opto2B mice, which is consistent with what was observed in the literature on WT animals (Paoletti et al, 2013). Cell numbers for each condition are indicated in Dataset EV1. All the recordings on brain slices were performed at physiological pH. All data are displayed as mean ± s.e.m. Exact $P$ values are summarized in Dataset EV1. Source data are available online for this figure.

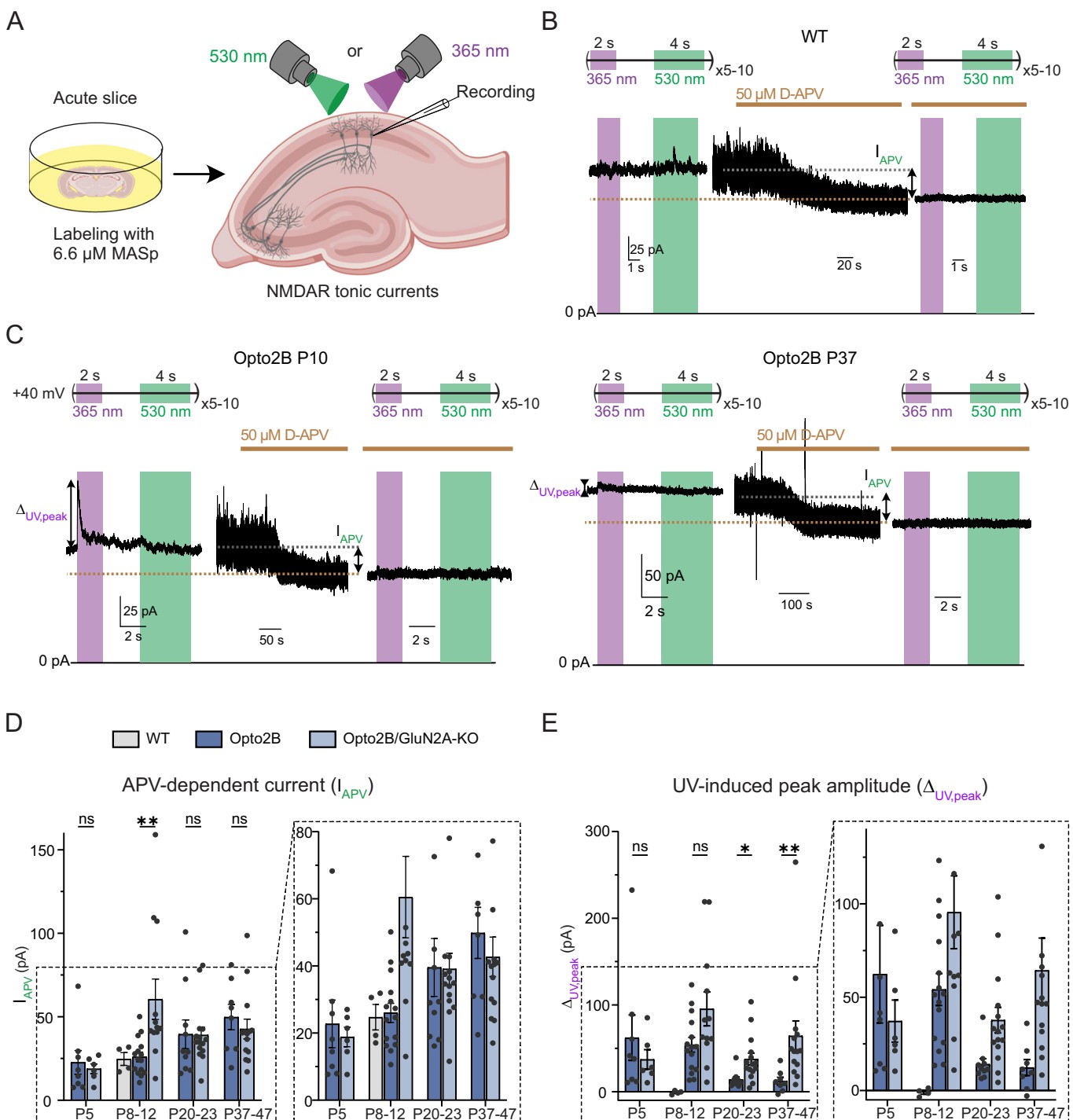

**Figure EV5. (related to Fig. 5): Determination of the photomodulation ratio of NMDA tonic currents.**

(A) Protocol of tonic current photomodulation. Note the absence of electrical stimulation: recorded tonic currents are mediated by low levels of tonic glutamate. See "Methods" and Main Text for more details. (B) Tonic current trace of a MASp-labeled, CA1 pyramidal neuron from a P10 WT mouse before (left) and during (middle and right) application of 50 µM D-APV. Left and right traces are the average of 5 to 10 traces. Note the absence of photomodulation. (C) Tonic current traces of MASp-labeled, CA1 pyramidal neurons from a P10 (left) or a P37 (right) Opto2B mouse before and during application of 50 µM D-APV. Left and right traces are the average of 5 to 10 traces. Displayed arrows indicate the currents measured in the following panels and used to calculate the photomodulation ratio (see also Fig. 3). (D) Amplitude of APV-sensitive currents (as shown in (A)) of MASp-labeled, CA1 pyramidal neurons from WT (grey), Opto2B (dark blue) and Opto2B GluN2A-KO (light blue) mice according to age ranges. Right, zoom on data for better visibility of bar graphs. (E) Amplitude of the UV-induced peak (as shown in (A)) of MASp-labeled, CA1 pyramidal neurons from WT (grey), Opto2B (dark blue) and Opto2B GluN2A-KO (light blue) mice according to age ranges. Right, zoom on data for better visibility of bar graphs. All recordings in brain slices were performed at physiological pH. Data are displayed as mean ± s.e.m. Cell numbers for each condition and exact P values are summarized in Dataset EV1. Source data are available online for this figure.

