## [Peer Review File · The EMBO Journal]

Optochemical profiling of NMDAR molecular diversity at synaptic and extrasynaptic sites

Antoine Sicard, Meilin Tian, Zakaria Mostefai, Sophie Shi, Cécile Cardoso, Joseph Zamith, Isabelle McCort-Tranchepain, Cecile Charrier, Pierre Paoletti, and Laetitia Mony

Corresponding author(s): Laetitia Mony (laetitia.mony@ens.psl.eu) , Pierre Paoletti (pierre.paoletti@ens.psl.eu)

Review Timeline:

Submission Date:	12th Dec 24
Editorial Decision:	3rd Feb 25
Revision Received:	6th Mar 25
Editorial Decision:	7th Apr 25
Revision Received:	12th May 25
Accepted:	30th May 25

Editor: Ioannis Papaioannou

Transaction Report:

Dear Dr. Mony,

Thank you for submitting your manuscript EMBOJ-2024-119900 for consideration by The EMBO Journal, and for your patience during peer review. Your manuscript has now been seen by three experts in the field, and we have received the full set of their well-informed and detailed reports, which are included below.

As you will see, the feedback of all three referees on your study is very supportive. They all find the manuscript interesting and addressing an important question in the field. They also point out that the experimental work is of high quality and the manuscript well-prepared. The referees also identify a few limitations and provide helpful and reasonable suggestions for the improvement of the study and the manuscript, which would further increase the impact of the work on the field.

Given the referees' positive comments and recommendations, I would like to invite you to submit a revised version of the manuscript along with a detailed point-by-point response addressing all referees' comments. I should add that it is The EMBO Journal policy to allow only a single round of major revision, and acceptance of your manuscript will therefore depend on the completeness of your responses in this revised version. Please let me know if you have any questions or comments that you would like to discuss with me.

We generally allow three months as standard revision time (May 2, 2025). As a matter of policy, competing manuscripts published during this period will not negatively impact our assessment of the conceptual advance presented by your study. However, we request that you contact us as soon as possible upon publication of any related work, to discuss how to proceed. Should you foresee a problem in meeting this three-month deadline, please let us know in advance and we may be able to grant an extension.

Thank you for the opportunity to consider your work for publication in The EMBO Journal. I look forward to your revision.

Best regards,

Ioannis

Instructions for preparing your revised manuscript

1. When you are ready to submit the revision, please upload:

- A Word file of the manuscript text (including legends of main Figures, EV Figures and Tables). Please make sure that changes are highlighted (or "tracked") to be clearly visible.

- Individual production-quality figure files (one file per figure). When assembling your figures, please refer to our figure preparation guidelines in order to ensure proper formatting and readability in print as well as on screen:

If the data shown in a figure are obtained from n {less than or equal to} 2, please use scatter plots showing the individual data points.

- i. the name of the statistical test used to generate error bars and P values
- ii. the number (n) of independent experiments (please specify technical or biological replicates) underlying each data point (discussion of statistical methodology can be reported in the Materials and Methods section, but figure legends should contain a basic description of n , P , and the test applied)
- iii. the nature of the bars and error bars (s.d., s.e.m.).

- A point-by-point response to the referees' comments, with a detailed description of the changes made (as a word file). All referees' concerns must be fully addressed and their suggestions taken on board. When preparing your letter of response to the referees' comments, please bear in mind that this will form part of the Review Process File and will therefore be available online

to the community. Please note that you have the possibility to opt out of the transparent process at any stage prior to publication by letting the editorial office know (contact@embojournal.org); if you do opt out, the Review Process File link will point to the following statement: "No Review Process File is available with this article, as the authors have chosen not to make the review process public in this case.". For more details on our Transparent Editorial Process, please visit our website: <https://www.embopress.org/page/journal/14602075/authorguide#transparentprocess>

- Expanded View (EV) files (replacing Supplementary Information) that are collapsible/expandable online. A maximum of 5 EV Figures can be typeset. EV Figures should be cited as "Figure EV1, Figure EV2" etc. in the text, and their respective legends should be included in the manuscript file after the legends of regular figures. See detailed instructions regarding Expanded View files here:

- For the figures that you do NOT wish to display as Expanded View figures, they should be bundled together with their legends in a single PDF file called "Appendix", which should start with a short Table of Contents (including page numbers). Appendix figures should be referred to in the main text as: "Appendix Figure S1, Appendix Figure S2" etc. Please see detailed instructions here: <https://www.embopress.org/page/journal/14602075/authorguide#expandedview>

- A complete author checklist, which you can download from our author guidelines (<https://www.embopress.org/page/journal/14602075/authorguide>). Please note that the checklist will also be part of the Review Process File.

2. Please note that no statistics should be calculated and shown in Figures if $n=2$. Please also note that each p value should be reported as an exact value.

3. Before submitting your revision, primary datasets (and computer code, where appropriate) produced in this study need to be deposited in appropriate public databases (see <https://www.embopress.org/page/journal/14602075/authorguide#dataavailability>). The accession numbers, database, and the specific URLs (links) should be listed in a formal "Data availability" section (placed after Methods), following the example below:

"The RNA-seq datasets produced in this study are available in the following database:
Gene Expression Omnibus GSE46843 (<https://www.ncbi.nlm.nih.gov/geo/query/acc.cgi?acc=GSE46843>)"

*** All links should resolve to a page where the data can be accessed. ***

*** Please remember to provide in the Data availability section of your revised manuscript reviewer passwords if the datasets are not yet public. ***

*** The Data Availability Section is restricted to new primary data that are part of this study. In case you have no data that require deposition in a public database, please state so instead of referring to the database: "Our study includes no data deposited in public repositories." under the heading "Data availability". ***

4. Please check that the title and the abstract of the manuscript are brief, yet explicit, even to non-specialists. The length of the title should not exceed 100 characters, and the abstract should be a single paragraph not exceeding 175 words.

5. Please also note our reference format: <https://www.embopress.org/page/journal/14602075/authorguide#referencesformat>.

7. Please remember: digital image enhancement is acceptable practice, as long as it accurately represents the original data and conforms to community standards. If a figure has been subjected to significant electronic manipulation, this must be noted in the figure legend or in the "Materials and Methods" section. The editors reserve the right to request original versions of figures and the original images that were used to assemble the figure.

8. Our journal encourages inclusion of data citations in the reference list to directly cite datasets that were obtained from public databases. Data citations in the article text are distinct from normal bibliographical citations and should directly link to the database records from which the data can be accessed. In the main text, data citations are formatted as follows: "Data ref: Smith et al, 2001" or "Data ref: NCBI Sequence Read Archive PRJNA342805, 2017". In the Reference list, data citations must be labeled with "[DATASET]". A data reference must provide the database name, accession number/identifiers, and a resolvable link to the landing page from which the data can be accessed at the end of the reference. Further instructions are available at: <https://www.embopress.org/page/journal/14602075/authorguide#referencesformat>.

9. We request authors to consider both actual and perceived competing interests. Please review our policy

(<https://www.embopress.org/page/journal/14602075/authorguide#conflictsofinterest>) and update your competing interests statement if necessary. Please name this section 'Disclosure and competing interests statement' and place it after the Acknowledgements section.

10. Please note that all corresponding authors are required to provide an ORCID ID upon submission of a revised manuscript (<https://orcid.org/>). Please find instructions on how to link your ORCID ID to your account in our manuscript tracking system in our Author guidelines (<https://www.embopress.org/page/journal/14602075/authorguide#authorshipguidelines>).

11. We use CRediT to specify the contributions of each author in the journal submission system. CRediT replaces the author contribution section, which should be removed from the manuscript. Please use the free text box to provide more detailed descriptions. See also guide to authors: <https://www.embopress.org/page/journal/14602075/authorguide#authorshipguidelines>.

13. We would also welcome the submission of cover suggestions or motifs to be used by our Graphics Illustrator in designing a cover.

14. Please use the link below to submit your revision:
<https://emboj.msubmit.net/cgi-bin/main.plex>

Referee #1:

In their manuscript, Sicard and colleagues describe a newly-developed photochemical tool (Opto2B) that enable specific and reversible modulation of GluN2B subunit diheteromers NMDA receptor. The tool is specific since other NMDA receptor subtypes, such as GluN2A/GluN2B triheteromers, are unaffected. Furthermore, they authors established the differential contribution of GluN2B diheteromers to synaptic and extrasynaptic NMDA receptor-mediated transmission over the first postnatal weeks. In mature hippocampal CA1 pyramidal cells, GluN2A-containing NMDA receptor represent a significant fraction of NMDA receptors, both at synaptic and extrasynaptic sites. Altogether, the study shows that GluN2A/GluN2B triheteromers likely constitute the majority of membrane NMDAR at mature brain stages, with GluN2A diheteromers and GluN2B diheteromers that remain present over time. This is an excellent and well-performed study that tackle an essential question in the field of neuroscience. The tool developed is of prime interest. Schematically, the study sheds new and exciting lights on a long-standing debate in the field, reshaping our understanding of NMDA receptor distribution at the surface of hippocampal neurons over development. The impact of the study is broad and major. I simply have two minor comments that could be clarified in a revised version.

i) Using Opto2B in GluN2A-KO mice, the authors nicely disentangle the impact of GluN2A subunit in the various opto-responses. This is well-described, for instance, in Figure 4. The authors indicate that the difference between Opto2B and Opto2B GluN2A-KO « increased with age ». The statistical support for such a claim is however lacking. In addition, this claim is possibly due to the high response variability at P8-12. I would thus tune down this claim. On the other hand, the authors very nicely show that the developmental decrease in Opto2B responses is similarly observed in Opto2B GluN2A-KO, suggesting that the sole presence of GluN2A subunit does not explain the developmental profile. The authors do not comment this important finding that should bring some nuance in the interpretation of the development course. Discussing this point in the manuscript will be highly valuable in my opinion. One may, for instance, propose that additional mechanisms are involved in the developmental decrease in Opto2B responses. As described by others, a developmental changes in the nanoscale organization of GluN2B/GluN2A-containing NMDA receptors could influence their transmission.

ii) The observation that GluN2A subunit contributes, as early as P12, in synaptic NMDAR pool whereas it only appears later in the extrasynaptic tool is surprising and of prime interest. The fact that Opto2B and Opto2B GluN2A-KO responses are undistinguishable until P20 suggests indeed the lack of GluN2A-NMDA receptor at extrasynaptic sites. However, a lower sensitivity of the tonic current response measures could account for this conclusion. The fact that the variability between Opto2B and Opto2B GluN2A-KO tonic current responses seem very different at P8-12 could suggest a contribution of GluN2A-NMDA receptor. Here, as above, the authors could mention putative limits that should tune down the claim that early in development GluN2A-NMDA receptor are solely synaptic and not extrasynaptic.

Referee #2:

This remarkable and excellent study investigates the molecular diversity of NMDA receptors (NMDARs) at synaptic and extrasynaptic sites using a newly developed optochemical tool, Opto2B. The authors aim to identify the contribution of GluN2B diheteromers to synaptic and extrasynaptic currents at various developmental stages, which has been a challenge with traditional pharmacological and genetic approaches. Opto2B, a photoswitchable spermine derivative, enables selective and reversible modulation of GluN2B diheteromers without affecting triheteromers. By using electrophysiological recordings in heterologous expression systems, HEK293 cells, cultured hippocampal neurons, and brain slices of Knock-In animals expressing the mutated GluN2B binding Opto2B, the authors demonstrate that GluN2B diheteromers contribute minimally to extrasynaptic NMDAR pools in adult hippocampal CA1 pyramidal cells, challenging previous assumptions. Their findings clarify discrepancies in the literature about the localization of GluN2B-containing receptors. The study also highlights the potential of Opto2B in precise molecular and spatiotemporal interrogation of NMDAR subtypes, which could be beneficial for future drug discovery and neurophysiological research.

This study is altogether exceptionally well performed and clean. It reports the very detailed and precise characterization of this new GluN2B di-heteromers light sensitive regulator. The authors cleverly exploit then this tool in a knock-in animal to establish unambiguously for the first time the respective contribution of GluN2B di heteromers in synaptic and extrasynaptic currents over development. Interestingly, the very fast control of GluN2B diheteromers by light allows separation of the two opposite effects of polyamine modulation: increase of channel P_o and decrease of glutamate potency.

The most striking novel result of their study is demonstrating that GluN2B diheteromers are present at CA3-CA1 synapses throughout development. However, they also observed a clear decrease of the extent of photomodulation with age reflecting a decreased contribution of synaptic GluN2B diheteromers during development, so that this population becomes a minority at adult stage. They further establish the near disappearance of extrasynaptic GluN2B diheteromers in mature neurons.

Altogether, I don't have much to add to study per se as it so extremely well performed and presented. The real strength of the study is that it presents a novel optochemical approach that provides unprecedented resolution in distinguishing NMDAR subtypes. Unlike traditional pharmacological agents (e.g., ifenprodil, zinc), Opto2B selectively modulates only GluN2B diheteromers. The authors combine structural biochemistry (NMR, mass spectrometry), cellular electrophysiology (*Xenopus* oocytes, HEK cells, cortical neurons), and *in vivo/ex vivo* recordings (hippocampal slices, optogenetic manipulations) and employ a rigorous validation process, including mutation analyses and control experiments for Opto2B.

Scientifically, the study refutes the long-held assumption that GluN2B diheteromers dominate at extrasynaptic sites at all stages of development.

To improve the Manuscript, I would suggest:

- 1) Explicit the technique used for selective expression of NMDAR triheteromers at the cell surface"
- 2) To clarify the paragraph of the GluN2A KO experiments which does not make full sense to me ("Compared to Opto2B slices, the photomodulation of NMDA-EPSC peak currents was markedly and systematically higher for Opto2B/GluN2A-KO slices, showing co-existence of GluN2B diheteromers with GluN2A-NMDARs at synaptic sites already at young ages") this should be better explained.
- 3) Clarify the paragraph ("This difference increased (DECREASED?) with age (Fig. 4C,D and Table S4), consistent with an increasing expression and incorporation of the GluN2A subunit in synaptic NMDARs during postnatal development), that seems to contradict the figure 4C.
- 4) Improve the discussion on GluN2A/B tri-heteromers. Indeed a weakness of the study is that it does not allow to distinguish between GluN2A di hets, and GLUNA/B tri hets.

More generally, but likely for future studies, it would be nice to use this tool to acutely potentiate GluN2B di heteromers to study their contribution plasticity phenomenon.

Referee #3:

Sicard et al. report a novel optogenetic tool, Opto2B, which incorporates GluN2B with R187C/C395S mutations labeled with MASp to modulate GluN2B activity using light. The tool is developed with precision and applied to neuronal systems, including the generation of GluN2B R187C knock-in mice, where NMDAR-EPSCs were characterized under light modulation. While the development and *in vivo* characterization of Opto2B are outstanding, the manuscript primarily contributes as a methodological advance rather than a significant biological insight. Considering the big potential of this novel tool, the development of tool by itself will advance the field significantly. Addressing assay inconsistencies and refining the writing will make the manuscript publishable.

Major Comments

Opto2B Tool Development and Applications

The methodology for Opto2B is well-conceived: GluN2B mutant expression → MASp incubation and wash → light stimulation and NMDAR activation → NMDAR current measurement. However, each step could potentially introduce unwanted effects. For future applicability, the manuscript should comprehensively document these potential artifacts.

Expression and Assembly of Mutant GluN2B

The expression and assembly of GluN2B with R187C/(C395S) mutations may differ from wild-type GluN2B. This difference could impact functional outcomes. The authors should show protein expression.

All experiments showing I366/I490 should include raw data values to evaluate its effect on basal currents.

In Fig. 1E, NMDAR activity is smaller in oocytes expressing R187C compared to Q180C. This suggests R187C might be constitutively suppressed by MASp conjugation, with 365 nm light relieving this suppression. While this result is unnecessary for confirming Opto2B incorporation, it is essential for demonstrating the lack of effect in basal conditions, as claimed by the authors.

In Vivo Characterization of GluN2B Mutant

Include total GluN2B protein levels via western blotting and NMDA/AMPA ratios in hippocampi from wild-type and R187C homozygous mice (Fig. 4).

Clarify the population of triheteromers in Fig. 2D. Control experiments should confirm the presence of only triheteromers, given the significant current amplitude differences in Fig. 2D traces. This is critical for ensuring accuracy in subsequent neuronal analyses.

Comparison of Oocyte and Neuron Assays

There are inconsistencies between the oocyte and neuron assays that should be addressed:

The statement that "MASp labeling by itself induced a marked (~3-fold) increase in receptor channel open probability (Po)" in oocytes but not in EPSCs needs further scrutiny. The authors claim this is due to MASp's non-reactivity with GluN2B-C395 in neurons. However, this conclusion is confounded by differences in recording conditions: Oocytes were recorded at -70 mV, while neurons were at +40 mV.

The authors should provide consistent analyses of glu/gly-evoked surface NMDAR responses and EPSCs at the same holding potential.

Examine the contribution of C395 to the 365/490 nm ratio. Comparing ratios from C395S-mutant oocytes and unmutated C395 neurons is problematic.

Fig. 1E: NMDAR responses in oocytes were recorded with continuous Glu/Gly application during wavelength changes, while EPSC recordings used continuous light with brief glutamate pulses from presynaptic terminals.

The authors should replicate the R187C oocyte data under EPSC recording conditions (continuous light with brief glutamate pulses at +40 mV) to facilitate direct comparison.

Writing and Organization

Clarification of Biological Questions

The abstract claims, "Our study clarifies decades of controversial research on extrasynaptic NMDARs," but this controversy is not introduced in the introduction. To strengthen the narrative: Move and expand the paragraph on page 13 discussing "proportions of GluN2A...subject to controversy" to the introduction. Clearly outline the controversies and how Opto2B addresses them.

Introduction

Paragraph 1 contains extraneous information (e.g., GABA receptors) that is irrelevant to the manuscript's focus. Streamline the introduction to focus on NMDAR-specific questions.

The statement, "These inherent limitations hamper drawing unambiguous conclusions," should specify which conclusions were affected and by what experiments or results.

Page 4, last sentence: The claim of "complete independence of GluN2A/GluN2B triheteromers" is inaccurate. Fig. 2B and 2E show small but significant differences in GluN2A diheteromers and GluN2B triheteromers after light activation. Revise accordingly.

Page 5, bottom two paragraphs: Swap their order for better logical flow.

Figure Legend: Clarify in the main text that the cysteine mutation was made on the GluN2B-C395S background, as noted in the figure legend.

Several typos, unwanted spaces, and misplaced commas need correction throughout the manuscript.

Suggestions for the Name "Opto2B"

The authors describe Opto2B as a PAM (or NAM) rather than an activator. The term "Opto-activator" might be misleading. Consider renaming it to more accurately reflect its function.

Manuscript EMBOJ-2024-119900
Response to reviewers

We thank the three reviewers for their very positive and constructive comments on our manuscript. Please find below a point-to-point response to all reviewer's comments. Page numbers refer to the revised version with tracked changes

Referee #1:

In their manuscript, Sicard and colleagues describe a newly-developed photochemical tool (Opto2B) that enables specific and reversible modulation of GluN2B subunit diheteromers NMDA receptor. The tool is specific since other NMDA receptor subtypes, such as GluN2A/GluN2B triheteromers, are unaffected. Furthermore, they authors established the differential contribution of GluN2B diheteromers to synaptic and extrasynaptic NMDA receptor-mediated transmission over the first postnatal weeks. In mature hippocampal CA1 pyramidal cells, GluN2A-containing NMDA receptor represent a significant fraction of NMDA receptors, both at synaptic and extrasynaptic sites. Altogether, the study shows that GluN2A/GluN2B triheteromers likely constitute the majority of membrane NMDAR at mature brain stages, with GluN2A diheteromers and GluN2B diheteromers that remain present over time. This is an excellent and well-performed study that tackle an essential question in the field of neuroscience. The tool developed is of prime interest.

Schematically, the study sheds new and exciting lights on a long-standing debate in the field, reshaping our understanding of NMDA receptor distribution at the surface of hippocampal neurons over development. The impact of the study is broad and major. I simply have two minor comments that could be clarified in a revised version.

- i) Using Opto2B in GluN2A-KO mice, the authors nicely disentangle the impact of GluN2A subunit in the various opto-responses. This is well-described, for instance, in Figure 4. The authors indicate that the difference between Opto2B and Opto2B GluN2A-KO « increased with age ». The statistical support for such a claim is however lacking. In addition, this claim is possibly due to the high response variability at P8-12. I would thus tune down this claim. On the other hand, the authors very nicely show tht the developmental decrease in Opto2B responses is similarly observed in Opto2B GluN2A-KO, suggesting that the sole presence of GluN2A subunit does not explain the developmental profile. The authors do not comment this important finding that should bring some nuance in the interpretation of the development course. Discussing this point in the manuscript will be highly valuable in my opinion. One may, for instance, propose that additional mechanisms are involved in the developmental decrease in Opto2B responses. As described by others, a developmental changes in the nanoscale organization of GluN2B/GluN2A-containing NMDA receptors could influence their transmission.

We have quantified the difference of photomodulation ratios between Opto2B and Opto2B/GluN2A-KO the following way: at P20-23, EPSCs are potentiated in average by 28% in Opto2B slices and by 63% in Opto2B/GluN2A-KO. There is thus a ~2.3-fold difference of photomodulation ratio between the two genotypes. At P37-47, EPSCs are potentiated in average by 14% in Opto2B slices and by 50% in Opto2B/GluN2A-KO. There is thus a ~3.5-fold difference of photomodulation ratio between the two genotypes. Hence the difference of photomodulation between Opto2B and Opto2B/GluN2A-KO does increase between the third

and the fifth week of age. This is in line with the well-described progressive enrichment with NMDARs containing GluN2A during postnatal development (and effect absent in Opto2B/GluN2AKO animals). However, we agree that this difference might be blurred by the variability of photomodulation in each age/genotype condition. Most importantly, we do make, nor need, to make a strong case out of it for drawing the two key conclusions that i) GluN2A-NMDARs are already present at synaptic sites at juvenile ages and ii) GluN2B diheteromers are decreasing in abundance at synaptic sites during postnatal development. To clarify this aspect, we have reformulated the paragraph in the Results section (page 15), focusing on the fact that, while there is a statistically significant decrease of NMDA-EPSC photomodulation with age in Opto2B animals, this was not the case in Opto2B/GluN2A-KO animals (although there is still a trend towards a decrease with age, see below)

As noticed by the reviewer, there is indeed a trend towards a decrease of EPSC photomodulation in Opto2B/GluN2A-KO animals. Since this decrease was not statistically significant after adjustment for multiple comparisons (Opto2B/GluN2A-KO, P8-12 vs P37-45, $p = 0.105$, two-sided Mann-Whitney test), we had decided not to talk about it in the initial version of the manuscript. However, as raised by the reviewer, this trend suggests that other mechanisms than GluN2A incorporation might explain the decrease of photomodulation in Opto2B animals. We now mention this decrease of NMDA-EPSC photomodulation in Opto2B/GluN2A-KO animals in the Results section (Page 15). Altogether, we have rephrased the whole paragraph on the comparison of NMDA-EPSC photomodulation between Opto2B and Opto2B/GluN2A-KO animals:

‘Compared to Opto2B slices, photomodulation of NMDA-EPSCs was much stronger in Opto2B/GluN2A-KO slices at all tested age ranges (P8 to P47; Fig. 4C,D and Appendix Table S4). This shows that GluN2B diheteromers already co-exist with GluN2A-NMDARs at synaptic sites at young ages. Contrary to Opto2B animals, the difference of NMDA-EPSC photomodulation between P8-12 and P37-47 in Opto2B/GluN2A-KO mice was not statistically significant ($p = 0.105$ after correction for multiple comparisons; two-sided Mann-Whitney test), indicating that the principal factor for photomodulation decrease in Opto2B mice is the increase of GluN2A-NMDAR contribution to synaptic currents. Our photomodulation results are consistent with the developmental acceleration of NMDA-EPSC decay kinetics in slices from Opto2B mice (an effect absent in Opto2B/GluN2A-KO animals; Fig. EV4E,F). However, despite being not statistically significant, there is a trend towards a decrease of NMDA-EPSC photomodulation with age in Opto2B/GluN2A-KO mice. This suggests that other factors than GluN2A incorporation might also contribute to the developmental decrease of NMDA-EPSC photomodulation in Opto2B mice (see Discussion).’

We have furthermore added the following text in the Discussion (page 20): “A slight decrease in NMDA-EPSC photomodulation with age was also observed in Opto2B/GluN2A-KO mice, suggesting contributions of other mechanisms than incorporation of GluN2A-NMDARs at synaptic sites. These include a potential progressive incorporation of GluN1 subunits containing the exon 5 splice variant (GluN1-1b), known to decrease polyamine potentiation (Traynelis *et al*, 1995) or developmental modifications of the nanoscale organization (Kellermayer *et al.*, Neuron 2018) of GluN2B diheteromers at the synapse which might affect the extent of EPSC photomodulation by variations in glutamate concentrations sensed by the receptors. Altogether our findings raise key questions that

remain to be addressed about the rules governing NMDAR subunit assembly and their specific targeting to distinct neuronal compartments.

- ii) The observation that GluN2A subunit contributes, as early as P12, in synaptic NMDAR pool whereas it only appears later in the extrasynaptic pool is surprising and of prime interest. The fact that Opto2B and Opto2B GluN2A-KO responses are undistinguishable until P20 suggests indeed the lack of GluN2A-NMDA receptor at extrasynaptic sites. However, a lower sensitivity of the tonic current response measures could account for this conclusion. The fact that the variability between Opto2B and Opto2B GluN2A-KO tonic current responses seem very different at P8-12 could suggest a contribution of GluN2A-NMDA receptor. Here, as above, the authors could mention putative limits that should tune down the claim that early in development GluN2A-NMDA receptors are solely synaptic and not extrasynaptic.

In fact, the sensitivity of tonic current responses to the photomodulation is stronger than the one of synaptic currents thanks to the sharp peak current induced by UV-light. Moreover, we verified that at young ages, the absolute amplitude of tonic currents between the two genotypes were not drastically different (similar range of current amplitudes; see former Fig. S10, now Fig. EV5). Detailed examination of our data at young postnatal ages clearly shows that, if anything, the extent of photomodulation in Opto2B/GluN2A-KO mice is similar, or even on average a bit smaller (see box plots Fig. 5B), than in opto2B mice. Moreover, statistical analysis clearly indicates no significant differences between the two genotypes at P5 and P8-P12 ($P > 0.99$ for both). In contrast, at older ages, differences between the two genotypes are easily detected, providing some sort of internal control of the relevance of the approach. Based on these results, we believe that we can safely conclude that GluN2A-containing receptors are not or minimally present at extrasynaptic sites at young postnatal ages. However, to go along the reviewer's concern in a more general manner (see also reviewer 3), we now have in the Discussion a dedicated paragraph on the limitations of the opto2B approach (page 22).

Referee #2:

This remarkable and excellent study investigates the molecular diversity of NMDA receptors (NMDARs) at synaptic and extrasynaptic sites using a newly developed optochemical tool, Opto2B. The authors aim to identify the contribution of GluN2B diheteromers to synaptic and extrasynaptic currents at various developmental stages, which has been a challenge with traditional pharmacological and genetic approaches. Opto2B, a photoswitchable spermine derivative, enables selective and reversible modulation of GluN2B diheteromers without affecting triheteromers. By using electrophysiological recordings in heterologous expression systems, HEK293 cells, cultured hippocampal neurons, and brain slices of Knock-In animals expressing the mutated GluN2B binding Opto2B, the authors demonstrate that GluN2B diheteromers contribute minimally to extrasynaptic NMDAR pools in adult hippocampal CA1 pyramidal cells, challenging previous assumptions. Their findings clarify discrepancies in the literature about the localization of GluN2B-containing receptors. The study also highlights the potential of Opto2B in precise molecular and spatiotemporal interrogation of NMDAR

subtypes, which could be beneficial for future drug discovery and neurophysiological research. This study is altogether exceptionally well performed and clean. It reports the very detailed and precise characterization of this new GluN2B di-heteromers light sensitive regulator. The authors cleverly exploit then this tool in a knock-in animal to establish unambiguously for the first time the respective contribution of GluN2-di heteromers in synaptic and extrasynaptic currents over development. Interestingly, the very fast control of GluN2B diheteromers by light allows separation of the two opposite effects of polyamine modulation: increase of channel P_o and decrease of glutamate potency.

The most striking novel result of their study is demonstrating that GluN2B diheteromers are present at CA3-CA1 synapses throughout development. However, they also observed a clear decrease of the extent of photomodulation with age reflecting a decreased contribution of synaptic GluN2B diheteromers during development, so that this population becomes a minority at adult stage. They further establish the near disappearance of extrasynaptic GluN2B diheteromers in mature neurons.

Altogether, I don't have much to add to study per se as it so extremely well performed and presented. The real strength of the study is that it presents a novel optochemical approach that provides unprecedented resolution in distinguishing NMDAR subtypes. Unlike traditional pharmacological agents (e.g., ifenprodil, zinc), Opto2B selectively modulates only GluN2B diheteromers. The authors combine structural biochemistry (NMR, mass spectrometry), cellular electrophysiology (*Xenopus* oocytes, HEK cells, cortical neurons), and in vivo/ex vivo recordings (hippocampal slices, optogenetic manipulations) and employ a rigorous validation process, including mutation analyses and control experiments for Opto2B.

Scientifically, the study refutes the long-held assumption that GluN2B diheteromers dominate at extrasynaptic sites at all stages of development.

To improve the Manuscript, I would suggest:

- 1) Explicit the technique used for selective expression of NMDAR triheteromers at the cell surface"

As suggested, we have added an explanation of the technique used for selective membrane expression of triheteromeric NMDARs (page 9): "For that purpose, we used a previously published approach based on the endoplasmic-reticulum (ER) retention signals (hereby named r1 and r2) of GABA_B receptors fused to the GluN2 subunits (Stroebel *et al*, 2014) (see also (Hansen *et al*, 2014)). NMDAR complexes containing either two GluN2-r1 or two GluN2-r2 subunits are retained in the ER and cannot reach the cell surface. On the contrary, association of r1 with r2 masks the ER retention signals allowing selective cell surface expression of triheteromeric NMDARs containing one GluN2-r1 and one GluN2-r2 subunit (Stroebel *et al*, 2014; Hansen *et al*, 2014). »

- 2) To clarify the paragraph of the GluN2A KO experiments which does not make full sense to me

("Compared to Opto2B slices, the photomodulation of NMDA-EPSC peak currents was markedly and systematically higher for Opto2B/GluN2A-KO slices, showing co-existence of GluN2B diheteromers with GluN2A-NMDARs at synaptic sites already at young ages") this should be better explained.

We have rephrased our sentence to make it clearer (page 15): "Compared to Opto2B slices, photomodulation of NMDA-EPSCs was much stronger in Opto2B/GluN2A-KO slices at all

tested age ranges (P8 to P47; Fig. 4C,D and Appendix Table S4). This shows that GluN2B diheteromers already co-exist with GluN2A-NMDARs at synaptic sites at young ages.”

3) Clarify the paragraph (“This difference increased (DECREASED?) with age (Fig. 4C,D and Table S4), consistent with an increasing expression and incorporation of the GluN2A subunit in synaptic NMDARs during postnatal development), that seems to contradict the figure 4C.

In this sentence, we refer to the difference in photomodulation between Opto2B and Opto2B/GluN2A-KO slices at a given age range. At P20-23, EPSCs are potentiated in average by 28% in Opto2B slices and by 63% in Opto2B/GluN2A-KO. There is thus a ~2.3-fold difference of photomodulation ratio between the two genotypes. At P37-47, EPSCs are potentiated in average by 14% in Opto2B slices and by 50% in Opto2B/GluN2A-KO. There is thus a ~3.5-fold difference of photomodulation ratio between the two genotypes. Hence the difference of photomodulation between Opto2B and Opto2B/GluN2A-KO increases between the third and the fifth week of age. To clarify this point we have rephrased the paragraph relative to the comparison of NMDA-EPSCs between Opto2B and Opto2B/GluN2A-KO genotypes (page 15, see response to first reviewer).

‘Compared to Opto2B slices, photomodulation of NMDA-EPSCs was much stronger in Opto2B/GluN2A-KO slices at all tested age ranges (P8 to P47; Fig. 4C,D and Appendix Table S4). This shows that GluN2B diheteromers already co-exist with GluN2A-NMDARs at synaptic sites at young ages. Contrary to Opto2B animals, the difference of NMDA-EPSC photomodulation between P8-12 and P37-47 in Opto2B/GluN2A-KO mice was not statistically significant ($p = 0.105$ after correction for multiple comparisons; two-sided Mann-Whitney test), indicating that the principal factor for photomodulation decrease in Opto2B mice is the increase of GluN2A-NMDAR contribution to synaptic currents. Our photomodulation results are consistent with the developmental acceleration of NMDA-EPSC decay kinetics in slices from Opto2B mice (an effect absent in Opto2B/GluN2A-KO animals; Fig. EV4E,F). However, despite being not statistically significant, there is a trend towards a decrease of NMDA-EPSC photomodulation with age in Opto2B/GluN2A-KO mice. This suggests that other factors than GluN2A incorporation might also contribute to the developmental decrease of NMDA-EPSC photomodulation in Opto2B mice (see Discussion).’

4) Improve the discussion on GluN2A/B tri-heteromers. Indeed a weakness of the study is that it does not allow to distinguish between GluN2A di hets, and GLUN2A/B tri hets.

More generally, but likely for future studies, it would be nice to use this tool to acutely potentiate

GluN2B di heteromers to study their contribution plasticity phenomenon.

We are actively working on the development of a new tool that allows selective photomodulation of GluN2A/GluN2B triheteromers. It will be particularly interesting to implement this tool for studies on synaptic plasticity. In the revised Discussion (page 22), we better point out that fact the we still have no easy way to distinguish between GluN2A diheteromers and GluN2A/GluN2B triheteromers, a limitation of the current study.

Referee #3:

Sicard et al. report a novel optogenetic tool, Opto2B, which incorporates GluN2B with R187C/C395S mutations labeled with MASp to modulate GluN2B activity using light. The tool is developed with precision and applied to neuronal systems, including the generation of GluN2B R187C knock-in mice, where NMDAR-EPSCs were characterized under light modulation. While the development and in vivo characterization of Opto2B are outstanding, the manuscript primarily contributes as a methodological advance rather than a significant biological insight. Considering the big potential of this novel tool, the development of tool by itself will advance the field significantly. Addressing assay inconsistencies and refining the writing will make the manuscript publishable.

Major Comments

Opto2B Tool Development and Applications

The methodology for Opto2B is well-conceived: GluN2B mutant expression → MASp incubation and wash → light stimulation and NMDAR activation → NMDAR current measurement. However, each step could potentially introduce unwanted effects. For future applicability, the manuscript should comprehensively document these potential artifacts.

We believe that many of the potential artifacts of the different steps have already been tackled in the current manuscript, with extensive data presented in supplementary figures. We have, in particular, shown that MASp could react to an endogenous cysteine located on GluN2B (C395) and that such reaction increased the channel open probability without conferring any light sensitivity to the channel (Figs. 1E and former Fig. S1E-G, now Fig. EV1E-G). This is why we performed our study on the GluN2B-C395S background. In former Fig. S2E-I (now Fig. EV2E-I), we have tested the impact of the Q180C and R187C cysteine mutations, as well as the impact of MASp attachment on these cysteine mutations under different light conditions, on receptor open probability (P_o) and pharmacology. In addition, the absence of effect of light stimulation on *Xenopus* oocytes expressing wt GluN2B, GluN2C- and GluN2D- containing NMDARs and treated with MASp clearly shows that there are no artefactual effects of light on NMDA currents. Overall, we genuinely think that we have performed a robust and fairly comprehensive analysis of the various mutant receptors with their appropriate controls in the various experimental conditions, thus providing a solid and coherent set of results and conclusions. Of course, as for any new biological tool, Opto2B has limitations. We now have a dedicated paragraph in the Discussion section (page 22) that lists such potential limitations.

Expression and Assembly of Mutant GluN2B

The expression and assembly of GluN2B with R187C/(C395S) mutations may differ from wild-type

GluN2B. This difference could impact functional outcomes. The authors should show protein expression.

The mutation R187C-C395S might indeed affect NMDAR expression in *Xenopus* oocytes, although we haven't observed major differences in current amplitudes between wt and cysteine-mutated GluN2B-NMDARs. However, the level of receptor expression usually does

not impact the receptor channel properties. As a confirmation, we have plotted the level of UV-induced potentiation (I_{365}/I_{490}) against the current amplitude under 490 nm light (basal current state, a proxy of NMDAR expression level) for GluN1/GluN2B-R187C-C395S receptors (Fig. r1). We confirm that the level of photomodulation is independent of the receptor level of expression.

Figure r1: Photomodulation of GluN1/GluN2B-R187C-C395S receptors does not depend on receptor expression levels in *Xenopus* oocytes. Photomodulation ratio ($I_{365 \text{ nm}} / I_{490 \text{ nm}}$) as a function of the NMDA current in basal conditions, i.e. under 490 nm light illumination (I_0). The photomodulation ratio stays constant around 3-4-fold regardless of the amount of NMDA current, i.e regardless of the level of NMDAR expression.

All experiments showing I_{366}/I_{490} should include raw data values to evaluate its effect on basal currents.

Raw current amplitudes under 365 and 490 nm do not allow us to infer the impact of MASp on basal currents because these latter depend on both channel P_o and receptor expression. To study the impact of MASp on the receptor basal properties, we have studied several parameters (P_o , agonist sensitivities, proton sensitivity) following labeling with MASp under 365 nm and 490 nm light conditions, and compared them to the properties of unlabeled receptors (former Fig. S2, now Fig. EV2). We showed that, in terms of P_o , MASp-labeled, GluN1/GluN2B-R187C-C395S receptors were in their basal state under 490 nm light (former Fig. S2B, now Fig. EV2B), while GluN1/GluN2B-Q180C-C395S receptors were in their basal state under 365 nm light. In addition, MASp labeling had a low impact on receptor agonist and allosteric modulation properties in their basal P_o state (former Fig. S2E-I, now Fig. EV2E-I).

In Fig. 1E, NMDAR activity is smaller in oocytes expressing R187C compared to Q180C. This suggests R187C might be constitutively suppressed by MASp conjugation, with 365 nm light relieving this suppression. While this result is unnecessary for confirming Opto2B incorporation, it is essential for demonstrating the lack of effect in basal conditions, as claimed by the authors.

Following up on the previous point, we have shown in former Fig. S2 (now Fig. EV2) that MASp acts as a potentiator for both GluN2B-Q180C-C395S and GluN2B-R187C-C395S mutants, but that its light dependence is different. For GluN2B-Q180C-C395S, MASp maintains the receptor in its basal state under UV light and potentiates receptor activity under green light (*trans*-on photoswitch, Fig. 1E). On the contrary, for GluN2B-R187C-C395S,

MASp maintains the receptor in its basal state under green light and potentiates receptor activity under UV light (*cis*-on photoswitch, Fig. 1E). The sizes of the currents in Fig. 1E were normalized to the basal state of each receptor and they are actually of similar size for each construct (compare 2Bwt, any light condition, to 2B*-Q180C under UV light and 2B*-R187C under green light). There is thus no suppressing effect of MASp conjugation of 2B*-R187C activity.

In Vivo Characterization of GluN2B Mutant

Include total GluN2B protein levels via western blotting and NMDA/AMPA ratios in hippocampi from wild-type and R187C homozygous mice (Fig. 4).

Since we were mostly interested in the composition of NMDARs in pyramidal cells of the CA1 region of the hippocampus, we preferred characterizing NMDAR content of our mutant mice using electrophysiology rather than by Western blots, which cannot easily discriminate between cell-types and synapses. By monitoring the EPSC decay times and the effects of subtype-selective NMDAR pharmacology, we showed that the proportions of synaptic GluN2A- and GluN2B-containing receptors were similar between wt and GluN2B-R187C homozygous animals, arguing against a major redistribution of NMDAR subunits in the mutant animals (former Fig. S9, now Fig. EV4). We however agree that for future studies on the role of GluN2B-diheteromers on plasticity or behavior, a broader characterization of NMDAR subunit expression levels (and potentially other proteins) will be important.

Clarify the population of triheteromers in Fig. 2D. Control experiments should confirm the presence of only triheteromers, given the significant current amplitude differences in Fig. 2D traces. This is critical for ensuring accuracy in subsequent neuronal analyses.

The reviewer is right to notice that currents from N1/2B*-R187C-r1/2B*-R187C-r2 were on average lower than the ones from N1/2B-r1/2B*-R187C-r2 and N1/2A-r1/2B*-R187C-r2 (Fig. r2). However, we made sure that the populations recorded in Fig. 2D are truly triheteromeric. Indeed, before performing our photomodulation experiments, we systematically checked the absence of escape currents from the corresponding diheteromers (we always do so when we study triheteromeric receptors to validate the approach; see, for example, our past publications Stroebel et al., J Neurosci 2014 or Serraz et al., Neuropharmacology, 2016). As mentioned in our Methods section, this was made by injecting a mixture GluN1/GluN2-r1 and GluN1/GluN2-r2 constructs, which are not supposed to reach the cell surface, in *Xenopus* oocytes. On the day of the experiment, currents from “escaped” diheteromers (i.e. from GluN1/GluN2-r1 and GluN1/GluN2-r2 constructs) were monitored and experiments were only performed if these escape currents represent <10% of the triheteromeric currents (i.e. currents from the GluN1/GluN2-r1/GluN2-r2 condition). This precision has now been added to the Methods section (page 29):

“On the day of the experiment, currents from escaped diheteromeric receptors (i.e. from oocytes injected with GluN1-6A/GluN2-r1 or GluN1-6A/GluN2-r2) were systematically monitored and subsequent experiments were only performed if these currents were < 10% of triheteromeric currents.”

To further support our claim, we show below (Fig. r2A-C) that the level of photomodulation observed in the different constructs using the ER retention system is independent of the

basal NMDAR current level, ruling out a contribution of escaped receptors in oocytes displaying higher expression levels.

Figure r2 : Photomodulation of NMDARs using an ER-retention system does not depend on receptor expression. Photomodulation ratios ($I_{365\text{ nm}} / I_{490\text{ nm}}$) of constructs expressed using r1 and r2 ER retention motifs to control association of GluN2 subunits, as a function of the NMDA current in basal conditions, i.e. under 490 nm light illumination (I_0). Related to manuscript Fig. 2D.

Comparison of Oocyte and Neuron Assays

There are inconsistencies between the oocyte and neuron assays that should be addressed: The statement that "MASp labeling by itself induced a marked (~3-fold) increase in receptor channel open probability (P_o)" in oocytes but not in EPSCs needs further scrutiny. The authors claim this is due to MASp's non-reactivity with GluN2B-C395 in neurons. However, this conclusion is confounded by differences in recording conditions: Oocytes were recorded at -70 mV, while neurons were at +40 mV.

The authors should provide consistent analyses of glu/gly-evoked surface NMDAR responses and EPSCs at the same holding potential.

Examine the contribution of C395 to the 365/490 nm ratio. Comparing ratios from C395S-mutant oocytes and unmutated C395 neurons is problematic.

The GluN2B-selective potentiating effect of spermine is voltage-independent (Traynelis et al., 1995; Paoletti et al., Neuron 1995) and we can safely assume that this is also the case for MASp potentiation. We therefore do not expect major differences in photomodulation between -70 and +40 mV holding potentials.

Regardless of the holding potential parameter, our finding that GluN2B-C395 is likely not accessible to MASp in neurons comes from the comparison of photomodulation ratios between HEK cells and cultured cortical neurons, which were all recorded in the same conditions (same extracellular and intracellular solutions) and at the same holding potential of -70 mV (former Fig. S8, now Appendix Fig. S5).

In both HEK cells and dissociated cortical neurons, we have investigated the role of C395 by comparing the amount of photomodulation of cells expressing GluN1/GluN2B-R187C (intact C395) and GluN1/GluN2B-R187C-C395S, and labeled with MASp (former Fig. S8, now Appendix Fig. S5). In HEK cells, photomodulation of MASp-labeled, GluN1/GluN2B-R187C receptors was lower than the one of GluN1/GluN2B-R187C-C395S receptors (former Fig. S8A, now Appendix Fig. S5A). This is consistent with the fact that conjugation of MASp to C395 increases P_o and thus decreases the range of possible potentiation. On the contrary, in dissociated neurons, we did not observe any difference of photomodulation between the GluN1/GluN2B-R187C and GluN1/GluN2B-R187C-C395S conditions. This suggests that MASp has not conjugated to C395 in neurons. This is explicitly mentioned in

our manuscript (Results section page 14). In brain slices, we observed similar photomodulation ratios in slices from animals electroporated with GluN2B-R187C-C395S and from GluN2B-R187C KI (Opto2B) animals. This makes us think that C395 is also not accessible to MASp in *ex vivo* experiments (also explicitly mentioned in the manuscript). We agree that the arguments leading to this conclusion are indirect but investigating the occupancy of this cysteine in native conditions is beyond the scope of this paper.

Finally, we would like to stress that our conclusions about the relative abundance of GluN2B diheteromers in brain slices does not stem from the comparison of photomodulation values between *in vitro* and *ex vivo* conditions, but between the photomodulation ratios obtained in Opto2B vs Opto2B/GluN2A-KO mice (which were obtained in the same recording conditions). We agree that there are several parameters (pH, temperature, receptor micro-environment) that could impact absolute values of photomodulation. However, the fact that we obtained similar maximal values of photomodulation (~3-fold) *in vitro* (in HEK cells and dissociated neurons) and *ex vivo* in brain slices highlights the high transposability of our assay across different experimental conditions. To make this point clearer we have reorganized and rephrased the corresponding paragraph pages 15-16:

“At early postnatal ages (before P12), a large, and quantitatively similar, UV potentiation of NMDA tonic currents (>2.5-fold) was observed for Opto2B mice and Opto2B/GluN2A-KO animals (Fig. 5A,B and Appendix Table S4), indicating a lack of GluN2A-NMDARs. Therefore, at early postnatal ages, extrasynaptic NMDARs appear to be almost entirely GluN2B diheteromers. Interestingly, the maximal photopotential observed on tonic NMDAR currents was similar to that observed *in vitro* in GluN2B-R187C transduced cultured neurons ($I_{365\text{ nm}}/I_{530\text{ nm}} = 3.30 \pm 0.39$, $n = 15$ for P8-P12 Opto2B slices, Fig. 5B and Appendix Table S4, vs $I_{365\text{ nm}}/I_{525\text{ nm}} = 3.04 \pm 0.25$, $n = 20$ for GluN2B-R187C-expressing cultured cortical neurons, Appendix Fig. S5 and Appendix Table S1), showing high transposability of our tool across experimental conditions. “

Fig. 1E: NMDAR responses in oocytes were recorded with continuous Glu/Gly application during wavelength changes, while EPSC recordings used continuous light with brief glutamate pulses from presynaptic terminals.

The authors should replicate the R187C oocyte data under EPSC recording conditions (continuous light with brief glutamate pulses at +40 mV) to facilitate direct comparison.

We have tested different conditions of illumination in HEK cells (illumination before, during, and before+during agonist application) (former Fig. S3C,D, now Appendix Fig. S1C,D). The condition that mimics the illumination conditions during EPSC recordings is the “Resting+active” condition (illumination before and during the NMDA EPSC). We did not observe any difference in photomodulation between the different illumination conditions (former Fig. S3C,D, now Appendix Fig. S1C,D).

Writing and Organization

Clarification of Biological Questions

The abstract claims, "Our study clarifies decades of controversial research on extrasynaptic NMDARs," but this controversy is not introduced in the introduction. To strengthen the narrative:

Move and expand the paragraph on page 13 discussing "proportions of GluN2A...subject to controversy" to the introduction. Clearly outline the controversies and how Opto2B addresses them.

We have now explicitly explained the controversies in the Introduction by adding the following sentences:

'In addition, the subcellular distribution of these various receptor subtypes, i.e. their distribution between synaptic and extrasynaptic compartments, is also contentious. In the adult forebrain, many studies argue for an enrichment of GluN2A-containing NMDARs (GluN2A-NMDARs) at synapses, and an enrichment of GluN2B-NMDARs at extrasynaptic sites (Fellin *et al*, 2004; Papouin *et al*, 2012; Scimemi *et al*, 2004; Tovar & Westbrook, 1999; Köhr, 2006; Groc *et al*, 2009). However, other studies found no difference in NMDAR subunit composition between synaptic and extrasynaptic sites (Petralia *et al*, 2010; Harris & Pettit, 2007; Meur *et al*, 2007; Mohrmann *et al*, 2000).'

Introduction

Paragraph 1 contains extraneous information (e.g., GABA receptors) that is irrelevant to the manuscript's focus. Streamline the introduction to focus on NMDAR-specific questions.

The ability to manipulate membrane receptors with subunit stoichiometry resolution is a question of general interest in the field of neuronal signaling (and beyond). Moreover the challenges to be met to overcome the current methodological limitations are shared between signaling proteins and modalities. Accordingly, we believe that this introductory start with a fairly broad vision is quite relevant and should appeal to many readers, within the field of glutamate receptors and beyond.

The statement, "These inherent limitations hamper drawing unambiguous conclusions," should specify which conclusions were affected and by what experiments or results.

We have rephrased the corresponding paragraph to make it clearer:

"Such controversies likely stem, in part, from inherent limitations of genetic and pharmacological studies, which hamper drawing unambiguous conclusions about the prevalence of the different receptor populations. Indeed, genetic manipulation of GluN2A or GluN2B subunits indifferently affects diheteromers and triheteromers. Similarly, available pharmacological agents, such as the 'GluN2A selective' inhibitors zinc and TCN-201, or the 'GluN2B selective' inhibitor ifenprodil (and derivatives), poorly discriminate between their respective diheteromers and triheteromers (Hansen *et al*, 2014; Hatton & Paoletti, 2005; Stroebel *et al*, 2018, 2014). »

Page 4, last sentence: The claim of "complete independence of GluN2A/GluN2B triheteromers" is inaccurate. Fig. 2B and 2E show small but significant differences in GluN2A diheteromers and GluN2B triheteromers after light activation. Revise accordingly.

We have deleted the word 'complete'. For the sake of accuracy, we would like to point out that illumination has no effect on GluN2A/GluN2B triheteromers (Fig 2E, bar on the far right). The reviewer likely referred to the middle bar which corresponds to recombinant receptors

with two different GluN2B subunits, as was purposely designed, but with no equivalent in native conditions.

Page 5, bottom two paragraphs: Swap their order for better logical flow.

We have swapped the two paragraphs. It does make reading more fluid, thank you.

Figure Legend: Clarify in the main text that the cysteine mutation was made on the GluN2B-C395S background, as noted in the figure legend.

Done. Please note that now that the two paragraphs of page 5 (now page 6) have been swapped (see previous point), the mention of the GluN2B-C395S background appears before the description of the screening.

Several typos, unwanted spaces, and misplaced commas need correction throughout the manuscript.

We have carefully checked the entire manuscript and hopefully corrected the typos.

Suggestions for the Name "Opto2B"

The authors describe Opto2B as a PAM (or NAM) rather than an activator. The term "Optoactivator" might be misleading. Consider renaming it to more accurately reflect its function.

The term 'Optoactivator' is indeed misleading since the receptors are modulated, but not directly activated, by light. The tool that we have designed allows optical modulation of GluN2B diheteromers with high selectivity, so we think that the name Opto2B is appropriate.

Dear Laetitia,

Thank you again for submitting your revised manuscript EMBOJ-2024-119900R for consideration by The EMBO Journal, and for your patience. Your revised manuscript has now been seen by two of the three referees that had previously assessed the original version of your manuscript, and we have received their comments, which you can find below.

I am glad to say that referee #1 finds all his/her previously raised concerns successfully addressed and endorses the publication of this high-quality and relevant study. Referee #3 also mentions that some of their previous comments have been adequately addressed, and that the manuscript is now overall improved. This referee also explains that he/she finds the development and in vivo characterization of Opto2B commendable and supports the publication of the manuscript in The EMBO Journal provided that the following data are provided:

- (1) raw current values (mean {plus minus} SEM) for all experiments showing I366/I490 ratios
- (2) quantitative assessment of at least total GluN2B and Opto2B expression levels in both WT and knock-in mice.

The referee explains in detail why he/she finds these additions essential, and -following our discussions within our team and with the referees- we would like to invite you to provide these data in a final version of your manuscript before it is accepted for publication in The EMBO Journal. Please include in your resubmission a point-by-point response fully addressing the remaining comments of referee #3.

From the editorial side, there are also a few changes and corrections we need you to make in the final version of your manuscript. Please include in your resubmission a cover letter detailing how the points below are addressed:

- I would kindly request you to consider whether "Article" or "Method" is the most suitable manuscript type for this work. It certainly combines aspects of both, but we think that "Method" might be more appropriate. If you agree, please change the type in the manuscript submission system or let us know and we will change it for you.
- We noticed that figure callouts are missing for Figure 6B. Please make sure that all Figure panels are called out in your revised manuscript.
- The source file names, titles, legends, and manuscript callouts all need to be updated to "Dataset EV1" instead of "Appendix Table S5".
- There are 2 "Appendix Figure S5" listed in the Table of Contents of the Appendix file, instead of "Appendix Figure S6".
- The materials and methods need to be described in the manuscript using our structured methods format, which is now required for all research articles. According to this format, the Methods section includes a single "Reagents and Tools Table" -listing key reagents, experimental models, software and relevant equipment including their sources and relevant identifiers- followed by a "Methods and Protocols" section describing the methods. Please download and fill our Reagents and Tools Table template (.docx), which you can find in our author guide:
<https://www.embopress.org/page/journal/14602075/authorguide#structuredmethods>. When submitting your revised manuscript, please do not include the Reagents and Tools Table in the Methods section of the manuscript but instead upload it as a separate file choosing the file type "Reagent Table".
- Source data files need to be reorganized/saved in a single folder per figure and then uploaded as a .zip folder. For example, all source data files for Figure 1 panels need to be saved in a single folder that must be zipped and uploaded as "SD Figure 1.zip" folder. For EV and/or Appendix Figures, please ZIP together all their source data in a single folder ("SD EV Appendix.zip").
- Please note that EMBO press papers are accompanied online by:
 - A) a short (2 sentences) summary of the findings and their significance,
 - B) 2-5 short bullet points highlighting the key results, and
 - C) a synopsis image in .jpg or .png format that is exactly 550 pixels wide and 300-600 pixels high (the height is variable). Please note that the text needs to be legible at the final size.Please upload this information along with your revised manuscript (the text for A and B should be provided in a separate Word file).
- During our routine pre-acceptance checks, our data editors have raised the following queries regarding figures, data, and legends. Please make sure that all requests below are completely addressed in the final version of your manuscript:
 1. Please note that the box plots need to be defined in terms of minima, maxima, centre, bounds of box and whiskers, and percentile in the legends of Figures 3C, E; 4D, 5B, EV3 C, D, G; supplementary figures 3H.
 2. Please note that the error bars are not defined in the legends of Figures EV1 H; supplementary figures 4B, D, E, G, H.

- The manuscript section order should be corrected as follows: Title page - Abstract & Keywords - Introduction - Results - Discussion - Methods - Data Availability - Acknowledgements - Disclosure and Competing Interests Statement - References - Figure Legends - main Table(s) (if there are any) - Expanded View Figure Legends.

Please also note that as part of the EMBO publications' Transparent Editorial Process, The EMBO Journal publishes online a Peer Review File along with each accepted manuscript. This File will be published in conjunction with your paper and will include the referee reports, your point-by-point response and all pertinent correspondence relating to the manuscript. You can opt out of this by letting the editorial office know (contact@embojournal.org). If you do opt out, the Peer Review File link will point to the following statement: "No Peer Review File is available with this article, as the authors have chosen not to make the review process public in this case."

We look forward to seeing a final version of your manuscript as soon as possible. Please let us know if you have any questions and use this link to submit your revision: <https://emboj.msubmit.net/cgi-bin/main.plex>.

Best regards,

Ioannis

Referee #1:

The authors have perfectly addressed all my comments. The study is of high quality and relevance for the field of neuroscience and beyond.

Referee #3:

The authors have addressed adequately some of my previous comments, and the manuscript has been improved. However, I believe the manuscript still requires further effort to establish Opto2B as a reliable tool, as well as to demonstrate its utility in enabling novel biological discoveries, as intended. In particular, the authors have not adequately shown and discussed the limitations of their tool. Every experimental tool comes with both strengths and weaknesses, and it is entirely acceptable to acknowledge such limitations with results. In fact, doing so is essential to ensure that readers are not misled about the tool's capabilities. I would like to emphasize that I remain impressed by the development and in vivo characterization of Opto2B, and I commend the authors for their technical achievement. I support the publication of this manuscript in its current journal, provided that the authors present their findings with precision and clarity to allow readers to better evaluate its applicability and to foster trust in its utility within the scientific community.

Expression and Assembly of Mutant GluN2B

The altered expression level of the Opto2B tool compared to wild-type (WT) GluN2B significantly impacts the utility and interpretation of the data presented. Therefore, it is critical that the authors provide the raw current values (average {plus minus} SEM) for all experiments showing 1366/1490 ratios, as previously requested. This is particularly important because the authors use this tool to assess the endogenous status of NMDAR triheteromers in knock-in mice. If Opto2B expression is substantially lower in heterologous cells (oocytes, HEK cells) and in the mouse brain compared to WT GluN2B, then the conclusions drawn in this study pertain specifically to the knock-in condition and may not accurately reflect the WT scenario.

Ironically, the authors have clarified now in the introduction that the main controversies in the field "likely stem, in part, from inherent limitations of genetic and pharmacological studies, which hamper drawing unambiguous conclusions about the prevalence of the different receptor populations." This concern is also reiterated in their response and in the abstract." If Opto2B expression in knock-in mice does not replicate GluN2B expression in WT mice, then the receptor composition characterized in this study is reflective of an altered expression context, which could bias interpretation. Thus, quantification of GluN2B and

Opto2B expression in both WT and knock-in mice is essential to clarify the conclusions under the circumstance.

In summary, the manuscript must include (1) Raw current values (mean {plus minus} SEM) for all experiments showing I366/I490 ratios and (2) Quantitative assessment of at least total GluN2B and Opto2B expression levels in both WT and knock-in mice.

Finally, while the authors have responded to all previous comments on this point, their responses did not directly address the core of the concern. Additionally, I would like to clarify that I did not raise questions regarding the role of GluN2B dimeromers in plasticity or behavior, though the authors responded in that direction.

Other Points

It is good to see the added statement regarding the control experiment in the Methods section (page 29).

If this review is made public, the authors' explanation regarding voltage dependency at -70 and +40 mV is acceptable, though the statement is somewhat bold. However, if the review remains private, the manuscript should include the following clarification, as stated in the authors' response: "The GluN2B-selective potentiating effect of spermine is voltage-independent (Traynelis et al., 1995; Paoletti et al., Neuron 1995), and we can safely assume that this is also the case for MASp potentiation. We therefore do not expect major differences in photomodulation between -70 and +40 mV holding potentials."

Lastly, the writing has improved significantly, and the current version presents a clearer logical flow.

Manuscript EMBOJ-2024-119900
Response to reviewers

We thank the two reviewers for their positive comments regarding the improvement of our paper during the first round of revisions. In this second round of revisions, we have addressed the remaining concerns raised by reviewer #3. Please find below a point-by-point response.

Referee #1:

The authors have perfectly addressed all my comments. The study is of high quality and relevance for the field of neuroscience and beyond.

We thank referee #1 for his/her very positive comment and for appreciating our study.

Referee #3:

The authors have addressed adequately some of my previous comments, and the manuscript has been improved. However, I believe the manuscript still requires further effort to establish Opto2B as a reliable tool, as well as to demonstrate its utility in enabling novel biological discoveries, as intended. In particular, the authors have not adequately shown and discussed the limitations of their tool. Every experimental tool comes with both strengths and weaknesses, and it is entirely acceptable to acknowledge such limitations with results. In fact, doing so is essential to ensure that readers are not misled about the tool's capabilities. I would like to emphasize that I remain impressed by the development and in vivo characterization of Opto2B, and I commend the authors for their technical achievement. I support the publication of this manuscript in its current journal, provided that the authors present their findings with precision and clarity to allow readers to better evaluate its applicability and to foster trust in its utility within the scientific community.

Expression and Assembly of Mutant GluN2B

The altered expression level of the Opto2B tool compared to wild-type (WT) GluN2B significantly impacts the utility and interpretation of the data presented. Therefore, it is critical that the authors provide the raw current values (average {plus minus} SEM) for all experiments showing I366/I490 ratios, as previously requested. This is particularly important because the authors use this tool to assess the endogenous status of NMDAR triheteromers in knock-in mice. If Opto2B expression is substantially lower in heterologous cells (oocytes, HEK cells) and in the mouse brain compared to WT GluN2B, then the conclusions drawn in this study pertain specifically to the knock-in condition and may not accurately reflect the WT scenario.

Ironically, the authors have clarified now in the introduction that the main controversies in the field "likely stem, in part, from inherent limitations of genetic and pharmacological studies, which hamper drawing unambiguous conclusions about the prevalence of the different receptor populations." This concern is also reiterated in their response and in the abstract." If Opto2B expression in knock-in mice does not replicate GluN2B expression in WT mice, then the receptor composition characterized in this study is reflective of an altered expression context, which could bias interpretation. Thus, quantification of GluN2B and Opto2B expression in both WT and knock-in mice is essential to clarify the conclusions under the circumstance.

In summary, the manuscript must include (1) Raw current values (mean {plus minus} SEM) for all experiments showing I₃₆₆/I₄₉₀ ratios and (2) Quantitative assessment of at least total GluN2B and Opto2B expression levels in both WT and knock-in mice.

Finally, while the authors have responded to all previous comments on this point, their responses did not directly address the core of the concern. Additionally, I would like to clarify that I did not raise questions regarding the role of GluN2B diheteromers in plasticity or behavior, though the authors responded in that direction.

We have compiled all raw current data for every NMDA construct on which photomodulation was tested, in *Xenopus* oocytes, HEK cells and cultured neurons. Average basal current values (I_{490 nm} or I_{525 nm}) ± s.e.m. are now displayed in Appendix Tables S3, S4 and S5, respectively. Individual current values are displayed in the Source Data alongside the corresponding photomodulation values. We have furthermore compiled NMDA EPSC values obtained from WT, Opto2B and Opto2B/GluN2A-KO, which are displayed in Appendix Table S8, with individual values in Source Data. Please note that in *Xenopus* oocytes, because of the strong (~4-fold) UV-induced current potentiation of GluN1/GluN2B*-R187C receptors, we chose to work with low basal currents to allow the full range of photomodulation. We did not observe differences in NMDA-EPSC amplitudes between WT and Opto2B animals.

Furthermore, as requested, we have quantified total expression of NMDAR GluN1, GluN2A and GluN2B subunits, as well as AMPAR GluA1 subunit by Western blot analysis, in brains from adult Opto2B animals and their WT littermates. For all tested subunits, we did not observe any significant difference of expression between WT and Opto2B animals (Fig. r1). These new results have been included in the Main Text (p. 13) and displayed in new Expanded Figure EV4.

Figure r1: Example blots (left) and quantification (right) of GluN2B (A), GluN2A (B), GluN1 (C), and GluA1 subunit expression in brains of P46-54 Opto2B mice (-/-) and their WT littermates (+/+). Quantification was performed by dividing the intensity of the subunit band (upper lane) by the intensity of the corresponding α -tubulin band (lower lane). Subunit/tubulin ratios were normalized to the average ratio of WT animals from the same blot.

Other Points

It is good to see the added statement regarding the control experiment in the Methods section (page 29).

If this review is made public, the authors' explanation regarding voltage dependency at -70 and +40 mV is acceptable, though the statement is somewhat bold. However, if the review remains private, the manuscript should include the following clarification, as stated in the authors' response: "The GluN2B-selective potentiating effect of spermine is voltage-independent (Traynelis et al., 1995; Paoletti et al., Neuron 1995), and we can safely assume that this is also the case for MASp potentiation. We therefore do not expect major differences in photomodulation between -70 and +40 mV holding potentials."

Lastly, the writing has improved significantly, and the current version presents a clearer logical flow.

Thank you for appreciating the clarifications we have made in the first round of revisions. The review will be public so our statement that MASp potentiation should be similar at -70 and +40 mV will be available to the reader.

Dear Laetitia,

I am happy to inform you that we have now received the input of referee #3 who reassessed the revised version of your manuscript, and they confirm that all remaining concerns have now been fully addressed (please see their comment below).

I am therefore pleased to inform you that your manuscript has been accepted for publication in The EMBO Journal. Thank you very much for comprehensively addressing all initially raised referees' concerns and editorial/formatting requests, and congratulations on an excellent work!

If you have any questions, please do not hesitate to contact the Editorial Office. Thank you for your contribution to The EMBO Journal. Working with you has been a pleasure.

Best regards,

Ioannis

Referee #3:

The authors have fully addressed all of my comments. I support the publication of this manuscript in its current form.
